# Integrating Large Language Models in Causal Discovery: A Statistical Causal Approach

**Masayuki Takayama**                              *masayuki-takayama@biwako.shiga-u.ac.jp*
*Data Science and AI Innovation Research Promotion Center, Shiga University*
*National Institute of Science and Technology Policy (NISTEP)*

**Tadahisa Okuda**
*Department of Health Data Science, Tokyo Medical University*
*Graduate School of Medicine Human Health Sciences, Kyoto University*

**Thong Pham**
*Data Science and AI Innovation Research Promotion Center, Shiga University*
*Graduate School of Medicine Human Health Sciences, Kyoto University*
*Center for Advanced Intelligence Project, RIKEN*

**Tatsuyoshi Ikenoue**
*Data Science and AI Innovation Research Promotion Center, Shiga University*
*Graduate School of Medicine Human Health Sciences, Kyoto University*

**Shingo Fukuma**
*Graduate School of Medicine Human Health Sciences, Kyoto University*
*Department of Epidemiology Infectious Disease Control and Prevention, Hiroshima University Graduate School of Biomedical and Health Sciences*
*Data Science and AI Innovation Research Promotion Center, Shiga University*

**Shohei Shimizu**                              *shohei-shimizu@ds.sanken.osaka-u.ac.jp*
*SANKEN, The University of Osaka*
*Faculty of Data Science, Shiga University*
*Center for Advanced Intelligence Project, RIKEN*
*Graduate School of Medicine Human Health Sciences, Kyoto University*
*Institute for the Advanced Study of Human Biology, Kyoto University*
*National Institute of Science and Technology Policy (NISTEP)*

**Akiyoshi Sannai**                              *sannai.akiyoshi.7z@kyoto-u.ac.jp*
*Department of Physics, Kyoto University*
*Data Science and AI Innovation Research Promotion Center, Shiga University*
*Graduate School of Engineering, The University of Tokyo*
*Center for Advanced Intelligence Project, RIKEN*
*Research and Development Center for Large Language Models, National Institute of Informatics*
*National Institute of Science and Technology Policy (NISTEP)*

**Reviewed on OpenReview:** *https://openreview.net/forum?id=Reh1S8rxfh*

## Abstract

In practical statistical causal discovery (SCD), embedding domain expert knowledge as constraints into the algorithm is important for reasonable causal models reflecting the broad knowledge of domain experts, despite the challenges in the systematic acquisition of background knowledge. To overcome these challenges, this paper proposes a novel method for causal inference, in which SCD and knowledge-based causal inference (KBCI) with a large language model (LLM) are synthesized through "statistical causal prompting (SCP)" for LLMs and prior knowledge augmentation for SCD. The experiments in this work have

revealed that the results of LLM-KBCI and SCD augmented with LLM-KBCI approach the ground truths, more than the SCD result without prior knowledge. These experiments have also revealed that the SCD result can be further improved if the LLM undergoes SCP. Furthermore, with an unpublished real-world dataset, we have demonstrated that the background knowledge provided by the LLM can improve the SCD on this dataset, even if this dataset has never been included in the training data of the LLM. For future practical application of this proposed method across important domains such as healthcare, we also thoroughly discuss the limitations, risks of critical errors, expected improvement of techniques around LLMs, and realistic integration of expert checks of the results into this automatic process, with SCP simulations under various conditions both in successful and failure scenarios. The careful and appropriate application of the proposed approach in this work, with improvement and customization for each domain, can thus address challenges such as dataset biases and limitations, illustrating the potential of LLMs to improve data-driven causal inference across diverse scientific domains.

The code used in this work is publicly available[1].

# 1 Introduction

## 1.1 Background

Understanding causal relationships is key to comprehending basic mechanisms in various scientific fields. The statistical causal inference framework, which is widely applied in areas such as medical science, economics, and environmental science, aids this understanding. However, traditional statistical causal inference methods generally rely on the assumed causal graph for determining the existence and strength of causal impacts. To overcome this challenge, data-driven algorithmic methods have been developed as statistical causal discovery (SCD) methods (Spirtes et al., 2000; Chickering, 2002; Silander & Myllymäki, 2006; Shimizu et al., 2006; Hoyer et al., 2008; Shimizu et al., 2011; Yuan & Malone, 2013; Huang et al., 2018; Xie et al., 2020; Rolland et al., 2022; Tu et al., 2022). In addition, benchmark datasets have been published for the evaluation of SCD methods (Mooij et al., 2016; Käding & Runge, 2023).

Despite advancements in SCD algorithms, data-driven acquisition of causal graphs without domain knowledge can be inaccurate. This is generally attributed to a mismatch between assumptions in SCD and real-world phenomena (Reisach et al., 2021). Moreover, obtaining sufficient experimental and systematic datasets for causal inference is difficult, whereas observational datasets, which are prone to selection bias and measurement errors, are more readily accessible (Abdullahi et al., 2020). Consequently, to obtain more persuasive and reasonable causal models supported by sufficient knowledge of domain experts, the augmentation with domain knowledge plays a critical role (Rohrer, 2017). In addition, with respect to efficiency and precision in the SCD, the importance of incorporating constraints on trivial causal relationships into the SCD algorithms has been highlighted (Inazumi et al., 2010; Chowdhury et al., 2023). Causal learning software packages have been augmented with prior knowledge, as demonstrated in "causal-learn" [2], and "LiNGAM" [3] (Zheng et al., 2023b; Ikeuchi et al., 2023).

Moreover, the systematic acquisition of domain expert knowledge is a challenging task. Although several examples of constructing directed acyclic graphs (DAGs) by domain experts exist, as demonstrated in health services research (Rodrigues et al., 2022), practical methods for this process have not been proposed.

The scenario has recently changed with rapid progress in the development of high-end large language models (LLMs). Owing to their high performance in the application of their domain knowledge acquired from vast amounts of data in the pretraining processes (OpenAI, 2023; Touvron et al., 2023; Gemini Team, Google, 2023), it is expected that LLMs can also be applied to causal reasoning tasks. In fact, several studies have

---

[1] https://github.com/mas-takayama/LLM-and-SCD
[2] https://github.com/py-why/causal-learn
[3] https://github.com/cdt15/lingam

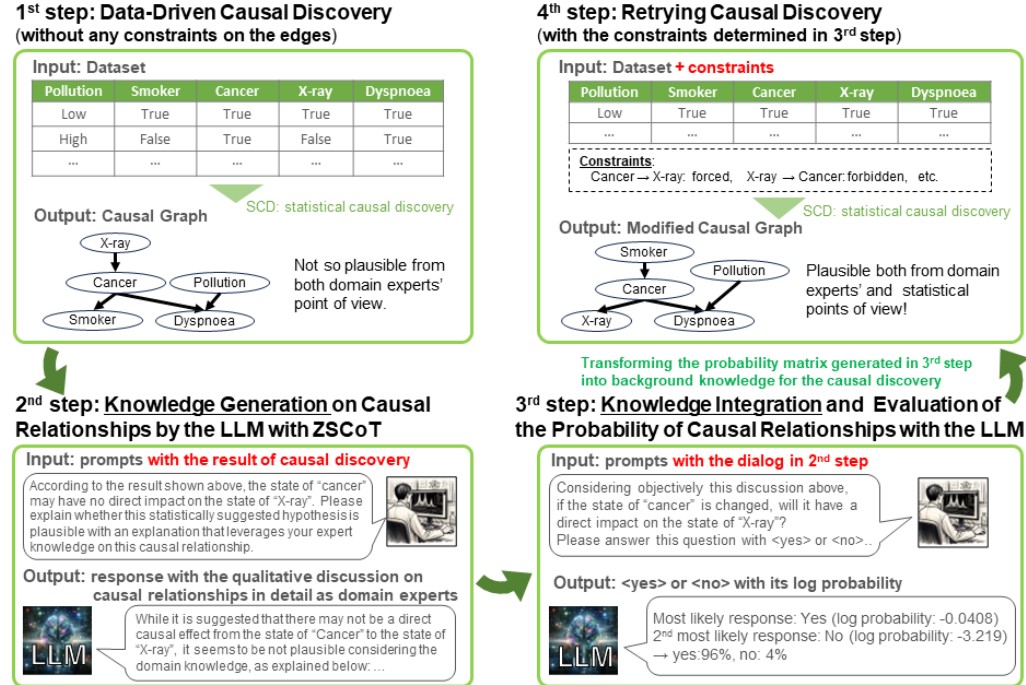

Figure 1: Overall framework of the statistical causal prompt in a large language model (LLM) and statistical causal discovery (SCD) with LLM-based background knowledge.

reported the trial results of LLM knowledge-based causal inference[4] (LLM-KBCI) (Jiang et al., 2023; Jin et al., 2024; Kiciman et al., 2024; Zečević et al., 2023; Jiralerspong et al., 2024; Zhang et al., 2024), and in particular, the performance enhancement in nonparametric SCD with the guides by LLMs was confirmed (Ban et al., 2023; Vashishtha et al., 2023; Khatibi et al., 2024). However, it remains unclear whether the enhancement in SCD accuracy with background knowledge augmented by LLMs is robustly observed when the task of the SCD depends on closed data uncontained in the pretraining datasets of LLMs.

## 1.2 Central Idea of Our Research

On the basis of the rapidly evolving techniques in the context of causal inference with LLMs, a novel methodology for SCD is proposed in this paper, in which the LLM prompted with the results of the SCD without background knowledge evaluates the probability of the causal relationships considering both the domain knowledge and the statistical characteristics suggested by SCD (Figure 1).

In the first step, an SCD is executed on a dataset without prior knowledge, and the results of the statistical causal analysis are output. To maximize the usage of the expert knowledge acquired in the pretraining process of the LLM, the method of generated knowledge prompting (Liu et al., 2022) is adopted, and then, the process of utilizing the LLM includes the second step (knowledge generation) and the third step (knowledge integration). In the second step, knowledge of the causal relationships between all pairs of variables is generated in detail from the domain knowledge of the LLM based on the zero-shot chain-of-thought (ZSCoT) prompting technique (Kojima et al., 2022). Here, the LLM can be prompted with the results of the SCD as supplemental information for LLM inference, expecting that the in-context learning (Brown et al., 2020) of the SCD results can lead to better performance of the LLM-KBCI in terms of statistics. We define this

---

[4]In this context, the term causal inference is used in a different sense from its conventional meaning in standard statistical frameworks, which typically aims to quantify the causal effect between a pair of variables. However, we intentionally refer to our process with LLMs as LLM-KBCI (LLM knowledge-based causal inference), as it can also be interpreted as an upstream task for constructing candidate causal models with SCD, to be used in subsequent causal effect analyses.

technique as "statistical causal prompting" (SCP). Thereafter, in the third step, the LLM judges whether causal relationships exist between all pairs of variables with "yes" or "no," thus objectively considering the dialogs of the second step. Here, the probabilities of the responses from the LLM are evaluated and transformed into the prior knowledge matrix. This matrix, the output of LLM-KBCI with SCP, is finally reapplied to SCD in the fourth step.

### 1.3 Our Contribution

The contributions of the method proposed in this study are as follows:

**(1) Realization of the Synthesis of LLM-KBCI and SCD in a Mutually Referential Manner**
We detail the practical method for realizing the proposed concept of Figure 1, and propose the SCP method. Furthermore, we demonstrate experiments with several benchmark datasets, which are open and have been widely used for the evaluation of SCD algorithms. They all consist of continuous variables.

**(2) Mutual Performance Enhancement of SCD and LLM-KBCI** We demonstrate that the augmentation by the LLM with SCP, improves the performance of several SCD algorithms in terms of domain expertise and statistics, and the performance of the LLM-KBCI is also enhanced by the SCP including the initial SCD results.

**(3) Improving the SCD results with the background knowledge provided by LLMs even if the dataset is uncontained in the pretraining materials** We prepare a closed health-screening dataset that is not included in the pretraining materials for LLMs, and demonstrate the experiment on a sub-dataset that has been randomly sampled from the entire dataset. Through this experiment, it is confirmed that the proposed method can lead to more statistically valid causal models with established domain knowledge. This fact proves that the LLMs can indeed contribute to background knowledge augmentation for SCD algorithms in practical situations, even if the dataset used for SCD is not memorized by LLMs.

**(4) Discussion on the Reliability and Risks of Integrating LLMs as a "black-box" into SCD**
In addition, we demonstrate some simulations of SCP in various conditions, confirm the existence of the patterns where SCP stably works as intended, and discuss the technical and statistical limitations of the proposed method. We also discuss realistic strategies and directions for future practical applications, to overcome the limitations of the proposed method, and to effectively obtain appropriate causal models from both expert and statistical points of view, avoiding critical errors.

## 2 Related Work and Originality in Our Work

**Augmentation of SCD Algorithms with Background Knowledge** As introduced in Section 1.1, several SCD algorithms [5] can be systematically augmented with background knowledge. Moreover, their software packages are open. For example, as a nonparametric[6] and the constraint-based SCD method, the Peter-Clark (PC) algorithm (Spirtes et al., 2000) is augmented with background knowledge of the forced or forbidden directed edges in "causal-learn." "Causal-learn" also provides the Exact Search algorithm (Silander & Myllymäki, 2006; Yuan & Malone, 2013) as a score-based SCD method, which can be augmented with the background knowledge of the forbidden directed edges as a superstructure matrix. With respect to a semiparametric approach, the DirectLiNGAM (Shimizu et al., 2011) algorithm is augmented with prior knowledge of the causal order (Inazumi et al., 2010) in the "LiNGAM" project (Ikeuchi et al., 2023).

---

[5]All of the algorithms adopted for the experiments in this paper can be used under the assumption of a DAG and with no hidden confounders.

[6]In this context, "nonparametric" indicates that neither the functional form characterizing the relationship between a cause-effect pair of variables nor the distribution of variables is assumed during the discovery process. In this sense, the PC algorithm can be classified as a nonparametric SCD method. Conversely, DirectLiNGAM is a semiparametric approach, as it estimates causal coefficients through regression analysis, although it does not assume any specific distribution of variables other than non-Gaussianity.

**Causal Inference in Knowledge-Driven Approach with LLMs**    In addition to the study of causality detection from natural language texts via language models with additional training datasets  (Khetan et al., 2022), the rapid growth of LLMs has made it possible to produce valuable works on causality, including causal inference using LLMs. Attempts have been made to use LLMs for causal inference among a set of variables, prompting with the metadata such as the names of variables, and without the SCD process with the benchmark datasets (Kiciman et al., 2024; Zečević et al., 2023; Jiralerspong et al., 2024; Zhang et al., 2024). Adopting a similar approach, the concept of causal modeling agents, which improves the precision of causal graphs through iteration of hypothesis generation from LLMs and model fitting to real-world data, has also been proposed  (Abdulaal et al., 2024). There are also studies on incorporating LLMs in the process of SCD as an alternative tool for conditional independence tests in the PC algorithm  (Cohrs et al., 2023), and for the identification of causal graphs beyond the Markov equivalent class  (Long et al., 2023). In addition, studies have focused on the use of LLMs to improve the SCD results(Ban et al., 2023; Vashishtha et al., 2023; Khatibi et al., 2024). However, all the experiments were conducted only on popular benchmark datasets, contained in the pretraining datasets of LLMs. Consequently, while acknowledging the valuable foundations laid by previous studies, it has remained uncertain whether the enhancements in SCD accuracy are truly driven by LLMs leveraging their vast knowledge for genuine causal inference, or merely by reproducing the memorized content of datasets (Vashishtha et al., 2023).

**Originality in Our Work**    In contrast to studies with similar focus on LLM-guided SCD (Ban et al., 2023; Vashishtha et al., 2023; Khatibi et al., 2024), this study also focuses on the construction of background knowledge in a quantitative manner based on the response probability of the LLM, which can reflect the credibility of the decision made by the LLM with SCP. In addition, in the case of a semiparametric SCD method such as LiNGAM, we detail herein how to achieve both the statistically well-fitting causal model and reasonable interpretation with respect to domain knowledge, by prompting with the causal coefficients and bootstrap probabilities of all the patterns of directed edges.

Moreover, we validate the proposed method on an unpublished dataset. For future practical applications, we also thoroughly discuss the limitations, risks of critical errors, expected improvement of techniques around LLMs, and the suggestion of realistic integration of domain expert checks of the results into this automatic methodology, with the simulations of SCP in both successful and failing scenarios. These approaches have not only demonstrated the practical utility and robustness of our method for future applications in each domain, but also opened new avenues for supporting the validity and applicability of existing works (Ban et al., 2023; Vashishtha et al., 2023; Khatibi et al., 2024).

## 3    Materials and Methods

In Section 3.1, we present an overview of the method in the form of an algorithm and explain the experimental conditions in detail. Furthermore, the prompting patterns used in the experiments are outlined in Section 3.1. Finally, the datasets used in the experiments are described in Section 3.3.

### 3.1    Algorithms and Elements for LLM-Augmented Causal Discovery

With respect to practicality, the method of Figure 1 is outlined as Algorithm 1. Following the notation in Algorithm 1, the input elements in the demonstration are explained below. First, the algorithms for SCD used in the first and fourth steps of Figure 1 are introduced. Then, for the second and third steps involving GPT-4, we explain the experimental conditions, the prompt templates, and the method for measuring GPT-4's confidence probabilities in causal relationships. Furthermore, to clarify how the output of the third step is applied to SCD in the fourth step, we describe the conversion from the probability matrix to the prior knowledge matrix in detail. Finally, with respect to practical application, we supplement our discussion with the theoretical computation time and scalability of the proposed method.

**Algorithms for Statistical Causal Discovery**    For the SCD method $S$, we adopted the PC algorithm (Spirtes et al., 2000) with Fisher's Z test(A., 1921), the Exact Search algorithm based on the A$^*$

---

**Algorithm 1** Background knowledge construction with the LLM prompted with the results of the SCD

---

**Input 1:** Data $X$ with variables$\{x_1, ..., x_n\}$
**Input 2:** SCD method $S$ (the one selected from PC, Exact Search, and DirectLiNGAM)
**Input 3:** Function for bootstrap $B$
**Input 4:** Response of the domain expert (LLM) $\epsilon_{\text{LLM}}$
**Input 5:** Log probability of the response $L(\epsilon_{\text{LLM}})$
**Input 6:** Prompt function for knowledge generation $Q_{ij}^{(1)}$
**Input 7:** Prompt function for knowledge integration $Q_{ij}^{(2)}$
**Input 8:** Transformation from the probability matrix to prior knowledge $T$
**Input 9:** Number of times to measure the probability $M$
**Output:** Result of SCD with prior knowledge $\hat{G}$ on $X$

SCD result without prior knowledge $\hat{G}_0 = S(X)$
bootstrap probability matrix $\boldsymbol{P} = B(S, X)$
**for** $i = 1$ **to** $n$ **do**
   **for** $j = 1$ **to** $n$ **do**
      $\overline{p_{i,i}} = \text{NaN}$
      **if** $i \neq j$ **then**
         prompt $q_{ij}^{(1)} = Q_{ij}^{(1)}(x_i, x_j, \hat{G}_0, \boldsymbol{P})$
         response $a_{ij} = \epsilon_{\text{LLM}}(q_{ij}^{(1)})$
         prompt $q_{ij}^{(2)} = Q_{ij}^{(2)}(q_{ij}^{(1)}, a_{ij})$
         **for** $m = 1$ **to** $M$ **do**
            $L_{ij}^{(m)} = L^{(m)}(\epsilon_{\text{LLM}}(q_{ij}^{(2)}) = \text{"yes"})$
         **end for**
         mean probability $\overline{p_{ij}} = \dfrac{\sum_{m=1}^{M} \exp(L_{ij}^{(m)})}{M}$
      **end if**
   **end for**
**end for**
probability matrix $\overline{\boldsymbol{p}} = (\overline{p_{ij}})$
prior knowledge $\boldsymbol{PK} = T(\overline{\boldsymbol{p}})$
**return** $\hat{G} = S(X, \boldsymbol{PK})$

---

algorithm[7] (Yuan & Malone, 2013), and the DirectLiNGAM algorithm (Shimizu et al., 2011), which can all be optionally augmented with prior knowledge, and are open in "causal-learn" (Zheng et al., 2023b) and "LiNGAM" (Ikeuchi et al., 2023).

Moreover, the form of $\hat{G}_0$, the SCD result without prior knowledge, is a DAG if $S$ is Exact Search or DirectLiNGAM, and is a completed partially directed acyclic graph (CPDAG) if $S$ is PC.

Furthermore, we also implement the bootstrap sampling function $B$ of the SCD algorithm to investigate the statistical properties such as the bootstrap probabilities $p_{ij}$ of the emergence of the directed edges from $x_j$ to $x_i$. In our experiments, the number of bootstrap resamplings was fixed to 1000.

---

[7]In particular, we utilized the A⋆ Exact Search algorithm from the "causal-learn" package (Zheng et al., 2023b), which employs the Bayes information criterion (BIC) (Schwarz, 1978) as its scoring metric.

**Conditions of the LLM and Prompting** To utilize the LLM as the domain expert, we adopted `GPT-4-1106-preview`[8] developed by OpenAI; the temperature, a hyperparameter for adjusting the probability distribution of the output, was fixed to 0.7.

The template for the first prompt $q_{ij}^{(1)}$ for knowledge generation is shown in Table 1. This prompting template[9] is based on the underlying principle of the ZSCoT technique[10] (Kojima et al., 2022), which was reported as a potential method to enhance the performance of the LLM generation tasks; enhancement is performed by guiding logical inference and eliciting the background knowledge acquired through the pretraining process from the LLM. Furthermore, expecting that the in-context learning (Brown et al., 2020) of the SCD results can lead to better performance of the LLM-KBCI in terms of statistics, the SCD results, e.g., the causal structure and bootstrap probabilities, can be included in ⟨blank 5⟩ and ⟨blank 9⟩, which are defined as "statistical causal prompting" (SCP). Because the information used in SCP is partially dependent on the SCD algorithms, a brief description of the patterns for constructing the contents of ⟨blank 5⟩ and ⟨blank 9⟩ is presented in Section 3.2.

As shown in Table 2, the generated knowledge is integrated into the second prompt, and GPT-4 is required to judge the existence of the causal effect from $x_j$ on $x_i$, considering the discussion of the second step regarding causal relationships from another perspective. Because the response from GPT-4 is required with "yes" or "no," it is simple to quantitatively evaluate the level of confidence of GPT-4 in the existence of a causal relationship based on both the SCD result and domain knowledge. The probability $p_{ij}$ of the assertion that there is a causal effect from $x_j$ on $x_i$ can be output from GPT-4 directly with the optional function of returning the log-probabilities of the most likely tokens at each token position[11]. Although $p_{ij}$ can be evaluated readily from the log probability of the GPT-4 response as "yes," there is a slight fluctuation in the log probability output from GPT-4 (Andriushchenko et al., 2024). Thus, we adopted the mean probability $\overline{p_{ij}}$ of the single-shot measurement $M$ times for the decision of prior knowledge matrix $\boldsymbol{PK}$, and we set $M = 5$ in the experiments. The combination of these prompt techniques can contribute to minimizing the risk of hallucination, and a reliable probability of the response of the LLM for the causal relationship between a pair of variables is expected to be obtained.

**Conversion from the Probability Matrix to the Prior Knowledge Matrix** For the subsequent SCD with prior knowledge, the probability matrix $\overline{\boldsymbol{p}}$ is transformed with $T$, as expressed by Algorithm 2, into the background knowledge matrix.

Here, if $PK_{ij} =$ Forbidden, the causal effect from $x_j$ to $x_i$ is forbidden from appearing in the SCD result. While only the existence of a directed edge from $x_j$ to $x_i$ is forbidden in the case of the PC and Exact Search algorithms, setting the $PK_{ij} =$ Forbidden as prior knowledge in DirectLiNGAM prohibits all direct paths from $x_j$ to $x_i$. However, if $PK_{ij} =$ Forced, the causal effect from $x_j$ to $x_i$ is forced to appear in the SCD result[12]. For the decision of a forbidden or forced causal relationship from $PK_{ij}$, we prepare the probability

---

[8]We recognize that various types of high-performance LLMs have been developed at several institutes, and that it is valuable to demonstrate that including a broader range of LLMs could provide valuable insights into the generalizability and scalability of our method across various LLM architectures. From this recognition, in addition to the main experiments with `GPT-4-1106-preview`, we have also demonstrated our approach with other LLMs, such as other GPT-series models (`GPT-3.5-turbo-1106`, `GPT-3.5-turbo-0125`, `GPT-4-0125-preview`, and `GPT-4o-2024-08-06`) and other open-source LLMs `Llama-3.1-70B-Instruct`, `gemma-2-27b-it`, and `Qwen2.5-7B-Instruct`, and have improved the performance of SCD, as described in Appendix H. However, our goal in this work is to explore the effectiveness of integrating LLMs into SCD via the SCP, which requires trials and comparisons of various SCP patterns and SCD algorithms, as described in Section 4. To maintain consistency and control across these trials, it is also important to fix the LLM in the experiments, which has advanced capabilities and state-of-the-art status in various domains. Moreover, we should adopt the LLMs that satisfy specific conditions for our experiments, such as the maximum input token capacity for the prompting processes, and the functionality to obtain the log-probability of the output. For these strategic and technical reasons, we adopted GPT-4 in our thorough experiments.

[9]The details of the adjustment of the wording in the prompt template are explained in Appendix I.

[10]Although the quality of the LLM outputs can be further enhanced, e.g., by fine-tuning with several datasets containing fundamental knowledge for the discussion of causality or retrieval-augmented generation (RAG), we adopt the idea of the ZSCoT to establish low-cost and simple methods, which can be universally applied independent of the targeted fields of causal inference.

[11]The technical detail of this probability sampling is explained in Appendix D.

[12]Although these constraints can be systematically adapted with the augmentation supported in "causal-learn" (Zheng et al., 2023b) and "LiNGAM" (Ikeuchi et al., 2023), the detailed meanings of forced and forbidden causal effects are slightly different among the SCD algorithms. The details are described in Appendix E

Table 1: Prompt template of $Q_{ij}^{(1)}(x_i, x_j, \hat{G}_0, \boldsymbol{P})$ used for the generation of expert knowledge of the causal effect from $x_j$ on $x_i$. The "blanks" enclosed with $\langle \ \rangle$ are filled with description words . The word for $\langle$blank 6$\rangle$ is selected from "a" or "no," depending on the content of $\langle$blank 5$\rangle$. Notations of the SCD result without prior knowledge $\hat{G}_0$ and the bootstrap probability matrix $\boldsymbol{P}$ are the same as those in Algorithm 1.

---

**Prompt Template of $q_{ij}^{(1)} = Q_{ij}^{(1)}(x_i, x_j, \hat{G}_0, \boldsymbol{P})$**

---

We want to carry out causal inference on $\langle$**blank 1. The theme**$\rangle$, considering $\langle$**blank 2. The description of all variables**$\rangle$ as variables.
First, we have conducted the statistical causal discovery with $\langle$**blank 3. The name of the SCD algorithm**$\rangle$, using a fully standardized dataset on $\langle$**blank 4. Description of the dataset**$\rangle$.

$\langle$**blank 5. This includes the information of $\hat{G}_0$ or $\boldsymbol{P}$.The details of the contents depend on the prompting patterns.**$\rangle$

According to the results shown above, it has been determined that there may be $\langle$**blank 6. a/no**$\rangle$ direct impact of a change in $\langle$**blank 7. The name of $x_j$**$\rangle$ on $\langle$**blank 8. The name of $x_i$**$\rangle$ $\langle$**blank 9. The value of causal coefficients or bootstrap probability**$\rangle$.
Then, your task is to interpret this result from a domain knowledge perspective and determine whether this statistically suggested hypothesis is plausible in the context of the domain.

Please provide an explanation that leverages your expert knowledge on the causal relationship between $\langle$**blank 7. The name of $x_j$**$\rangle$ and $\langle$**blank 8. The name of $x_i$**$\rangle$, and assess the naturalness of this causal discovery result.
Your response should consider the relevant factors and provide a reasoned explanation based on your understanding of the domain.

---

Table 2: Prompt template of $Q_{ij}^{(2)}(q_{ij}^{(1)}, a_{ij})$ used for the quantitative evaluation of the probability of GPT-4's assertion that there is a causal effect from $x_j$ on $x_i$.

---

**Prompt Template of $q_{ij}^{(2)} = Q_{ij}^{(2)}(q_{ij}^{(1)}, a_{ij})$**

---

An expert was asked the question below:
$\langle$**blank 10. $q_{ij}^{(1)}$**$\rangle$

Then, the expert replied with its domain knowledge:
$\langle$**blank 11. $a_{ij}$**$\rangle$

Considering objectively this discussion above, if $\langle$**blank 12. The name of $x_j$**$\rangle$ is modified, will it have a direct or indirect impact on $\langle$**blank 13. The name of $x_i$**$\rangle$?
Please answer this question with $\langle$yes$\rangle$ or $\langle$no$\rangle$.
No answers except these two responses are needed.

---

criterion for the forbidden path as $\alpha_1$ and that for the forced path as $\alpha_2$. In our experiments, $\alpha_1$ is fixed at 0.05, and $\alpha_2$ is fixed at 0.95 for common settings[13].

Furthermore, the differences in the constraints that can be adopted depending on the SCD algorithms should be considered [14]. In the Exact Search algorithm, the constraints of the forced edge cannot be applied. For the

---

[13]These heuristic thresholds follow the widely accepted conventions seen in statistical significance levels. Although the specific choice of threshold might influence the formation of the prior knowledge matrix, the probability outputs from LLMs for clear domain knowledge instances are either very high (close to 100 %) or very low (close to 0 %), ensuring that the essential insights are retained regardless of the threshold. In Appendix D, technical details of this sensitivity around these thresholds are described through the evaluation of the standard error of $\overline{p_{ij}}$, which can be realized by repeating single-shot measurements of $\overline{p_{ij}}$ $M$ times.

[14]For this reason stated here, it is certain that Algorithm 2, which considers several SCD algorithms, becomes somewhat complex due to the case-by-case approach depending on which SCD algorithm is adopted. Although it is ideal to construct a unified framework for the incorporation of $\boldsymbol{PK}$, this case-by-case approach in Algorithm 2 originates from the differences in how prior knowledge can be adopted in each SCD algorithm. Therefore, for simpler interpretation, from a practical point of view, it

---

**Algorithm 2** Transformation from the probability matrix into the prior knowledge matrix

---

**Input 1:** probability matrix $\boldsymbol{p} = (p_{ij})$
**Input 2:** SCD method $S \in \{$ PC, Exact Search, DirectLiNGAM $\}$
**Input 3:** probability criterion for the forbidden causal relationship $\alpha_1$
**Input 4:** probability criterion for the forced causal relationship $\alpha_2$
**Output:** prior knowledge matrix $\boldsymbol{PK} = (PK_{ij})$
**for** $i = 1$ **to** $n$ **do**
  **for** $j = 1$ **to** $n$ **do**
    $PK_{ij} = $ Unknown
    **if** $i = j$ **then**
      $PK_{ij} = $ Forbidden
    **else**
      **if** $p_{ij} < \alpha_1$ **then**
        $PK_{ij} = $ Forbidden
      **else if** $(p_{ij} \geq \alpha_2)$ **and** $(S \neq \text{Exact Search})$ **then**
        $PK_{ij} = $ Forced
      **end if**
    **end if**
  **end for**
**end for**
**if** $S = $ DirectLiNGAM **then**
  $\boldsymbol{PK} = A(\boldsymbol{PK})$ (acyclic transformation)
**end if**
**return** $\boldsymbol{PK}$

---

case of DirectLiNGAM, because prior knowledge is used for the decision of the causal order in the algorithm, the prior knowledge matrix must be an "acyclic" adjacency matrix when it is represented in the form of a network graph. Thus, when $S = $ DirectLiNGAM and $\boldsymbol{PK}$ is cyclic, an additional transformation algorithm $A$ is required. In addition, several acyclic transformation patterns can exist; only one acyclic matrix with some criteria should be selected. The algorithm for the transformation and the matrix selection criterion in this study are explained in Appendix F.

**Theoretical Computation Time and Scalability** It is also valuable to discuss the computation time and scalability with respect to the number of variables, as our proposed method includes steps involving the LLM in addition to the calculations performed by the SCD algorithm. The total time required for machine processing $T(\text{SCD}, \text{LLM}, n, N, \boldsymbol{PK})$ is expressed as follows:

$$
\begin{aligned}
T(\text{SCD}, \text{LLM}, n, N, M, \boldsymbol{PK}) &= T_{\text{Step.1}}(SCD, n, N) + T_{\text{Step.2}}(LLM, n) \\
&\quad + T_{\text{Step.3}}(LLM, n, M) + T_{\text{Step.4}}(SCD, n, N, \boldsymbol{PK}) \\
&\quad + T_{\text{Acyclic}}(\text{SCD}, \boldsymbol{PK})
\end{aligned}
\tag{1}
$$

The variable SCD can be one of the following algorithms: PC, ExactSearch, or DirectLiNGAM. The variable LLM refers to the model adopted in this process. Furthermore, $n$ denotes the number of variables, and $N$ represents the number of data points in the dataset. The computation time of the transformation algorithm $A$, described in Appendix F, is denoted as $T_{\text{Acyclic}}(\text{SCD}, \boldsymbol{PK})$. When SCD is PC or ExactSearch, this value is zero because no transformation process for adoptable $\boldsymbol{PK}$ is required.

The SCD computation time, $T_{\text{Step.1}}(\text{SCD}, n, N)$, has the same scalability as that of the SCD computation. Similarly, $T_{\text{Step.4}}(\text{SCD}, n, N, \boldsymbol{PK})$ represents the computation time of the SCD algorithm under the constraints of $\boldsymbol{PK}$. Because the constraints imposed by $\boldsymbol{PK}$ reduce the search space of the SCD algorithm, it is expected that $T_{\text{Step.4}}(\text{SCD}, n, N, \boldsymbol{PK}) \leq T_{\text{Step.1}}(\text{SCD}, n, N)$. Furthermore, the stronger constraints $\boldsymbol{PK}$ imposes, the smaller $T_{\text{Step.4}}(\text{SCD}, n, N, \boldsymbol{PK})$ becomes.

---

is recommended that the SCD algorithm adopted in this method be determined immediately after the dataset for analysis is determined.

However, the LLM computation times $T_{\text{Step.2}}(\text{LLM}, n)$ and $T_{\text{Step.3}}(\text{LLM}, n, M)$ have different scalability compared to the SCD computation times $T_{\text{Step.1}}(\text{SCD}, n, N)$ and $T_{\text{Step.4}}(\text{SCD}, n, N, \boldsymbol{PK})$. If we represent the time for a single cycle of reading the prompt $q$ and generation by an LLM as $t(\text{LLM}, q)$, because all the LLM's tasks in Steps 2 and 3 can be carried out independently, $T_{\text{Step.2}}(\text{LLM}, n)$ and $T_{\text{Step.3}}(\text{LLM}, n, M)$ can be expressed as follows:

$$T_{\text{Step.2}}(LLM, n) = \sum_{i \neq j} t(\text{LLM}, q_{ij}^{(1)}) \tag{2}$$

$$T_{\text{Step.3}}(LLM, n, M) = M \sum_{i \neq j} t(\text{LLM}, q_{ij}^{(2)}) \tag{3}$$

Furthermore, if the computation times $t(\text{LLM}, q_{ij}^{(1)})$ and $t(\text{LLM}, q_{ij}^{(2)})$ can be considered approximately independent of the pair of variables, we have $t(\text{LLM}, q_{ij}^{(1)}) \approx \overline{t_1}(\text{LLM})$ and $t(\text{LLM}, q_{ij}^{(2)}) \approx \overline{t_2}(\text{LLM})$. Therefore, Eq. (2)-(3) are approximately expressed as follows:

$$T_{\text{Step.2}}(LLM, n) \simeq N(N-1)\overline{t_1}(\text{LLM}) = O(N^2) \tag{4}$$

$$T_{\text{Step.3}}(LLM, n, M) \simeq MN(N-1)\overline{t_2}(\text{LLM}) = O(MN^2) \tag{5}$$

Consequently, when the computational costs in the steps involving the LLM are estimated, the scalability described in Eq. (4)-(5) should be considered.

Finally, when SCD = DirectLiNGAM and $\boldsymbol{PK}$ has a cyclic structure, $T_{\text{Acyclic}}(\text{SCD}, \boldsymbol{PK})$ becomes a non-zero value. Although we do not further discuss the cost of this process, as discussed in Appendix F, with an increasing number of cycles in $\boldsymbol{PK}$, $T_{\text{Acyclic}}(\text{SCD}, \boldsymbol{PK})$ may increase rapidly because of the complexity involved in removing cycles.

## 3.2 Experiment Patterns of SCP

Related to ⟨blank 5⟩ and ⟨blank 9⟩ in Table 1, we conducted experiments using several patterns of SCP. The notations of the prompting patterns in the experiments are presented with explanations below:

**Pattern 0: without SCP**    This pattern corresponds to the reference for the comparison with the other patterns, including SCD results in their prompts. Because the prompt template shown in Table 1 is not adequate for this pattern, we prepare a different template for Pattern 0, which is shown in Appendix B.

**Pattern 1: With the list of edges that appeared in the first SCD**    Directed or undirected[15] edges between $x_i$ and $x_j$ that emerged in the SCD are listed.

**Pattern 2: With the list of edges with their non-zero bootstrap probabilities**    Directed or undirected edges between $x_i$ and $x_j$ that emerged in the bootstrap process are listed with their bootstrap probabilities.

**Pattern 3: With the list of edges that emerged in the first SCD with the calculated causal coefficients (only for DirectLiNGAM)**    On the basis of the property of DirectLiNGAM, which outputs the causal coefficients with the DAG discovered in the algorithm, this pattern is attempted to elucidate whether more information, such as causal coefficients, in addition to Pattern 1, can improve the performance of LLM-KBCI and the subsequent SCD with prior knowledge.

**Pattern 4: With the list of edges with their non-zero bootstrap probabilities and calculated causal coefficients for the full dataset (only for DirectLiNGAM)**    We also attempt this pattern with the most information of the 1st SCD as a mixture of Patterns 2 and 3.

---

[15]In the PC algorithm, since it is possible that not all edge directions are determined by the orientation rules, the default output is a CPDAG, and undirected edges that appear as $x_i - x_j$ with respect to causal relationships between "$x_i$ and $x_j$" in which the direction cannot be determined, can be detected. The prompt template for reflecting this difference from directed edges is shown in Appendix B.

### 3.3 Datasets for the Experiments

Although several widely open benchmark datasets with well-known ground truths exist, particularly for Bayesian network-based causal structure learning (Scutari & Denis, 2014), several of them are fully or partially composed of categorical or discrete variables. However, considering Patterns 3 and 4 for the experiments in this study, because the basic structure causal model assumes the continuous properties of all the variables, it is more effective to adopt benchmark datasets fully composed of continuous variables.

Consequently, we select three benchmark datasets for the experiments, as follows: 1. Auto MPG data (Quinlan, 1993) (five continuous variables) , 2. Deutscher Wetterdienst (DWD) climate data (Mooij et al., 2016) (six continuous variables) , 3. Sachs protein data (Sachs et al., 2005) (eleven continuous variables) .

Furthermore, to demonstrate that GPT-4 can aid SCD with its domain knowledge, even if the dataset used in the SCD process and analytics on the dataset are not contained in the pretraining data of GPT-4, the proposed methods are also applied to our dataset of health-screening results, which has not been disclosed and therefore not learned by GPT-4. To demonstrate that the proposed methods can be applied when the dataset could contain biases, which may lead to highly inaccurate SCD results, the health-screening dataset for this experiment was sampled, and we deliberately chose a subset where certain biases[16] could be still present. Basic information on these datasets, such as the first SCD results and the ground truths is presented in Appendix C.

## 4 Results and Discussion

In this section, we first present the experimental results on benchmark datasets with known ground truths, which are evaluated via several metrics, as described in Section 4.1. In addition, Section 4.2 reports the experimental results on our unpublished health-screening dataset, demonstrating that the assistance of GPT-4 with SCP can help the output of SCD align more closely with the ground truths. Furthermore, we discuss the reliability of our method in Section 4.3, including simulations to identify both successful and failure cases. Finally, at the end of Section 4.3, we also discuss potential improvements in the prompting methodology for stabilizing GPT-4's confidence probability measurements, as well as realistic strategies for overcoming these issues in practical applications of our proposed method.

### 4.1 Results in Benchmark Datasets

For the interpretation of the experimental results, we evaluate $\boldsymbol{PK}$ (for measuring the performance of the LLM-KBCI with the prompts, including the first SCD results) and the adjacency matrix obtained in SCD with $\boldsymbol{PK}$ for each pattern (for measuring the performance of the SCD augmented with the LLM-KBCI), with the structural hamming distance (SHD), false positive rate (FPR), false negative rate (FNR), precision, and the F1 score, using the ground truth adjacency matrix $\boldsymbol{GT}$ as a reference.

All of these metrics are frequently used to evaluate how the results of the causal structure learning are close to the ground truths, and can be calculated solely from the adjacency matrix and $\boldsymbol{GT}$, as shown in Appendix E.3. In addition, this paper presents an evaluation of the comparative fit index (CFI) (Bentler, 1990), root mean square error of approximation (RMSEA) (Steiger & Lind, 1980) and BIC (Schwarz, 1978) of the causal structure obtained in SCD with $\boldsymbol{PK}$, under the assumption of linear-Gaussian data[17]; to evaluate the results with respect to the statistical validity of the calculated causal models. To evaluate the effect

---

[16]Although datasets can generally exhibit various types of bias—-such as selection bias, sampling bias, label bias, confirmation bias, and measurement bias-—it is often challenging to identify the specific biases present. As explained in Section 4.2, we selected a subsample that is much smaller than the original one, with which DirectLiNGAM produces the unreasonable edges from the perspective of domain experts, although that subsample dataset is randomly selected from the original dataset. In this context, as a result, there could be biases associated with a particular attribute, although there is no arbitral process in composing this subsample. However, the proposed method remains effective regardless of the types of biases present in the dataset. Therefore, we do not specify the exact type of bias here.

[17]Although this assumption of linear-Gaussian data for the calculation of the CFI, RMSEA, and BIC, does not match the assumption of a non-Gaussian error distribution in LiNGAM, we adopt these indices uniformly to evaluate and compare the results with respect to the same statistical method based on the typical analysis using the structural equation model with the linear-Gaussian assumption, irrespective of the difference in the SCD algorithms.

of **PK** augmentation by the LLM, the baseline result is that without **PK** (Baseline A), as a reference for comparison with the SCD results augmented with **PK**. In contrast, to evaluate the effect of the SCP, the results with the prompt in Pattern 0 become the baseline (Baseline B), for comparison with the results obtained in other SCP patterns. The indices for all patterns on the DWD, Auto MPG, and Sachs datasets are summarized in Table 3.

**Enhancing the performance with prior knowledge augmentation by GPT-4**   One of the characteristics in Table 3 is that, in most cases, the result of SCD augmented with **PK** is more similar to **GT** than the first SCD result without prior knowledge (Baseline A). This behavior is interpreted as the knowledge-based improvement of the causal graph by GPT-4 as a domain expert, which is qualitatively consistent with other related researches on LLM-guided causal inference (Ban et al., 2023; Vashishtha et al., 2023). Moreover, in many of the cases of Auto MPG and DWD data, the precision or F1 score are higher after the SCD augmented with **PK** than the pure **PK**, which are conclusions of LLM-KBCI, while they are almost comparable in the cases of Sachs data. This comparison implies that even if the LLM-KBCI is not optimal, the ground truths can be better approached by conducting SCD augmented with the LLM-KBCI. In addition, the BIC decreases in almost all the patterns from the SCD result without **PK** (Baseline A). The aforementioned properties suggest that knowledge-based augmentation from GPT-4 can improve the performance of SCD, indeed, in terms of consistency with respect to domain expert knowledge and the statistical causal structure. However, the amount of improvement can differ depending on the number of variables and the methods of SCD.

**Dependence on the number of variables**   In the case of Auto MPG data with only five variables, the amount of improvement for each SCD method is almost constant among all the prompting patterns. One of the possible reasons is that, within relatively small numbers of variables, the amount of information in the SCP becomes small, and the difference in the inference performance of GPT-4 among the prompt patterns becomes subtle. Moreover, because the space for discovery also decreases as the network shrinks, the SCD algorithm may reach a single optimal solution, even if **PK** is different.

On the other hand, in the cases of DWD data with six variables and Sachs data with eleven variables, the difference in the amount of improvement becomes clearer depending on the prompting patterns. This implies that the threshold of the number of variables over which the quality of the **PK** and SCD results depend on the amount of information included. In the SCP for GPT-4, it is approximately five.

Moreover, in many cases of DWD and Sachs data, particularly for Exact Search or DirectLiNGAM, the precision and F1 score of **PK** in Pattern 0 (Baseline B) are usually smaller than those of any of the other patterns, in which GPT-4 experiences SCP. This finding supports the performance enhancement of LLM-KBCI by the SCP.

**Prompting pattern dependence on SCD methods**   On the other hand, considering the output of the SCD augmented with **PK**, Pattern 0 (Baseline B) stably indicates relatively higher performance among all the patterns in terms of both domain knowledge and statistical model fitting. Furthermore, the results of Patterns 0 and 1 are almost the same when the PC or the Exact Search algorithm is adopted; in particular, the result of Pattern 0 with the PC algorithm on Sachs data is superior to that of Pattern 0.

Although it is difficult to explain this behavior, one of the possible reasons, if we focus only on the results on Sachs data, is that we adopted the ground truths partially determined by Bayesian network inference (Sachs et al., 2005). Indeed, the first SCD result in PC is already relatively close to **GT**, as shown in Figure 12 (a), and we interpret that, in this situation, the performance of SCD with **PK** cannot be improved.

The scenario differs when DirectLiNGAM is adopted. The performance of the SCD with **PK** either in Pattern 1 or 2, remains overall superior to that of Pattern 0 (Baseline B) from both statistical and domain expert points of view. This implies that the SCP can effectively improve the performance of DirectLiNGAM.

However, from the analysis of Patterns 3 and 4, in which GPT-4 is prompted with the causal coefficients of the first SCD results, it is also revealed that prompting with a greater amount of statistical information does not always lead to improved SCD results. In particular, although **PK** in Pattern 4 is closer to the ground

Table 3: Comparison of the SCD results in all the experimental patterns. While the SHD, FPR, FNR, precision, and the F1 score are compared to evaluate how close the results are to the ground truths, CFI, RMSEA, and BIC are compared to evaluate the statistical validity of the calculated causal models. Lower values are superior for the SHD, FPR, FNR, RMSEA, and BIC, and higher values for the precision, F1 score and CFI. The numbers in parentheses indicate the SHD, FPR, FNR, precision, and the F1 score of **PK** with ground truths, to evaluate the performance of LLM-KBCI. The values highlighted in bold are the optimal results among Patterns 0–4, if the dataset and the SCD algorithm are fixed. Baseline A is used for comparison with the SCD results with *PK* generated by the LLM, to evaluate the effect of **PK** augmentation by the LLM. Baseline B is used for comparison with the results obtained in other SCP patterns, to evaluate the effect of the SCP.

**1. Auto MPG data with 5 continuous variables**

| SCD algorithm | Pattern | SHD↓ | FPR↓ | FNR↓ | Precision↑ | F1score↑ | CFI↑ | RMSEA↓ | BIC↓ |
|---|---|---|---|---|---|---|---|---|---|
| PC | wo **PK**(Baseline A) | 8 | 0.40 | 0.80 | 0.11 | 0.14 | 1.00 | 0.00 | 71.65 |
| | Pattern 0 (Baseline B) | **3 (5)** | 0.15 (**0.25**) | 0.20 (0.20) | 0.57 (**0.44**) | 0.67 (**0.57**) | 1.00 | 0.07 | 65.62 |
| | Pattern 1 | 4 (7) | 0.15 (0.30) | 0.20 (0.20) | 0.57 (0.40) | 0.67 (0.53) | 1.00 | **0.00** | **59.71** |
| | Pattern 2 | **3 (6)** | 0.15 (0.30) | 0.20 (0.20) | 0.57 (0.40) | 0.67 (0.53) | 1.00 | 0.07 | 65.62 |
| Exact Search | wo **PK**(Baseline A) | 5 | 0.25 | 0.40 | 0.38 | 0.46 | 1.00 | 0.07 | 71.61 |
| | Pattern 0 (Baseline B) | 4 (5) | 0.20 (0.25) | 0.20 (0.20) | 0.50 (0.44) | 0.62 (0.57) | 1.00 | 0.09 | 71.59 |
| | Pattern 1 | 5 (5) | 0.20 (**0.20**) | 0.20 (0.20) | 0.50 (**0.50**) | 0.62 (**0.62**) | 1.00 | **0.07** | 65.62 |
| | Pattern 2 | 4 (5) | 0.20 (0.25) | 0.20 (0.20) | 0.50 (0.44) | 0.62 (0.57) | 1.00 | 0.09 | **71.59** |
| DirectLiNGAM | wo **PK**(Baseline A) | 8 | 0.40 | 0.80 | 0.11 | 0.14 | 1.00 | 0.05 | 77.61 |
| | Pattern 0 (Baseline B) | **3 (5)** | **0.15 (0.25)** | 0.20 (0.20) | **0.57 (0.44)** | **0.67 (0.57)** | 1.00 | 0.07 | **65.62** |
| | Pattern 1 | **3 (5)** | **0.15 (0.30)** | 0.20 (0.20) | **0.57 (0.40)** | **0.67 (0.53)** | 1.00 | 0.07 | **65.62** |
| | Pattern 2 | **3 (5)** | **0.15 (0.25)** | 0.20 (0.20) | **0.57 (0.44)** | **0.67 (0.57)** | 1.00 | 0.07 | **65.62** |
| | Pattern 3 | 4 (6) | 0.20 (0.35) | 0.20 (0.20) | 0.50 (0.36) | 0.62 (0.50) | 1.00 | **0.00** | 71.65 |
| | Pattern 4 | **3 (5)** | **0.15 (0.30)** | 0.20 (0.20) | **0.57 (0.40)** | **0.67 (0.53)** | 1.00 | 0.07 | **65.62** |

Although there are no significant differences in the metrics between the prompting patterns, it is clearly observed that the SCD results augmented with **PK** are more similar to **GT** than the initial SCD results without prior knowledge (Baseline A).

**2. DWD climate data with 6 continuous variables**

| SCD algorithm | Pattern | SHD↓ | FPR↓ | FNR↓ | Precision↑ | F1score↑ | CFI↑ | RMSEA↓ | BIC↓ |
|---|---|---|---|---|---|---|---|---|---|
| PC | wo **PK**(Baseline A) | 9 | 0.20 | 0.83 | 0.14 | 0.15 | 0.90 | 0.22 | 69.32 |
| | Pattern 0 (Baseline B) | 5 (**8**) | 0.03 (**0.20**) | 0.67 (0.33) | 0.67 (**0.40**) | 0.44 (**0.50**) | 0.71 | 0.36 | 32.70 |
| | Pattern 1 | 5 (9) | 0.03 (0.23) | 0.67 (0.33) | 0.67 (0.36) | 0.44 (0.47) | 0.71 | 0.36 | 32.70 |
| | Pattern 2 | 5 (**8**) | 0.03 (**0.20**) | 0.67 (0.33) | 0.67 (**0.40**) | 0.44 (**0.50**) | 0.71 | 0.36 | 32.70 |
| Exact Search | wo **PK**(Baseline A) | 6 | 0.20 | 0.17 | 0.45 | 0.59 | 0.91 | 0.28 | 92.87 |
| | Pattern 0 (Baseline B) | 5 (8) | **0.10 (0.20)** | 0.33 (0.33) | **0.57 (0.40)** | **0.62 (0.50)** | **0.98** | **0.12** | 58.38 |
| | Pattern 1 | **5 (5)** | **0.10 (0.13)** | 0.33 (**0.17**) | **0.57 (0.56)** | **0.62 (0.67)** | 0.91 | 0.19 | **57.73** |
| | Pattern 2 | 6 (9) | 0.13 (0.23) | 0.33 (0.33) | 0.50 (0.36) | 0.57 (0.47) | 0.91 | 0.20 | 63.58 |
| DirectLiNGAM | wo **PK**(Baseline A) | 10 | 0.33 | 0.67 | 0.17 | 0.22 | 1.00 | 0.00 | 99.53 |
| | Pattern 0 (Baseline B) | 4 (8) | 0.07 (0.20) | **0.33 (0.33)** | 0.67 (0.40) | **0.67 (0.50)** | **1.00** | **0.00** | 52.67 |
| | Pattern 1 | 8 (8) | 0.10 (**0.10**) | 0.83 (0.83) | 0.25 (0.25) | 0.20 (0.20) | 0.64 | 0.43 | **38.03** |
| | Pattern 2 | 4 (7) | **0.03 (0.17)** | 0.50 (0.33) | **0.75 (0.44)** | 0.60 (0.53) | 0.98 | 0.09 | 40.80 |
| | Pattern 3 | 5 (**6**) | 0.10 (0.13) | **0.33 (0.33)** | 0.57 (**0.50**) | 0.62 (0.57) | 0.93 | 0.16 | 57.90 |
| | Pattern 4 | 5 (**6**) | 0.10 (0.17) | **0.33 (0.16)** | 0.57 (**0.50**) | 0.62 (**0.62**) | 0.92 | 0.18 | 57.80 |

It is confirmed that the outputs of LLM-KBCI in many of Patterns 1–4 are likely to approach the ground truths more closely than those in Pattern 0. The performance of Exact Search in Pattern 1 (with the SCP) is the similar in quality to that in Pattern 0. The output of DirectLiNGAM in Pattern 2 (with SCP) can be a superior causal model than that in Pattern 0.

**3. Sachs' protein data with 11 continuous variables**

| SCD algorithm | Pattern | SHD↓ | FPR↓ | FNR↓ | Precision↑ | F1score↑ | CFI↑ | RMSEA↓ | BIC↓ |
|---|---|---|---|---|---|---|---|---|---|
| PC | wo **PK**(Baseline A) | 24 | 0.16 | 0.47 | 0.38 | 0.44 | 0.99 | 0.05 | 294.15 |
| | Pattern 0 (Baseline B) | **15 (19)** | **0.04 (0.11)** | **0.58 (0.47)** | **0.67 (0.48)** | **0.52 (0.50)** | 0.89 | 0.16 | **166.91** |
| | Pattern 1 | 25 (43) | 0.17 (0.53) | 0.68 (**0.16**) | 0.26 (0.23) | 0.29 (0.36) | 0.97 | 0.11 | 284.58 |
| | Pattern 2 | 23 (23) | 0.13 (0.24) | 0.74 (0.32) | 0.28 (0.35) | 0.27 (0.46) | 0.97 | **0.09** | 231.27 |
| Exact Search | wo **PK**(Baseline A) | 31 | 0.26 | 0.68 | 0.18 | 0.23 | 0.99 | 0.07 | 374.35 |
| | Pattern 0 (Baseline B) | 17 (19) | 0.07 (0.11) | 0.58 (0.47) | 0.53 (0.48) | **0.47 (0.50)** | 0.91 | 0.16 | 202.97 |
| | Pattern 1 | 17 (15) | **0.05 (0.06)** | 0.68 (0.58) | **0.55 (0.57)** | 0.40 (0.48) | 0.87 | 0.20 | **158.26** |
| | Pattern 2 | **16 (14)** | 0.11 (0.14) | **0.53 (0.32)** | 0.45 (0.48) | 0.46 (**0.57**) | **0.95** | **0.12** | 257.53 |
| DirectLiNGAM | wo **PK**(Baseline A) | 29 | 0.25 | 0.47 | 0.28 | 0.36 | 1.00 | 0.01 | 410.23 |
| | Pattern 0 (Baseline B) | 17 (19) | **0.07 (0.11)** | 0.53 (0.47) | 0.56 (0.48) | 0.51 (0.50) | 0.91 | **0.16** | **203.02** |
| | Pattern 1 | **15 (13)** | **0.07 (0.08)** | **0.47 (0.37)** | **0.59 (0.60)** | **0.56 (0.62)** | 0.88 | 0.18 | 220.32 |
| | Pattern 2 | 22 (21) | 0.14 (0.18) | 0.63 (0.47) | 0.33 (0.36) | 0.35 (0.43) | 0.70 | 0.31 | 269.76 |
| | Pattern 3 | 22 (23) | 0.15 (0.18) | 0.53 (0.53) | 0.38 (0.33) | 0.42 (0.39) | **0.92** | **0.16** | 301.51 |
| | Pattern 4 | 20 (16) | 0.12 (0.16) | 0.53 (**0.26**) | 0.43 (0.47) | 0.45 (**0.57**) | 0.89 | 0.19 | 264.97 |

Although the prompting patterns that lead to the best performance differ, a similar tendency to that observed in the DWD climate data is observed.

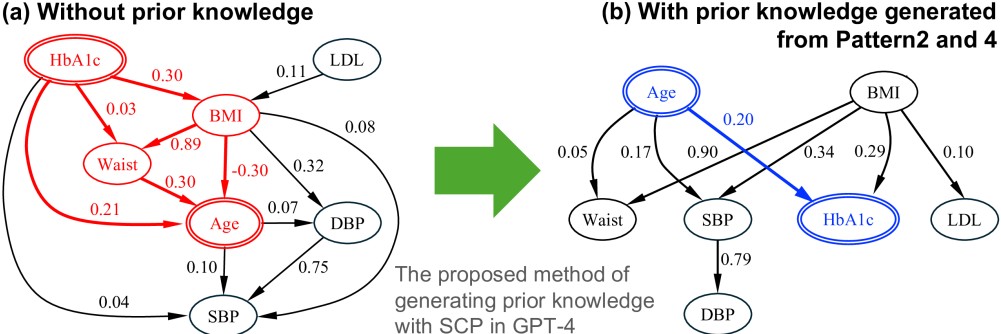

Figure 2: Results of DirectLiNGAM on the health-screening data. (a) Result without prior knowledge. (b) Result with prior knowledge, which is generated from GPT-4 with the SCP in Patterns 2 and 4. In this randomly selected subset, the DirectLiNGAM result without prior knowledge exhibits unnatural paths drawn in red in (a), which indicates that "Age" is influenced by "HbA1c." However, when the proposed method is applied, the unnatural behavior is clearly mitigated with the guidance of prior knowledge generated from GPT-4 with SCP, including the value of causal coefficients in (a) or the bootstrap probabilities of the emergence of directed edges.

truth matrix than that in Pattern 0 in the DWD or Sachs data [18], the final SCD result augmented with **PK** in Pattern 4 is inferior to that of Pattern 0. One of the possible reasons for this may be the pruning of the candidate edges suggested from **PK** in the SCD process. To elucidate this behavior, further research is required on what type and amount of information in SCP can truly maximize the performance of SCD.

## 4.2 Results for a Selected Subsample of Health-Screening Data Excluded from the GPT-4 Pretraining Dataset

It is difficult to assert that this improvement is solely due to the LLM's high performance in KBCI from the experimental results on open benchmark datasets, because it cannot be determined whether the improvement stems from the LLM's recall of the data obtained during the pretraining process on these datasets. Furthermore, assuming realistic situations, it is also important to confirm the robust effectiveness of this method, even if the range of the available dataset for statistical causal inference is limited to observation data, which may be statistically biased, and trivial causal relationships are not apparent in SCD without prior knowledge. Therefore, we also apply the proposed methods to the sub-dataset of health-screening results, which has been sampled from the entire dataset[19], and the natural ground truth is not presented in the SCD results without **PK**. The original dataset consists of personal health-screening data from Japanese health insurance records. Owing to legal restrictions in Japan, the raw data are not publicly available. Anonymized data are used in a restricted environment, are not directly exposed to the Internet for analysis, and are not shared with third parties. On the basis of these properties, it is reasonable to conclude that our unpublished health-screening dataset was not included in the pretraining data of GPT-4, although it is impossible to strictly prove the absence of any overlap between our dataset and the pretraining data, as noted in Appendix C.4.

In Figure 2(a), the result of DirectLiNGAM without **PK** is shown, and unnatural directed edges to "Age" from other variables are suggested, although the parts of the ground truths from expert knowledge are reversed relationships from these edges, such as "Age"→"HbA1c". However, when the causal discovery is assisted by **PK** generated from GPT-4 with the SCP in Patterns 2 and 4, the causal graph becomes more natural: "Age" is not influenced by other variables, and the ground truth "Age"→"HbA1c", which cannot be detected without **PK** in this randomly selected subset, appears in the causal graph, as shown in Figure 2(b). Because this sub-dataset and the analysis results are not disclosed and have been completely excluded from the pretraining data for GPT-4, GPT-4 cannot respond to prompts asking for causal relationships merely by reproducing the knowledge acquired from the same data. On the basis of the above, it is verified that the assistance of GPT-4 with SCP can cause the result of SCD to approach the ground truths to some extent,

---

[18]This fact reinforces the reliability of our interpretation that the SCP enhances the performance of LLM-KBCI.

[19]The details of the sampling method for this experiment are presented in Appendix C

Table 4: Quantitative evaluation of the characteristic results of SCD in the proposed methods on the subset of health-screening data, which could contain certain biases.

A: Elements of $\boldsymbol{P}$ generated in each prompting pattern for all the appropriate ground truth causal relationships in these variables. The values enclosed in parentheses are the bootstrap probabilities of the directed edges in the DirectLiNGAM algorithm without $\boldsymbol{PK}$. It is suggested that the mean probabilities of the positive response on the causal relationships from GPT-4 are influenced by the results of SCD and bootstrap probabilities when they are included in SCP. Moreover, all the probabilities of reversed, unnatural causal relationships in which "age" is influenced by other factors are extremely close to zero.

B: CFI, RMSEA, and BIC were evaluated on the basis of model fitting with the structure discovered by DirectLiNGAM, under the assumption of linear-Gaussian data. The values enclosed in parentheses are the statistics of the structure calculated without $\boldsymbol{PK}$.

It can be inferred that SCP with bootstrap probabilities in Patterns 2 and 4, enhances the confidence in "Age"→"DBP" by GPT-4 and improves the BIC compared with Pattern 0.

| | | Pattern 0 | Pattern 1 | Pattern 2 | Pattern 3 | Pattern 4 |
|---|---|---|---|---|---|---|
| **A. Probability of reproducing ground truth from GPT-4** | **"Age"→"BMI"** (0.166) | 0.901 | 0.076 | 0.093 | 0.306 | 0.037 |
| | **"Age"→"SBP"** (0.550) | 0.626 | 0.302 | 0.207 | 0.235 | 0.795 |
| | **"Age"→"DBP"** (0.308) | 0.001 | 0.019 | 0.115 | 0.095 | 0.926 |
| | **"Age"→"HbA1c"** (0.327) | 0.986 | 0.170 | 0.723 | 0.046 | 0.176 |
| **B. Statistics of linear-Gaussian fitting with the results of DirectLiNGAM** | **CFI↑** (1.002) | 0.999 | 0.992 | 0.995 | 0.986 | 0.995 |
| | **RMSEA↓** (0.000) | 0.018 | 0.054 | 0.032 | 0.057 | 0.032 |
| | **BIC↓** (124.332) | 103.581 | 89.738 | 89.740 | 103.506 | 89.740 |

even when the dataset is not learned by GPT-4 and could contain biases. This finding suggests that our proposed method can work well with the domain knowledge acquired by the LLM during training, even if the exact same data are not memorized by the LLM. Moreover, the effectiveness of the proposed method demonstrated on both open benchmark datasets and unpublished health-screening datasets, also supports the validity and applicability of existing works (Ban et al., 2023; Vashishtha et al., 2023; Khatibi et al., 2024), where knowledge-based improvements with LLMs of the causal graph on open benchmark datasets have been demonstrated via their own methods with unique originality. Our research, by showing versatility across both open and unpublished data, further enhances the validity of these existing works.

For further clarity regarding the behaviors of SCP, the mean probabilities of the positive response on the causal relationships from GPT-4 for ground truths in the health-screening data, in addition to the statistics of the fitting results with the structure suggested by DirectLiNGAM with the SCP, are presented in Table 4.

Finally, it is also confirmed that the BIC decreases with the assistance of GPT-4 compared with that without $\boldsymbol{PK}$. In particular, the values in Patterns 1, 2, and 4 are lower than those in Pattern 0. This suggests that SCP can contribute to the discovery of causal structures with more adequate statistical models.

### 4.3 Discussion of Reliability Evaluations from the Domain Knowledge and Prevention of Misuse Toward Practical Applications

However, from a realistic point of view, it is also impossible to control the LLMs as perfectly as possible, for adequate output of the LLM-KBCI tasks. Therefore, for the practical application of the proposed method, it is important not only to confirm the effective points of view shown above, but also, to analyze the case of failed outputs of the LLM-KBCI. In fact, as shown in Table 4, while the existence of "Age"→"HbA1c" is supported in the probabilities over 0.95 in Pattern 0 without SCP, the probability decreases for other patterns with SCP. This behavior implies that while GPT-4 itself, without any prompting, accepts "Age"→"HbA1c" as one of the ground truths, the SCP, including the result that does not statistically support this edge, reduces the confidence of GPT-4 in this causal relationship.

The primary cause of this behavior is our reliance solely on the knowledge possessed by GPT-4, and the inherent performance limitations of GPT-4 itself may influence these results. Generally, it is challenging to comprehensively confirm what LLMs know or do not know. Although benchmark tests or equivalent performance measurements, as described in the development reports of each LLM, can confirm their knowledge to some extent, it is practically difficult to exhaustively grasp all of their knowledge (Kaddour et al., 2023; Pezeshkpour, 2023).

Moreover, although we have adopted several prompt techniques to reduce hallucinations as much as possible with our method, we cannot predict when and how hallucinations might occur during practical applications. Therefore, it is necessary to recognize this point and exercise caution during deployment. Specifically, examining the impact of prompting patterns reveals that, while incorporating SCD results through the SCP tends toimprove outcomes overall compared with not including them, at a granular level such as each causal relationship between a pair of variables, SCP might inadvertently guide LLMs' judgment in the wrong direction.

Recognizing these issues, mechanically applying our method without considering potential countermeasures raises concerns, especially when reliable decisions are expected in critical domains such as healthcare. Therefore, in this section, we also propose a simulation method to verify these issues and conduct experiments to discuss the two types of risk mentioned above.

In particular, on the basis of the discussion of causality in the realm of health-screening as a use case, we demonstrate that certain patterns exist where even for GPT-4, it is obvious that causal relationships cannot exist, and that false positives do not occur even if GPT-4 undergoes SCP with the wrong causal relationships suggested by SCD. In addition, we identify patterns that are considered ground truth from domain knowledge but are not necessarily obvious to GPT-4, resulting in a risk of false negatives through SCP. Furthermore, the potential improvement of the prompt methodology for stabilizing the GPT-4 confidence probability measurement is also discussed. Finally, we discuss realistic strategies to overcome these problems when applying our proposed method.

## Simulation Methods for the Assessment of Robustness and Vulnerabilities of the Proposed Method and the Influence of SCP

For the simulations mentioned above, we make a minor change in the method shown in Figure 1, and the overall process of these simulations is illustrated in Figure 3. To investigate quantitatively the effect of SCP including the causal coefficients or bootstrap probabilities, on the confidence of GPT-4 in causal relationships, we focus on the case of DirectLiNGAM.

Process A is based on the first step in Figure 1, and "pseudo-results" of SCD is prepared for SCP. We prepare the SCD results from DirectLiNGAM, and find the edge that should obviously not appear, or should appear from the domain knowledge. By changing the causal coefficient $c$ and bootstrap probability $b$ of that edge from the DirectLiNGAM result, "pseudo-results" can be easily manufactured. Processes B and C correspond to the second and third steps in Figure 1, and we evaluate the average probability $p_{ij}(c, b)$ of the output in Process C, as a function of the input causal coefficient $c$ and the bootstrap probability $b$.

To clearly examine the dependency of $p_{ij}$ on $c$ and $b$, we consider the following prompt patterns from all the patterns shown in Section 3.2:

- Pattern 2: Prompting only with the bootstrap probabilities (without assigning causal coefficients). In our simulations, $p_{ij}$ is taken in this pattern while varying $b$ in increments of 0.1, from 0 to 1.

- Pattern 3: Prompting only with the causal coefficients (without assigning bootstrap probabilities).In our simulations, $p_{ij}$ is taken in this pattern while varying $c$ in increments of 0.05, from $-2$ to 2.

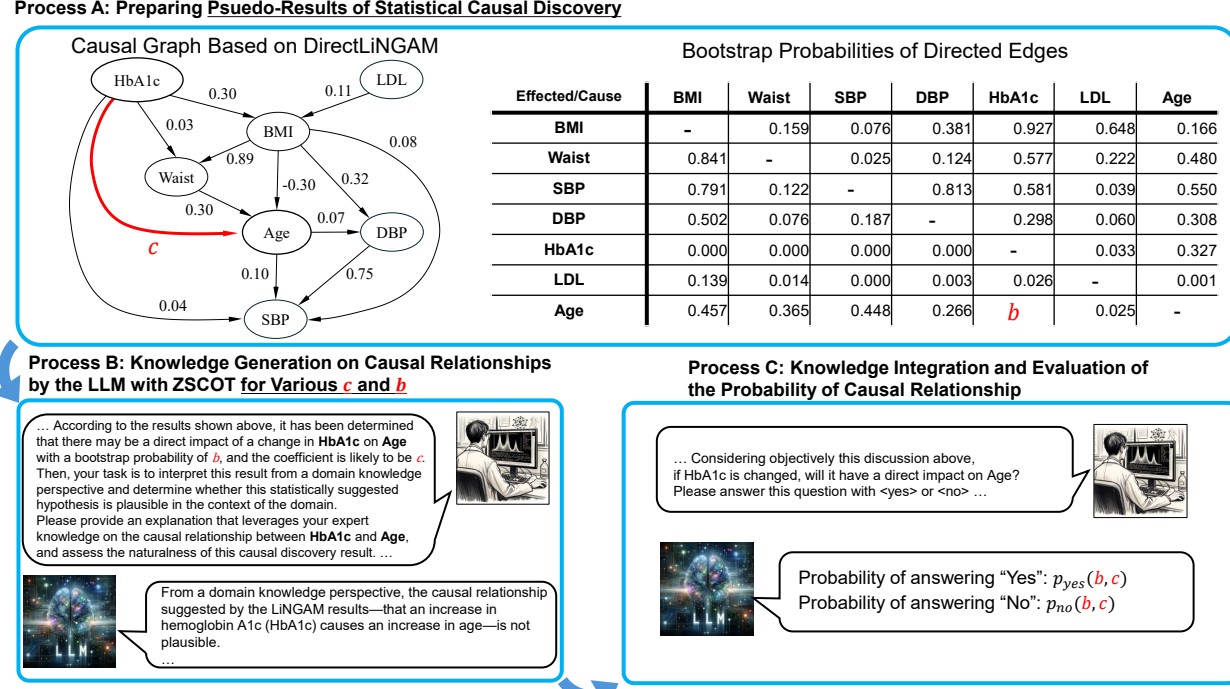

Figure 3: Framework of the simulations for assessing the robustness and vulnerabilities of the proposed method and the influence of SCP. In particular, the processes of Simulation A (for assessing the LLM's confidence in edges, where causal relationships are recognized as unreasonable considering domain knowledge) are exemplified, and the entire framework is almost the same as the proposed method illustrated in Figure 1.

- Pattern 4: Prompting with causal coefficients and bootstrap probabilities (assigning both). In our simulations, $p_{ij}$ is taken in this pattern while varying $b$ in increments of 0.1, from 0.1 to 1 and varying $c$ in increments of 0.1, from $-1$ to $1$[20].

In the simulation, following the experimental setup shown in Section 3.1, we adopted `GPT-4-1106-preview`, set the temperature to 0.7, and use a single-shot measurement of repetition count of $M = 5$, calculating the average of $p_{ij}(c, b)$ over five runs as $\overline{p_{ij}}(c, b)$.

By examining the dependency of the average probability $\overline{p_{ij}}(c, b)$ on the coefficient $c$ and bootstrap probability $b$, we analyze the performance of GPT-4 and the influence of SCP. If $\overline{p_{ij}}$ consistently takes values close to 0 or 1 regardless of these parameters, it suggests that GPT-4 has firm domain knowledge for recognizing an edge as extremely unnatural (in the case of 0) or natural (in the case of 1). Conversely, if $\overline{p_{ij}}(c, b)$ changes depending on $c$ or $b$, it indicates that GPT-4's knowledge about the existence of the edge is not firm, and that SCP can influence its judgment.

**Simulation A: LLMs' Confidence in Edges Where Causal Relationships Are Recognized as Unnatural (Discussion on Firmly Successful Cases)**

First, as in Simulation A, we examine cases where obviously unnatural edges appear, such as "HbA1c" $\rightarrow$ "Age", and investigate whether changes in causal coefficients or bootstrap probabilities affect the confidence probabilities. We base this analysis on the DirectLiNGAM results from a subsampled dataset of 1,000 instances where such edges actually appear, as shown in Figure 2(a). Process A in Figure 3 clearly illustrates

---

[20]Although the experimental range of $c$ in Pattern 3 is $-2 \leq c \leq 2$ for thorough investigation of the dependence on input $c$, we shrink this range in Pattern 4 into $-1 \leq c \leq 1$, for practical feasibility of the experiments in the two-dimensional parameter space

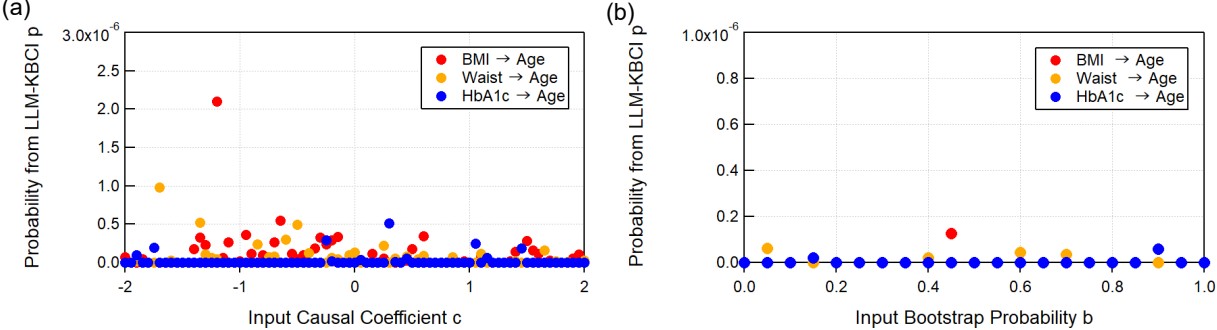

Figure 4: Results of Simulation A through the methodology illustrated in Figure 3, with the pseudo-results based on the SCD results for the subsampled health-screening dataset, which is used in the experiment of Section 4.2. Simulations were conducted on unrealistic edges "BMI" → "Age", "Waist" → "Age" and "HbA1c" → "Age." (a) Results for Pattern 2 (various values of causal coefficient $c$ are prompted with in Process B). (b) Results for Pattern 3 (various values of bootstrap probability $b$ are prompted with in Process B). Considering that the maximum values of the left vertical axes for these graphs are smaller than $1.0 \times 10^{-5}$, GPT-4's confidence probability in these unrealistic edges is fixed around 0, regardless of the causal coefficients or bootstrap probabilities in the pseudo-results of SCD.

Table 5: Results of linear regression of $\overline{p}_{ij}$ with the causal coefficient $c$ and the bootstrap probability $b$, for the results of Simulation A on unrealistic edges "BMI" → "Age," "Waist" → "Age," and "HbA1c" → "Age." The values enclosed in parentheses are the root mean squared error (RMSE) obtained through regression. Although we do not discuss whether these coefficients and constants are significantly different from 0 at any significance level, it is obviously illustrated that $\overline{p_{ij}}$ does not depend on $c$ or $b$ for the directed edges above, since all the values are approximately 0 (all of the absolute values are less than $10^{-6}$).

| Causal Relationship | Variable of Regression | Pattern 2 | Pattern 3 | Pattern 4 |
|---|---|---|---|---|
| **"BMI" → "Age"** | **Constant** | $7.7 \times 10^{-9}$ $(\pm 1.2 \times 10^{-8})$ | $1.0 \times 10^{-7}$ $(\pm 2.8 \times 10^{-8})$ | $3.3 \times 10^{-9}$ $(\pm 6.8 \times 10^{-9})$ |
| | **Causal Coefficient $c$** | — | $-3.3 \times 10^{-8}$ $(\pm 2.4 \times 10^{-8})$ | $-9.8 \times 10^{-9}$ $(\pm 5.2 \times 10^{-9})$ |
| | **Bootstrap Probability $b$** | $-3.3 \times 10^{-9}$ $(\pm 2.1 \times 10^{-8})$ | — | $1.7 \times 10^{-8}$ $(\pm 1.1 \times 10^{-8})$ |
| **"Waist" → "Age"** | **Constant** | $1.2 \times 10^{-8}$ $(\pm 7.4 \times 10^{-9})$ | $5.3 \times 10^{-8}$ $(\pm 1.5 \times 10^{-8})$ | $-1.5 \times 10^{-8}$ $(\pm 3.0 \times 10^{-8})$ |
| | **Causal Coefficient $c$** | — | $-2.5 \times 10^{-8}$ $(\pm 1.3 \times 10^{-8})$ | $-2.1 \times 10^{-9}$ $(\pm 2.3 \times 10^{-8})$ |
| | **Bootstrap Probability $b$** | $-9.4 \times 10^{-9}$ $(\pm 1.3 \times 10^{-8})$ | — | $7.9 \times 10^{-8}$ $(\pm 4.8 \times 10^{-8})$ |
| **"HbA1c" → "Age"** | **Constant** | $-5.7 \times 10^{-10}$ $(\pm 5.7 \times 10^{-9})$ | $2.1 \times 10^{-8}$ $(\pm 8.5 \times 10^{-9})$ | $6.8 \times 10^{-8}$ $(\pm 3.3 \times 10^{-8})$ |
| | **Causal Coefficient $c$** | — | $1.7 \times 10^{-9}$ $(\pm 7.2 \times 10^{-9})$ | $2.2 \times 10^{-9}$ $(\pm 2.5 \times 10^{-8})$ |
| | **Bootstrap Probability $b$** | $8.6 \times 10^{-9}$ $(\pm 9.7 \times 10^{-9})$ | — | $-5.9 \times 10^{-8}$ $(\pm 5.3 \times 10^{-8})$ |

how the causal graph and bootstrap probability matrix are modified and mounted in SCP, in the case of examining "HbA1c" → "Age". In addition, we also examine the cases of "Waist" → "Age" and "BMI" → "Age", all of which are similarly unnatural paths, since "Age" is determined only by birthday and the passage of time.

Figure 4 shows the values of $\overline{p_{ij}}$ and its dependence on causal coefficients in (a) and on bootstrap probabilities in (b). These graphs confirm that, regardless of the values of $c$ or $b$, the confidence probabilities in causal relationships of "HbA1c" $\rightarrow$ "Age", "Waist" $\rightarrow$ "Age" and "BMI" $\rightarrow$ "Age" remain almost zero (below $1 \times 10^{-5}$)).

Moreover, the results of the regression analysis are also shown in Table 5. Here, the function for the regression [21] is set as follows:

$$\overline{p_{ij}} = \begin{cases} p_0 + \alpha c & \text{(pattern 2)} \\ p_0 + \beta b & \text{(pattern 3)} \\ p_0 + \alpha c + \beta b & \text{(pattern 4)} \end{cases} \tag{6}$$

Here, $\alpha$ is the coefficient for $c$ (the causal coefficient in the pseudo-results of SCD), $\beta$ is the coefficient for $b$ (the bootstrap probability in the pseudo-results of SCD), and $p_0$ is the constant. From the regression results shown in Table 5, it can be confirmed that $\overline{p_{ij}}$ for the unnatural causal assumption "HbA1c" $\rightarrow$ "Age," "Waist" $\rightarrow$ "Age," and "BMI" $\rightarrow$ "Age" is indeed independent of $c$ and $b$, staying around 0.

This finding suggests that, regardless of how the SCP is performed, the judge of GPT-4 is firm, and it can correctly judge that such unnatural edges as that $\overline{p_{ij}}$ for unnatural causal assumptions "HbA1c" $\rightarrow$ "Age", "Waist" $\rightarrow$ "Age" and "BMI" $\rightarrow$ "Age" should not appear. Therefore, it is possible to identify these unnatural causal hypotheses that even GPT-4 can reject regardless of SCP through the experiments above, and our method indeed effectively contributes to generating prior knowledge for SCD, which effectively avoids the risk of false positives. As a result, the proposed method makes it possible to focus on truly possible causal relationships from the experts' point of view.

### Simulation B: LLMs' Confidence in Edges Where Causal Relationships Should Be Recognized and the Influence of SCP (Discussion on Failure Cases)

On the other hand, it is also important to discuss how the LLMs are confident in the causal relationships that are widely accepted as ground truths from an expert's point of view. To clarify this problem, we try Simulation B, the method of which is almost the same as that of Simulation A shown in Figure 3, except for the targeted edges in this simulation, and the base of the pseudo-results included in SCP.

Although the basic experimental processes of Simulation B are the same as those of Simulation A illustrated in Figure 3, there are two types of modifications from Simulation A to demonstrate the GPT-4's confidence test in ground truth directed edges as follows:

- The basis of the pseudo-results generated in process A is replaced with the DirectLiNGAM results[22] on "entire" health-screening data with 123151 points, as illustrated in Figure 5. Since this basis holds all the ground truth directed edges, the causal coefficients and bootstrap probabilities of these edges can be easily modulated in Process A.

- The modulated directed edges are replaced with "Age" $\rightarrow$ "BMI," "Age" $\rightarrow$ "SBP," "Age" $\rightarrow$ "DBP," and "Age" $\rightarrow$ "HbA1c." (from "HbA1c" $\rightarrow$ "Age," "Waist" $\rightarrow$ "Age," and "BMI" $\rightarrow$ "Age")

The experimental results in the case of "Age" $\rightarrow$ "HbA1c" are graphically illustrated in Figure 6 as examples, and the entire results of linear regression of $\overline{p_{ij}}$ with causal coefficient $c$ and the bootstrap probability $b$ are shown in Table 6. From Figure 6 (a), although a large and irregular fluctuation of $\overline{p_{ij}}$ is clearly observed, the dependence of $\overline{p_{ij}}$ on the causal coefficient $c$ in the case of "Age" $\rightarrow$ "HbA1c" is also confirmed: whereas $\overline{p_{ij}}$ is relatively unstable and rarely approaches 1 when $c < 0$ is satisfied, it becomes more stable and often approaches 1 when $c > 0$ is satisfied. Conversely, as illustrated in Figure 6 (b), the dependence of $\overline{p_{ij}}$ on the bootstrap probability $b$ is more unstable than that on the causal coefficient $c$. However, it is also clearly confirmed that $\overline{p_{ij}}$ never becomes stable even when $b$ approaches 0 or 1.

---

[21] For more strict discussion, since the range of the confidence probability $\overline{p_{ij}}$ is $0 \le \overline{p_{ij}} \le 1$, it is more natural to introduce a logistic function for fitting. However, to simply understand the dependence of $\overline{p_{ij}}$ on $c$ or $b$, we tentatively adopt the linear function for the regression in this paper.

[22] The causal graph and bootstrap probability matrix are shown in Appendix C.4.

**Pseudo-Results of SCD for Simulation B**
**(Testing the Influence of SCP on GPT-4's Confidence Probability for Ground Truth Edges)**

Causal Graph Based on DirectLiNGAM
for entire health-screening dataset with 123151 points

Bootstrap Probabilities of Directed Edges

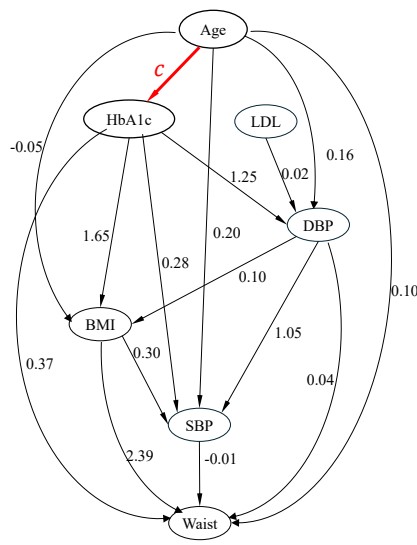

| Effected/Cause | BMI | Waist | SBP | DBP | HbA1c | LDL | Age |
|---|---|---|---|---|---|---|---|
| **BMI** | – | 0.000 | 0.000 | 1.000 | 1.000 | 1.000 | 1.000 |
| **Waist** | 1.000 | – | 1.000 | 1.000 | 1.000 | 0.000 | 1.000 |
| **SBP** | 1.000 | 0.000 | – | 1.000 | 0.723 | 0.720 | 1.000 |
| **DBP** | 0.000 | 0.000 | 0.000 | – | 1.000 | 1.000 | 1.000 |
| **HbA1c** | 0.000 | 0.000 | 0.000 | 0.000 | – | 1.000 | *b* |
| **LDL** | 0.000 | 0.000 | 0.000 | 0.000 | 0.000 | – | 0.000 |
| **Age** | 0.000 | 0.000 | 0.000 | 0.000 | 0.000 | 0.000 | – |

Figure 5: Example of pseudo-results of DirectLiNGAM in the entire health-screening dataset with 123151 for Simulation B. In this example, on the basis of the initial DirectLiNGAM results shown in Figure 13, the causal coefficient and the bootstrap probability of the directed edge "Age" → "HbA1c" are replaced with the variables $c$ and $b$, respectively. (The value of the causal coefficient without replacement is 0.02, and the bootstrap probability is 1.000.) Because of the large number of data points in the entire health-screening dataset, some directed edges, including the ground truth bootstrap probabilities, are 1.000. In Simulation B, in addition to this example, the pseudo-results modulating the causal coefficients and bootstrap probabilities of "Age" → "BMI," "Age" → "SBP," and "Age" → "DBP," are also embedded similarly to Figure 3.

Moreover, the results of linear regression of $\overline{p_{ij}}$ with $c$ or $b$ are summarized in Table 6, for all the directed edges of "Age" → "BMI," "Age" → "SBP," "Age" → "DBP," and "Age" → "HbA1c." In contrast to the results of Simulation A illustrated in Table 5, all the absolute values of the estimated coefficients and constants are greater than 0.01. In addition, many of the estimated values are considered statistically significant at the 1 % level. These results suggest that GPT-4's confidence probabilities of these directed edges, which have been accepted as ground truths by medical experts, can be somehow influenced by the incorporation of $c$ or $b$ into SCP. Moreover, the minus coefficients of $b$ are suggested in Pattern 2 of "Age" → "SBP." From this result, it can be interpreted that the higher the bootstrap probability $b$, which indicates a greater likelihood of statistical causal relationships, can sometimes lead to a lower confidence probability of GPT-4, although the detailed mechanism is unknown. Therefore, it must be recognized that SCP can also induce the risk of false negatives, especially when LLMs are not absolutely confident in the judgment of causal relationships. Adopting these uncertain results as prior knowledge could lead to incorrect causal graph generation.

Missing true causal relationships is critical, especially in important domains such as healthcare. Therefore, for practical applications of the proposed method, we discuss below how to mitigate this risk from both technical and operational perspectives.

### Toward Stabilization of LLMs' Confidence Probability Measurement with Improvement of Prompting Techniques

For more practical use, the origin of the large fluctuations of $\overline{p_{ij}}$ illustrated in Figure 6 should also be discussed. The main causes of variation in confidence probability are considered as follows:

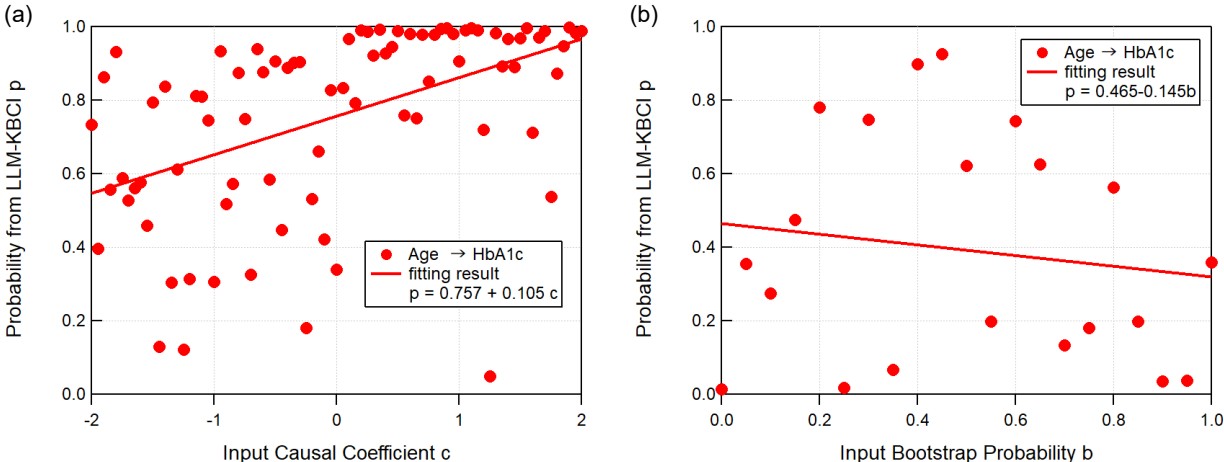

Figure 6: Representative results of Simulation B through the methodology illustrated in Figure 3, with the pseudo-results based on the SCD results for the full health-screening dataset shown in Figure 5. Simulations were conducted on unrealistic edges "Age" → "BMI," "Age" → "SBP," "Age" → "DBP," and "Age" → "HbA1c," we representatively show the results on "Age" → "HbA1c."
(a) Results for Pattern 2 (various values of causal coefficient $c$ are prompted with in Process B). although a large and irregular fluctuation of $\overline{p_{ij}}$ is observed, the dependence of $\overline{p_{ij}}$ on the causal coefficient $c$ in the case of "Age" → "HbA1c" is also confirmed: while $\overline{p_{ij}}$ is relatively unstable and rarely approaches 1 when $c < 0$ is satisfied, it becomes more stable and often approaches 1 when $c > 0$ is satisfied.
(b) Results for Pattern 3 (various values of bootstrap probability $b$ are prompted with in Process B). The dependence of $\overline{p_{ij}}$ on the bootstrap probability $b$ is more unstable than that on the causal coefficient $c$. However, it is also clearly confirmed that $\overline{p_{ij}}$ never becomes stable even when $b$ approaches 0 or 1.

- The dependency of GPT-4's judgment in Process C of Figure 6 (Step 3 of Figure 2) on the content of the sentences generated in Process B (Step 2 of Figure 2)

- Fluctuations in the log-probabilities in Process C (Step 3 of Figure 2) itself (as already explained in Section 3.1)

Although the latter can be mitigated by evaluating $\overline{p_{ij}}$ obtained through multiple measurements, the former arises from the separation of GPT-4's judgment process into Processes B and C (Steps 2 and 3 in Figure 1). As explained in Sections 1.2 and 3.1, we have adopted the method of generated knowledge prompting (Liu et al., 2022) to suppress hallucination. Therefore, each task of Processes B and C (Steps 2 and 3 in Figure 1) is relatively simple, and this separation also makes it possible to detect the tendency of the hallucination, which can further improve the performance of the LLM's causal reasoning with its domain knowledge. For example, if $\overline{p_{ij}}$ generated in Process C (Step 3 in Figure 1) is completely different from the expectation of human experts, there may be a fatal defect in the factual statement or logic in the generated knowledge in Process B (Step 2 in Figure 1), which leads to incorrect judgment in Process C (Step 3 in Figure 1). In that case, checking the generated texts in Process B (Step 2 in Figure 1) from an expert's point of view can also be an effective process for improving the performance of the proposed method. Moreover, this additional process with domain experts can lead to the discovery of new discussions on causal relationships, especially when there is no systematic understanding of the causality between a pair of variables.

However, careful attention is also required so that analysis of the long text generated in Process B (Step 2 in Figure 1) is complicated, because all the tokens are randomly generated with the next token prediction on the basis of the probabilistic process. To illustrate this difficulty, we analyze the fluctuation observed around

Table 6: Results of linear regression of $\overline{p}_{ij}$ with the causal coefficient $c$ and the bootstrap probability $b$, for the results of the simulation on realistic edges "Age" → "BMI", "Age" → "SBP", "Age" → "DBP" and "Age" → "HbA1c," which are strongly supported in the medical domain and are interpreted as ground truths. The values enclosed in parentheses are the RMSEs obtained through regression. The asterisks $*$, $**$, and $***$ indicate that the coefficients or constants are significantly different from zero at the 10, 5, and 1 % levels, respectively.

| Causal Relationship | Variable of Regression | Pattern 2 | Pattern 3 | Pattern 4 |
|---|---|---|---|---|
| "Age" → "BMI" | Constant | 0.1347 * (± 0.0764) | 0.6623 *** (± 0.0312) | 0.0558 (± 0.0361) |
| | Causal Coefficient $c$ | — | -0.0491 * (± 0.0267) | -0.0661 ** (± 0.0276) |
| | Bootstrap Probability $b$ | 0.0243 (± 0.1310) | — | 0.6000 *** (± 0.0581) |
| "Age" → "SBP" | Constant | 0.7786 *** (± 0.1180) | 0.8106 *** (± 0.0255) | 0.8106 *** (± 0.0292) |
| | Causal Coefficient $c$ | — | -0.0553 ** (± 0.0218) | -0.0114 (± 0.0223) |
| | Bootstrap Probability $b$ | -0.8103 *** (± 0.2010) | — | -0.0101 (± 0.0471) |
| "Age" → "DBP" | Constant | 0.5438 *** (± 0.1070) | 0.8058 *** (± 0.0268) | 0.3681 *** (± 0.0411) |
| | Causal Coefficient $c$ | — | -0.0284 (± 0.0229) | 0.0548 * (± 0.0314) |
| | Bootstrap Probability $b$ | -0.3726 ** (± 0.1830) | — | 0.4405 *** (± 0.0662) |
| "Age" → "HbA1c" | Constant | 0.4650 *** (± 0.1330) | 0.7570 *** (± 0.0247) | 0.1814 *** (± 0.0393) |
| | Causal Coefficient $c$ | — | 0.1048 *** (± 0.0211) | 0.1915 *** (± 0.0300) |
| | Bootstrap Probability $b$ | -0.1446 (± 0.2270) | — | 0.5272 *** (± 0.0633) |

$c = 1.25$ in Figure 6(a) in terms of the generated text in Process B. Whereas $\overline{p_{ij}}$ becomes only 0.0493 when $c = 1.25$, $\overline{p_{ij}}$ becomes 0.9833 when $c = 1.30$. To elucidate the origin of this sudden behavior, we compare the generated text in Process B for $c = 1.25$ illustrated in Table 7, with the text for $c = 1.30$ illustrated in Table 8. It is confirmed that both contents of the texts in these tables do not include critical errors, from the medical domain experts' points of view. However, although only the value $c$ is changed, the textual composition and the tones of generated knowledge in Process B radically change as follows:

- While the discussion is separated into the detailed aspects of the physiological factors of glucose metabolism in the generated text in Table 7, the text of Table 8 breaks down the issue into the validity of statistical causal inference, not only from the medical point of view, but also from the perspective of statistical causal inference methodology.

- While the statement in Table 7 is based on the connection between variables, Table 8 presents the discussion of causality more aggressively.

- While the validity pseudo-result of the SCD is carefully interpreted in terms of the interpretation of the magnitude of the causal coefficient in Table 7, the discussion in Table 8 is positively stated including the magnitude interpretation and consistency with the previous studies.

Table 7: Textual output in Process B of Simulation B in the case of "Age" → "HbA1c" when the causal coefficient is modulated as $c = 1.25$ in SCP Pattern 2.

---

**Output in Process B for $c = 1.25$  (Pattern 2)**

---

The LiNGAM algorithm has suggested a direct causal relationship between age and hemoglobin A1c (HbA1c) with a positive coefficient of 1.25. This implies that as age increases, HbA1c levels are also expected to increase, according to the model.

From the perspective of domain knowledge in medicine and epidemiology, the relationship between age and HbA1c can be assessed for plausibility. Hemoglobin A1c is a measure of the average blood glucose levels over the past two to three months and is used as an indicator of blood sugar control in individuals with diabetes. It reflects the glycation of hemoglobin, a process that is influenced by the concentration of glucose in the blood.

As people age, several physiological changes can affect glucose metabolism:

**1. Insulin Sensitivity:** Aging is often associated with a decrease in insulin sensitivity, which is also known as insulin resistance. This can lead to higher blood glucose levels because the body's cells are less effective at absorbing glucose from the bloodstream.

**2. Pancreatic Function:** Beta-cell function in the pancreas may decline with age, leading to less insulin production. Insulin is necessary for glucose uptake in the cells, and a deficit can result in higher blood glucose levels.

**3. Lifestyle Factors:** Older adults may have changes in their lifestyle, such as decreased physical activity and altered dietary patterns, which can contribute to poorer glycemic control.

**4. Comorbidities:** With age, the prevalence of comorbid conditions that can impact glucose metabolism, such as obesity and chronic inflammation, tends to increase.

**5. Hormonal Changes:** Hormonal regulation changes with age and can affect glucose metabolism. For example, growth hormone levels and cortisol rhythms can influence blood sugar levels.

Given these factors, an increase in HbA1c with age can be considered plausible from a biological perspective. However, the coefficient of 1.25 is a standardized metric, and its interpretation depends on the scale of standardization. In a non-standardized real-world setting, this coefficient would typically be much smaller, as the actual increase in HbA1c due to aging alone might be modest. But without the context of the units and the standard deviations of the age and HbA1c variables, it's difficult to assess the magnitude of this relationship's impact.

It's also important to note that while the causal relationship suggested by LiNGAM is statistically inferred, causal claims should also be substantiated by domain knowledge and, if possible, experimental or longitudinal studies. The plausibility of the relationship in this context is supported by domain knowledge, but other factors not included in the model could also be influencing both age and HbA1c levels.

In conclusion, the statistical relationship discovered is consistent with domain knowledge about the physiology of aging and its effect on glucose metabolism. However, the strength and practical significance of this relationship would need further investigation, ideally through longitudinal studies and intervention trials, to confirm the causal pathways and to quantify the effect more precisely.

---

**confidence probability generated in Process C:** $\overline{p_{ij}} = 0.0493$

---

Table 8: Textual output in Process B of Simulation B in the case of "Age" $\rightarrow$ "HbA1c" when the causal coefficient is modulated as $c = 1.30$ in SCP Pattern 2.

---

**Output in Process B for $c = 1.30$  (Pattern 2)**

---

The result from the LiNGAM analysis suggests that there is a direct causal effect of age on hemoglobin A1c (HbA1c) levels with a coefficient of approximately 1.3. This implies that, in the structural causal model derived from your data, an increase in age is associated with an increase in HbA1c levels. To interpret this finding from a domain knowledge perspective, we need to consider the biological relationship between age and HbA1c and whether the magnitude and direction of the effect make sense.

Hemoglobin A1c is a form of hemoglobin that is chemically linked to glucose. The HbA1c test measures the average blood glucose levels over the past 2 to 3 months. It is commonly used as an indicator of long-term glucose control in individuals with diabetes and as a diagnostic tool for diabetes and prediabetes.

From a medical perspective, it is known that as people age, they tend to experience various physiological changes that can affect glucose metabolism. This includes changes such as increased insulin resistance, decreased insulin secretion, and altered hormonal control of blood glucose levels. These changes can contribute to higher blood glucose levels and therefore higher HbA1c levels in older adults.

The plausibility of the causal relationship suggested by the statistical analysis can be assessed by considering the following factors:

**1. Biological Mechanism:** There is a clear biological mechanism by which aging can affect glucose metabolism and insulin function, leading to higher HbA1c levels.

**2. Consistency with Previous Research:** The direction of the effect is consistent with epidemiological studies that have found a positive association between age and HbA1c levels.

**3. Magnitude of the Effect:** The coefficient of 1.3 is a standardized coefficient, representing the change in standard deviations of HbA1c for each standard deviation change in age. The magnitude of this effect should be interpreted within the context of the data distribution and the units of measurement. In general, the magnitude seems large, and it would be important to compare it to existing literature to see if such a change is reasonable.

**4. Confounding Factors:** The LiNGAM analysis assumes that all relevant variables have been measured and included in the model. If there are unmeasured confounders that are associated with both age and HbA1c, the estimated causal effect could be biased. For example, physical activity, diet, and other lifestyle factors can change with age and affect HbA1c levels.

**5. Population-Specific Factors:** The analysis is based on working-age adults (40 to 64 years old). The effect of age on HbA1c within this specific age range may differ from that in younger or older populations.

In conclusion, the statistically suggested hypothesis that age has a direct impact on HbA1c is biologically plausible and consistent with prior knowledge. However, further analysis should be conducted to ensure that all relevant confounding factors have been accounted for, and the magnitude of the effect should be compared with existing research to validate the results. Additionally, the clinical significance of this finding should be evaluated in the context of the overall health and management of individuals in the working-age population.

---

**confidence probability generated in Process C:** $\overline{p_{ij}} = 0.9833$

---

Consequently, it is inferred that these differences in aspects and tones lead to a significant gap in the confidence probabilities of GPT-4 at the same directed edge "Age" → "HbA1c." However, because the only difference in the SCP between Tables 7 and 8 is whether the causal coefficient of this edge is 1.25 or 1.30, the main factor of these gaps in aspects and tones is the complexity of sentence generation with a probabilistic process.

Therefore, considering the discussion above, it is practically challenging to quantitatively elucidate the tendencies of sentences generated in Process B (Step 2 in Figure 1), including the analysis of reproducibility, the factual likelihood and the tone of the generated knowledge as a long text. For the suppression and realistic evaluation of the fluctuation of GPT-4's confidence probability in Step 3 of the proposed method in this paper, other substitute prompt techniques should be considered.

For example, a more straightforward approach might be to integrate Steps 2 and 3 in Figure 1, applying the approach of "LLM-as-a-Judge" (Zheng et al., 2023a; Yuan et al., 2021; Kocmi & Federmann, 2023). Under the condition of maintaining the measurement of GPT-4's confidence probability in causal relationships in the proposed method, the position for the measurement of token generation probability and for the generation of "yes" or "no" in the generated text from GPT-4 must be fixed to the first or last one, from a practical point of view. Considering the limitations of ZSCoT and the property of the next generation of tokens from LLMs, fixing to the last token produces a better result than fixing to the first token, because the context generated before the token "yes" or "no" is expected to suppress hallucination.

However, continuous caution will still be required for the reproducibility of the last token generation probability. In general, the probability of token generation $w_n$ is expressed by the conditional probability related to the prompt $Q$ and the $\{w_1, w_2, ..., w_{n-1}\}$ as follows:

$$P(w_n|Q, w_1, w_2, ..., w_{n-1}) \tag{7}$$

Therefore, the probability that the last generated token $w_L$ becomes "yes," is calculated as follows:

$$P(w_L = \text{yes}) = \sum_{\text{all}} \prod_{k=1}^{L-1} P(w_k|Q, w_1, w_2, ..., w_{k-1})P(w_L = \text{yes}|Q, w_1, w_2, ..., w_{L-1}) \tag{8}$$

In contrast, the only probability that can be repeatedly measured from a practical point of view, is $P(w_L = \text{yes}|Q, w_1, w_2, ..., w_{L-1})$ in Eq.(8). Although it is possible to evaluate the mean value or distribution of $P(w_L = \text{yes}|Q, w_1, w_2, ..., w_{L-1})$, which can be measured through repetitive measurements as carried out in this paper, there is no guarantee that the true value of $P(w_L = \text{yes})$ can be obtained approximately, as it has not been clarified how $\prod_{k=1}^{L-1} P(w_k|Q, w_1, w_2, ..., w_{k-1})$ can be measured. Therefore, the integration of Steps 2 and 3 can hardly be considered a fundamental solution, since the effective method for the stabilization of tones and aspects of the generated knowledge for judgment is still lacking.

Consequently, regardless of the integration or separation of Steps 2 and 3, the prompting technique for the stabilization of the generated knowledge is expected to be constructed in detail. Although the discussion of this problem from this point forward requires further complex and careful examination beyond this paper, the possible aspects are as follows:

- **Appropriately enumerating the perspectives that need consideration, distinguishing between those that depend on the domain's inherent characteristics and those that do not**
  This comprehensive listing of perspectives, and including them in the prompt can lead to appropriate domain-specific nuances, while maintaining the consistency of analytical knowledge generation and tone.

- **Confirmation of the characteristics of each variable and grouping, before the interpretation process following the listed perspectives as mentioned above**
  In the case of a health-screening dataset, for example, "SBP" and "DBP" are similar variables, that are simultaneously measured in the health-screening process and share the same unit of measurement.

Therefore, these two variables are expected to have similar causal relationships with other variables, and that "SBP" and "DBP" are somehow correlated, although the detailed causal structure, including "SBP" and "DBP" has not been completely determined. However, as discussed earlier, the variable "Age" has a completely different characteristic from the other variables. This variable is mechanically determined from the birthday and the passage of time. Confirmation of LLMs' understanding of such properties of the variables is expected to contribute to the suppression of hallucinations.

- **A careful examination of how the SCD results in the SCP are interpreted, particularly in handling anomalies that arise owing to limitations of the models for SCD**
  For example, the causal graph discovered through any SCD algorithm without prior knowledge, sometimes displays inappropriate directed edges with a large bootstrap probability, whose orientations are judged as obviously reversed from the experts' point of view, because of the strong statistical correlation between the pairs of variables. Since the SCP proposed in our method includes all the results of SCD, it is expected that the statistical information included in the prompts is handled effectively and appropriately, considering the tendency and limitations of SCD. Addressing these issues necessitates meticulous discussion to solidify a comprehensive understanding of the results, highlighting a significant area for future research.

- **Transforming the standardized coefficients back to values in their original dimensions and including them into the prompts  (for Patterns 2 and 4)**
  In the method proposed in this study, the causal coefficients are calculated via DirectLiNGAM, with the datasets standardized before causal discovery, and without transforming them back to the values with original units. We have intentionally adopted the standardized causal coefficients, to ensure consistent scaling, which can lead to a relative comparison of the causal coefficients between different directed edges. However, as shown in Table 7, GPT-4 can insist, "without the context of the units and the standard deviations of the age and HbA1c variables, it is difficult to assess the magnitude of this relationship's impact." Therefore, if it is expected that LLMs possess benchmarks or knowledge for judging the validity of prompted coefficients in some units, including the coefficients that are transformed back into the original dimensions into the prompt may enhance the capability of quantitative judgment.

Once this framework is in place, adopting the technique of few-shot prompting  (Brown et al., 2020) with accurate examples can further improve the stability and reliability of the model output. By providing the LLMs with ideal and correct exemplars, one can improve their ability to generate stabilized, consistent, and accurate responses, which contributes to the practical discussion of the reproducibility of LLM-KBCI.

To achieve even greater consistency in the outputs, the instruction tuning  (Wei et al., 2022) can also be adopted as an alternative to few-shot prompting. Instruction tuning involves fine-tuning the model via a dataset of instruction-response pairs, aligning the model's behavior more closely with specific analytical requirements. However, this approach presents practical challenges: the cost of data generation for instruction tuning is much higher than that for few-shot prompting, and only fully open-source models (such as Llama2 (Touvron et al., 2023) etc.) currently allow such fine-tuning. Therefore, although instruction tuning offers potential advantages in further stabilizing model outputs, its implementation is limited by resource constraints and model availability.

Moreover, fine-tuning or employing RAG with some domain-specific knowledge sources can be expected to enhance the domain knowledge of the LLM. These approaches could enable LLMs to make more appropriate judgments between various pairs of variables.  In particular, adopting RAG can also contribute to the stabilization of the tones of knowledge generation, because the range of information searches can be limited.

### Toward Appropriate Application of the Proposed Method

As illustrated above, while GPT-4 has completely denied apparently unnatural directed edges from the domain knowledge such as "HbA1c" → "Age" under any conditions, it has not always been confidently judged in the ground truth edges where causal relationships should exist, such as "Age" → "HbA1c." Moreover, the SCP can sometimes unnecessarily decrease the confidence probability of a ground truth edge in LLM-KBCI, although

it is still unknown what kind of ground truth causal relationships are likely to be worsened. Therefore, using our proposed method as a "black-box" may lead to unintended erroneous analysis results.

To mitigate this crucial risk as much as possible, first of all, although it is technically possible to connect all the steps shown in Figure 1 and fully automate as described in Algorithm 1, it is important to verify the validity of the output at each step. Recognizing this, we have carefully evaluated the results of all the steps throughout this work. In particular, to mitigate the risks arising from the use of LLMs, it is essential to assess the performance of the LLM-related processes, such as the second and third steps in Figure 1, and to flexibly incorporate other optimal prompting techniques to improve the reliability the results.

Based on this premise, in addition to further technical clarification as discussed above, benchmarking the basic knowledge of LLMs in the domain in which the application of the proposed method is to be applied, is also important. Furthermore, before applying our method to construct data-driven causal models, especially when there are firm prior knowledge edges that must be reflected, it is advisable to conduct preliminary experiments as performed in Simulations A and B. In addition, having experts review the actual results of the proposed method, including the textual output of Step 2, can be effective. Through these processes, it becomes possible to discuss what domain knowledge should be additionally learned by LLMs, for more precise and effective completion of the proposed method in each domain.

However, although high confidence can be achieved without critical errors through the approaches of adopting expert checks, especially when particularly focusing on small systems, the domain expert check becomes more impractical, as the number of variables increases, because the combinations of variable pairs increase quadratically. Because the strength of the proposed method in this paper lies in automating areas where expert verification becomes unmanageable owing to the large number of variables, premising expert checks in large-scale systems is counterproductive.

Therefore, continuing technical examinations, to overcome these issues, as discussed in this section, remain important, This ongoing effort is essential to enhance the reliability and practical applicability of our proposed method.

Finally, in terms of perfect realization of the expected role of the SCP, LLMs are also required to recognize the validity of the causal models prompted in the first step, and to point out the inadequate part. To achieve these capabilities, accumulation of the modification process by domain experts as mentioned above, and structuring as a dataset for additional reinforcement learning of LLMs are also valuable. This framework can be interpreted as reinforcement learning with human feedback (RLHF) of causal graphs, and is expected to contribute to further suppression of hallucinations, toward the practical application of SCP in any domain. It is also possible to introduce ensemble methods of several LLMs, which have been intensively studied to mitigate hallucinations(Maksimov et al., 2024).

## 5   Conclusion

In this study, a novel methodology for causal inference, in which SCD and LLM-KBCI are synthesized with SCP, and prior knowledge augmentation was developed and demonstrated.

It has been revealed that GPT-4 can cause the output of LLM-KBCI and the SCD result with prior knowledge from LLM-KBCI to approach the ground truth, and that the SCD result can be further improved, if GPT-4 undergoes SCP. Furthermore, with an unpublished real-world dataset, we have demonstrated that GPT-4 with SCP can assist in SCD with respect to domain expertise and statistics, and that the proposed method is effective even if the dataset has never been included in the training data of the LLM and the sample size is not sufficiently large to obtain reasonable SCD results.

In addition, we have further simulated the SCP under various conditions, and confirmed both successful and failure cases on the basis of the SCD results on the health-screening dataset with LiNGAM. In successful cases, the existence of patterns where the SCP stably works as intended has been confirmed, especially when the hypothesis of the causal relationship (such as the hypothesis that "Age" is influenced by other factors) is apparently unnatural. In contrast, through the discussion of the failure cases, the technical limitations and risks of critical errors have been thoroughly discussed. Through these discussions, we have presented the

expected improvements in techniques around LLMs, and the suggestion of realistic integration of domain expert checks of the results into this automatic methodology.

**Technical Limitations of This Work**

The details of the technical limitations and future directions for improving the proposed method in this paper are illustrated in Section 4.3. Here, we briefly discuss the general limitations and risks of adopting LLMs in causal inference tasks as the proposed method in this paper.

We fixed the LLM in our experiment to GPT-4 for its extensive general knowledge and capabilities in common-sense reasoning, and our experiments on several real-world datasets, typically, within the realm of common-sense reasoning, have illustrated the potential utility of the proposed method across various scientific domains. We expect that the proposed method will work even if other LLMs are adopted, because the capability of LLM-KBCI has already been demonstrated in several models (Zečević et al., 2023). Considering the continuous emergence of new LLMs and the rapid update of existing LLMs, it is expected that the universal effectiveness of the proposed method across different LLMs will be verified.

Moreover, it is also important to recognize the risks of using current LLMs in our methods: hallucinations at Steps 2 and 3 in Figure 1 can worsen the results of LLM-KBCI and SCD. Considering that hallucinations arise from factors such as memorization of training data (McKenna et al., 2023; Carlini et al., 2023), statistical biases (Jones & Steinhardt, 2022; McMilin, 2022), and predictive uncertainty (Yang et al., 2023; Lin et al., 2024), it is essential to be especially careful when our method is applied in highly specialized academic domains (George & Stuhlmueller, 2023; Pal et al., 2023). To reduce these risks when the proposed method is applied in domains with deeper specificity in the future, it is necessary to further improve our method to avoid hallucinations more effectively. Although we have already introduced ZSCoT (Kojima et al., 2022) and generated knowledge prompting (Liu et al., 2022), employing optimal LLMs that are specifically pretrained with materials on specific domains is one of the possible solutions. Moreover, incorporating techniques such as fine-tuning and RAG could be necessary. Therefore, systematic research in the context of optimal LLM-KBCI is also required on the selection of the optimal LLMs for each domain and on the techniques for utilizing the LLMs.

Furthermore, for the generalization of the method we have proposed, we also note that although SCP is indeed likely to enhance the performance of SCD more than prompting without SCP, whether this improvement is observed can depend on the datasets used for SCD. Moreover, there remains a discussion on the potential for overfitting in SCP, especially with small or biased datasets. Although we have shown the effectiveness of the SCP on a subsample of the health-screening dataset, which could contain biases, it is still an open question under what conditions the LLMs and prompt techniques can guarantee robustness for small or biased datasets. In terms of the more reliable application of the proposed method, further basic research on the effects of causal coefficients or bootstrap probabilities on the results of the LLM-KBCI is required.

**Broader Impact Statement**

This paper proposed a novel approach that integrates SCD methods with LLMs. This research has the potential to contribute to more accurate causal inference in fields where understanding causality is crucial, such as healthcare, economics, and environmental science. However, the use of LLMs such as GPT-4 necessitates the extensive consideration of data privacy and biases. Moreover, it should also be noted that incorrect causal inference, caused by hallucinations, biased training data, or inappropriate use of LLMs, could lead to significant consequences for society. In particular, these risks arise when the proposed method is used for completely "black-box" decision making. To prevent these risks, the transparency and interpretability of the whole system should be clarified.

This study highlights the responsible use of artificial intelligence, considering ethical implications and societal impacts. With appropriate guidelines and ethical standards, the proposed methodology can advance scientific understanding and provide extensive widespread benefits to society.

To mitigate the risk of data privacy and security properly, we have designed the method to avoid providing LLMs with the raw dataset, which could directly lead to the leakage of personally identifiable information. In

the proposed method, the SCD processes with the raw dataset and LLM processes are completely separated, and only statistical results such as causal coefficients and bootstrap probabilities are used in the processes with LLMs. Therefore, even when the raw dataset contains highly confidential information, analysis through the proposed method should not raise privacy and security concerns.

## Acknowledgements

We sincerely appreciate Dr. Hitoshi Koshiba for his invaluable assistance in shaping the overall structure of this paper and for engaging in fruitful discussions. This work was supported by the Japan Science and Technology Agency (JST) under CREST Grant Number JPMJCR22D2 and PRESTO Grant Number JPMJPR2123, and by the Japan Society for the Promotion of Science (JSPS) under KAKENHI Grant Numbers JP20K03743, JP23H04484 and JP24K20741.

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

# A  Ethics Review

**Ethical Considerations in Methodology and AI Use**  This paper proposes a novel approach that integrates SCD with LLMs. We have thoroughly considered the issues of data privacy and biases associated with the use of LLMs. The proposed methodology enhances the accuracy and efficiency of causal discovery; however, it does not introduce explicit ethical implications beyond those generally applicable to machine learning. We are committed to the responsible use of AI and welcome the scrutiny of the ethics review committee.

**Institutional Review and Consent Compliance for Health-Screening Data**  The institutional review board of Kyoto University approved this study. As we only analyzed anonymized data from the database, the need for informed consent was waived.

# B    Details of the Contents in Each Prompting Pattern

In this section, the details of the prompting in each pattern are presented. For Pattern 0, another prompting template is detailed instead of the sentences shown in Table 1. Moreover, for Patterns 1, 2, 3, and 4, the contents filled in ⟨blank 5⟩ and ⟨blank 9⟩ of the prompt template for the SCP shown in Table 1 are described.

**For Pattern 0**    Compared with other patterns of SCP, Pattern 0 does not include any results of SCD without prior knowledge. As a result, the prompt template in Table 1 is not suitable for Pattern 0, as it includes blanks filled with descriptions of the dataset and the SCD result. Thus, we prepare another prompt template for Pattern 0, which is completely independent of the SCD result, and requires GPT-4 to generate the response solely from its domain knowledge. Table 9 presents the prompt template in Pattern 0, which is composed mainly of the ZSCoT concept. Because it does not include information on the SCD method and relies solely on domain knowledge in GPT-4, the probability matrix obtained from GPT-4 in Pattern 0 is applied independently of the SCD method.

Table 9: The prompt template of $Q_{ij}^{(1)}(x_i, x_j)$ for Pattern 0 for the generation of expert knowledge of the causal effect from $x_j$ on $x_i$. The "blanks" enclosed with ⟨ ⟩ are filled with description words of the theme of the causal inference and variable names.

| **Prompt Template of $q_{ij}^{(1)} = Q_{ij}^{(1)}(x_i, x_j)$ in Pattern 0 (for all SCD methods)** |
| --- |
| We want to carry out causal inference on ⟨**blank 1. The theme**⟩, considering ⟨**blank 2. The description of all variables**⟩ as variables. |
| If ⟨**blank 7. The name of $x_j$**⟩ is modified, will it have a direct impact on ⟨**blank 8. The name of $x_i$**⟩? |
| Please provide an explanation that leverages your expert knowledge on the causal relationship between ⟨**blank 7. The name of $x_j$**⟩ and ⟨**blank 8. The name of $x_i$**⟩. 
 Your response should consider the relevant factors and provide a reasoned explanation based on your understanding of the domain. |

**For Patterns 1–4 (in the case of Exact Search and DirectLiNGAM)**    Following the concept of each SCP pattern, the contents filled in ⟨blank 5⟩ shown in Table 1 are summarized in Table 10. In this table, the names of the causes and effected variables are represented as ⟨ cause $i$ ⟩ and ⟨ effect $i$ ⟩, respectively, and the bootstrap probability of this causal relationship in SCD $P_i$ and the causal coefficient of LiNGAM $b_i$ can be included in Patterns 2–4. In Patterns 2 and 4, only the causal relationships with $P_i \neq 0$ are listed in ⟨ blank 5 ⟩. In Pattern 3, the causal relationships with $b_i \neq 0$ are listed in ⟨ blank 5 ⟩.

The contents of ⟨blank 6⟩ and ⟨blank 9⟩ also depend on the SCP patterns, and are shown in Table 11. Here, we define the bootstrap probability of $x_j \rightarrow x_i$ as $P_{ij}$. We also define the causal coefficient of $x_j \rightarrow x_i$ in LiNGAM as $b_{ij}$, because the structural equation of LiNGAM is usually defined as[23]:

$$x_i = \sum_{i \neq j} b_{ij} x_j + e_i \tag{9}$$

**Prompt template in the case of the PC algorithm**    Although the causal relationships are ultimately represented only as directed edges in Exact Search and DirectLiNGAM, the situation changes slightly when we adopt the PC algorithm along with the codes in "causal-learn." This is because the PC algorithm can also output undirected edges, when the causal direction between a pair of variables cannot be determined. Therefore, we have tentatively decided to include not only directed edges but also undirected edges in SCP. Additionally, we have prepared another prompt template for SCP in the case of PC, as shown in Table 12.

---

[23]Here, the error distribution function $e_i$ is also assumed to be non-Gaussian.

Table 10: Contents filled in ⟨blank 5⟩ shown in Table 1, when Exact Search or DirectLiNGAM is adopted for the SCD process.

| SCP Pattern | Content in ⟨blank 5⟩ |
|---|---|
| **Pattern 1**
Directed edges | All of the edges suggested by the statistical causal discovery are below:
⟨ cause 1 ⟩ → ⟨ effect 1 ⟩
⟨ cause 2 ⟩ → ⟨ effect 2 ⟩
⋮ |
| **Pattern 2**
Bootstrap probabilities
of directed edges | All of the edges with non-zero bootstrap probabilities
suggested by the statistical causal discovery are below:
⟨ cause 1 ⟩ → ⟨ effect 1 ⟩ (bootstrap probability = $P_1$)
⟨ cause 2 ⟩ → ⟨ effect 2 ⟩ (bootstrap probability = $P_2$)
⋮ |
| **Pattern 3**
(Only for DirectLiNGAM)
Non-zero causal coefficients
of directed edges | All of the edges and their coefficients of the structural causal model
suggested by the statistical causal discovery are below:
⟨ cause 1 ⟩ → ⟨ effect 1 ⟩ (coefficient = $b_1$)
⟨ cause 2 ⟩ → ⟨ effect 2 ⟩ (coefficient = $b_2$)
⋮ |
| **Pattern 4**
(Only for DirectLiNGAM)
Non-zero causal coefficients
and bootstrap probabilities
of directed edges | All of the edges with non-zero bootstrap probabilities
and their coefficients of the structural causal model
suggested by the statistical causal discovery are below:
⟨ cause 1 ⟩ → ⟨ effect 1 ⟩ (coefficient = $b_1$, bootstrap probability = $P_1$)
⟨ cause 2 ⟩ → ⟨ effect 2 ⟩ (coefficient = $b_2$, bootstrap probability = $P_2$)
⋮ |

Table 11: Contents filled in ⟨blank 6⟩ and ⟨blank 9⟩ shown in Table 1, when Exact Search or DirectLiNGAM is adopted for the SCD process.

| SCP Pattern | Case Classification | ⟨blank 6⟩ | Content in ⟨blank 9⟩ |
|---|---|---|---|
| **Pattern 1**
Directed edges | $x_j \rightarrow x_i$ emerged | a | -
(No values are filled in) |
| | $x_j \rightarrow x_i$ not emerged | no | -
(No values are filled in) |
| **Pattern 2**
Bootstrap probabilities
of directed edges | $P_{ij} \neq 0$ | a | with a bootstrap probability of $P_{ij}$ |
| | $P_{ij} = 0$ | no | -
(No values are filled in) |
| **Pattern 3**
(Only for DirectLiNGAM)
Non-zero causal coefficients
of directed edges | $b_{ij} \neq 0$ | a | with a causal coefficient of $b_{ij}$ |
| | $b_{ij} = 0$ | no | -
(No values are filled in) |
| **Pattern 4**
(Only for DirectLiNGAM)
Non-zero causal coefficients
and bootstrap probabilities
of directed edges | $P_{ij} \neq 0$ and $b_{ij} \neq 0$ | a | with a bootstrap probability of $P_{ij}$,
and the coefficient is likely to be $b_{ij}$ |
| | $P_{ij} \neq 0$ and $b_{ij} = 0$ | a | with a bootstrap probability of $P_{ij}$,
but the coefficient is likely to be 0 |
| | $P_{ij} = 0$ | no | -
(No values are filled in) |

This template is slightly modified from the one in Table 1. The description for ⟨ blank 5 ⟩ in each SCP pattern is augmented by Table 13, and the description for ⟨ blank 6 ⟩ is similarly augmented by Table 14.

As shown in Table 13, the directed and undirected edges are separately listed and clearly distinguished by the edge symbols "→" for directed edges and "—" for undirected edges. These are then filled in ⟨ blank 5 ⟩. The pairs of variables connected by undirected edges are represented as ⟨ cause or effect $i$-1 ⟩ and ⟨ cause or effect $i$-2 ⟩, and the bootstrap probability of the emergence of these relationships is represented as $P_i^u$. On the other hand, the bootstrap probability of ⟨ cause $i$ ⟩ → ⟨ effect $i$ ⟩ is represented as $P_i^d$.

The division of descriptions in ⟨ blank 6 ⟩ is shown in Table 14. The bootstrap probabilities of the appearance of $x_j \rightarrow x_i$ and $x_j$—$x_i$ are represented as $P_{ij}^d$ and $P_{ij}^u$, respectively.

Table 12: The prompt template of $Q_{ij}^{(1)}(x_i, x_j, \hat{G}_0, \boldsymbol{P^d}, \boldsymbol{P^u})$ in the case of the PC algorithm. The "blanks" enclosed with $\langle \ \rangle$ are filled with descriptive words considering the theme of the causal inference, variable names, and the SCD result with the PC algorithm.

---

**Prompt Template of $q_{ij}^{(1)} = Q_{ij}^{(1)}(x_i, x_j, \hat{G}_0, \boldsymbol{P^d}, \boldsymbol{P^u})$ for PC**

We want to carry out causal inference on $\langle$**blank 1. The theme**$\rangle$, considering $\langle$**blank 2. The description of all variables**$\rangle$ as variables.

First, we have conducted the statistical causal discovery with the PC (Peter-Clark) algorithm, using a fully standardized dataset on $\langle$**blank 4. The description of the dataset**$\rangle$.

$\langle$**blank 5. This includes the information of $\hat{G}_0$ (for Pattern 1), or $\boldsymbol{P^d}$ and $\boldsymbol{P^u}$ (for Pattern 2). The details of the contents depend on the prompting patterns, both for directed and undirected edges.**$\rangle$

According to the results shown above, it has been determined that
$\langle$**blank 6. The details of the interpretation of whether there is a causal relationship between $x_j$ and $x_i$ from the result shown in blank 5**$\rangle$.
Then, your task is to interpret this result from a domain knowledge perspective and determine whether this statistically suggested hypothesis is plausible in the context of the domain.

Please provide an explanation that leverages your expert knowledge on the causal relationship between $\langle$**blank 7. The name of $x_j$**$\rangle$ and $\langle$**blank 8. The name of $x_i$**$\rangle$, and assess the naturalness of this causal discovery result.
Your response should consider the relevant factors and provide a reasoned explanation based on your understanding of the domain.

---

Table 13: Contents filled in ⟨blank 5⟩ shown in Table 12.

| SCP Pattern | Content in ⟨blank 5⟩ |
|---|---|
| **Pattern 1**
Directed and undirected edges | All of the edges suggested by the statistical causal discovery are below:
⟨ cause 1 ⟩ → ⟨ effect 1 ⟩
⟨ cause 2 ⟩ → ⟨ effect 2 ⟩

$\vdots$

In addition to the directed edges above, all of the undirected edges
suggested by the statistical causal discovery are below:
⟨ cause or effect 1-1 ⟩ — ⟨ cause or effect 1-2 ⟩
⟨ cause or effect 2-1 ⟩ — ⟨ cause or effect 2-2 ⟩

$\vdots$ |
| **Pattern 2**
Bootstrap probabilities
of directed and undirected edges | All of the edges with non-zero bootstrap probabilities
suggested by the statistical causal discovery are below:
⟨ cause 1 ⟩ → ⟨ effect 1 ⟩ (bootstrap probability = $P_1^d$)
⟨ cause 2 ⟩ → ⟨ effect 2 ⟩ (bootstrap probability = $P_2^d$)

$\vdots$

In addition to the directed edges above, all of the undirected edges
suggested by the statistical causal discovery are below:
⟨ cause or effect 1-1 ⟩ — ⟨ cause or effect 1-1 ⟩ (bootstrap probability = $P_1^u$)
⟨ cause or effect 2-1 ⟩ — ⟨ cause or effect 2-2 ⟩ (bootstrap probability = $P_2^u$)

$\vdots$ |

Table 14: Contents filled in ⟨blank 6⟩ shown in Table 12.

| SCP Pattern | Case Classification | Content in ⟨blank 6⟩ |
|---|---|---|
| **Pattern 1**
Directed and undirected edges | $x_j \rightarrow x_i$ | there may be a direct impact of
a change in ⟨**blank 7. The name of $x_j$**⟩ on ⟨**blank 8. The name of $x_i$**⟩ |
| | $x_j — x_i$ | there may be a direct causal relationship
between ⟨**blank 7. The name of $x_j$**⟩ and ⟨**blank 8. The name of $x_i$**⟩,
although the direction has not been determined |
| | no edge between $x_i$ and $x_j$ | there may be no direct impact of
a change in ⟨**blank 7. The name of $x_j$**⟩ on ⟨**blank 8. The name of $x_i$**⟩ |
| **Pattern 2**
Bootstrap probabilities of
directed and undirected edges | $P_{ij}^d \neq 0$ and $P_{ij}^u \neq 0$ | there may be a direct impact of
a change in ⟨**blank 7. The name of $x_j$**⟩ on ⟨**blank 8. The name of $x_i$**⟩
with a bootstrap probability of $P_{ij}^d$.
In addition, it has also been shown above that
there may be a direct causal relationship
between ⟨**blank 7. The name of $x_j$**⟩ and ⟨**blank 8. The name of $x_i$**⟩
with a bootstrap probability of $P_{ij}^u$,
although the direction has not been determined |
| | $P_{ij}^d \neq 0$ and $P_{ij}^u = 0$ | there may be a direct impact of
a change in ⟨**blank 7. The name of $x_j$**⟩ on ⟨**blank 8. The name of $x_i$**⟩
with a bootstrap probability of $P_{ij}^d$ |
| | $P_{ij}^d = 0$ and $P_{ij}^u \neq 0$ | there may be a direct causal relationship
between ⟨**blank 7. The name of $x_j$**⟩ and ⟨**blank 8. The name of $x_i$**⟩
with a bootstrap probability of $P_{ij}^u$,
although the direction has not been determined |
| | $P_{ij}^d = 0$ and $P_{ij}^u = 0$ | there may be no direct impact of
a change in ⟨**blank 7. The name of $x_j$**⟩ on ⟨**blank 8. The name of $x_i$**⟩ |

## C    Details of Datasets used in Demonstrations

In this section, the details of the dataset used in the main body and the appendix are clarified. The ground truths set for the evaluation of the SCD and LLM-KBCI results for each dataset are presented.

### C.1    Auto MPG data

Auto MPG data were originally open in the UCI Machine Learning Repository (Quinlan, 1993), and used as a benchmark dataset for causal inference (Spirtes et al., 2010; Mooij et al., 2016). This dataset consists of variables related to the fuel consumption of cars. We adopt five variables: "Weight", "Displacement", "Horsepower", "Acceleration" and "Mpg" (miles per gallon). Moreover, the number of points of this dataset in the experiment is 392. The ground truth of the causal relationships we adopt in this paper is shown in Figure 7; the original has been shown as an example of the kPC algorithm (Spirtes et al., 2010). The differences from the original study (Spirtes et al., 2010) are presented below:

**(1) Loss of "Cylinders"**    Although there is also a discrete variable of "Cylinders" in the original data (Quinlan, 1993), it is omitted in the experiments to focus solely on the continuous variables.

**(2) Directed edge from "Weight" to "Displacement"**    The "Weight" and "Displacement" are connected with an undirected edge, which indicates that the direction cannot be determined in the kPC algorithm, although a causal relationship between these two variables is suggested. However, it is empirically acknowledged that large and heavy vehicles use engines with larger displacements to provide sufficient power to match their size. Thus, we temporally set the direction of the edge between these two variables as "Weight" → "Displacement."

We also recognize that another ground truth was interpreted in the process of reconstructing the Tübingen database for cause-effect pairs (Mooij et al., 2016) [24], and "Mpg" and "Acceleration" were interpreted as effected variables from other elements. This ground truth seems to be reliable, if we do not significantly discriminate between direct and indirect causal effects and target the identification of cause and effect from a pair of variables. However, we adopt the ground truth based on the result from the kPC algorithm, because our target is to approach the true causal graph, including multistep causal relationships.

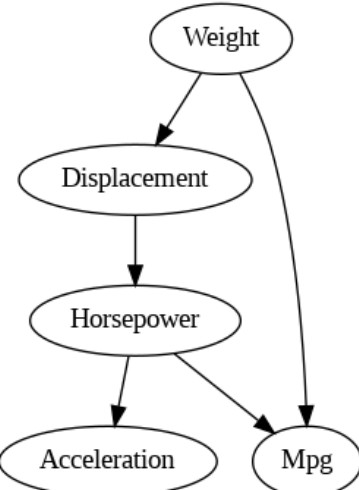

Figure 7: Causal graph of the ground truth adopted for the Auto MPG data in this study.

In Figure 8, the results of basic causal structure analysis via the PC, Exact Search, and DirectLiNGAM algorithms without prior knowledge are presented. Several reversed edges from ground truths such as "Mpg" → "Weight" are observed.

---

[24]https://webdav.tuebingen.mpg.de/cause-effect/

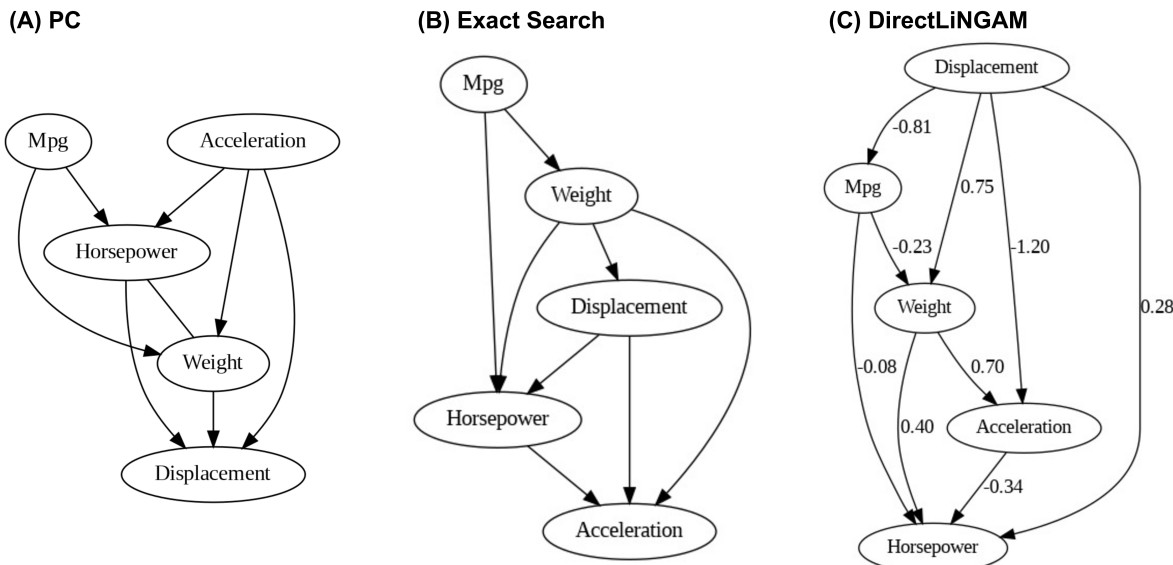

Figure 8: Results of SCD on Auto MPG data with (A) PC, (B) Exact Search, and (C) DirectLiNGAM.

## C.2    DWD climate data

The DWD climate data were originally provided by the DWD [25], and several of the original datasets were merged and reconstructed as a component of the übingen database for causal-effect pairs (Mooij et al., 2016). This dataset consists of six variables: "Altitude", "Latitude", "Longitude", "Sunshine" (duration), "Temperature" and "Precipitation". The number of points of this dataset is 349, which corresponds to the number of weather stations in Germany without missing data.

Because there is no ground truth on this dataset advocated, except for that in the übingen database for causal-effect pairs (Mooij et al., 2016), we adopt it temporally in this experiment, as shown in Figure 9.

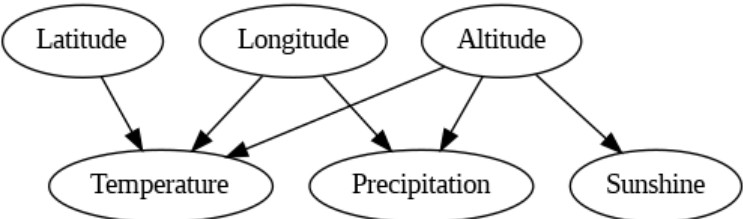

Figure 9: Causal graph of the ground truth adopted for the DWD climate data in this study.

In Figure 10, the results of basic causal structure analysis via the PC, Exact Search, and DirectLiNGAM algorithms without prior knowledge are presented. In all the causal graphs in Figure 10, several unnatural behaviors are observed, such as "Altitude" being affected by other climate variables, which we interpret as reversed causal relationships from the ground truths.

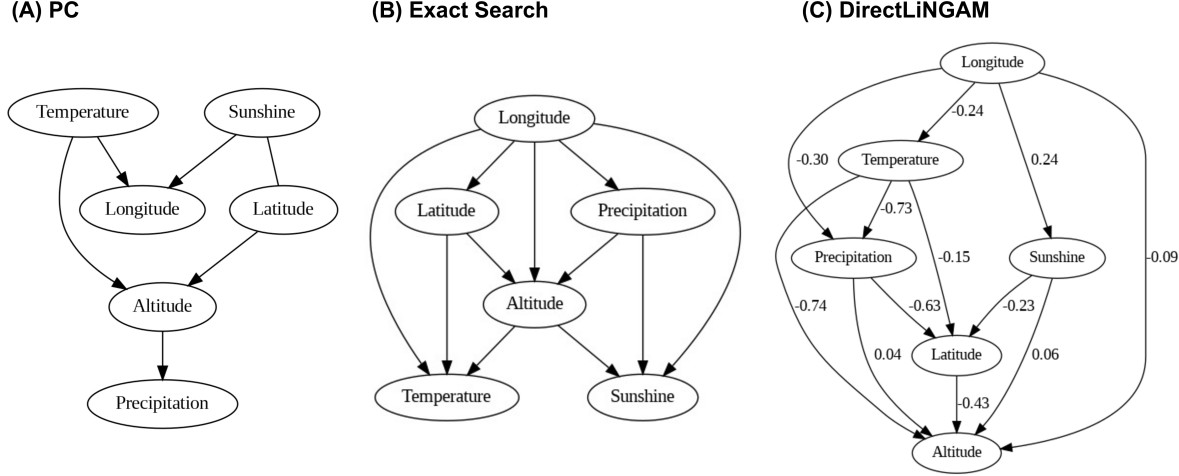

Figure 10: Results of SCD on DWD climate data with (A) PC, (B) Exact Search, and (C) DirectLiNGAM.

[25]https://www.dwd.de/

### C.3 Sachs protein data

This dataset consists of variables related to the phosphorylation levels of proteins and lipids in primary human immune cells, which were originally constructed and analyzed with the nonparametric causal structure learning algorithm by Sachs et al. (2005). It contains 11 continuous variables: "raf", "mek", "plc", "pip2", "pip3", "erk", "akt", "pka", "pkc", "p38" and "jnk". The number of points in this dataset is 7466.

The ground truth adopted in this study is almost the same as the interpretation shown in the study by Sachs et al. (2005). The differences from the causal graph visually displayed in the original paper are presented below:

**(1) Reversed edge between "pip3" and "plc"**    Although the directed edge "plc" → "pip3" was detected in the original study, it was denoted as "reversed," which may be the reversed direction from the expected edge. Thus, we adopt the causal relationship of "pip3" → "plc" which Sachs *et al.* anticipated as true from an expert point of view.

**(2) Three missed edges in the original study**    In the study by Sachs *et al.*, "pip2" → "pkc'," "plc" → "pkc," and "pip3" → "akt" did not appear in the Bayesian network inference result, although they were expected to be direct causal relationships from the domain knowledge. We adopt these three edges for the ground truth, considering that they may not appear under certain SCD conditions and assumptions.

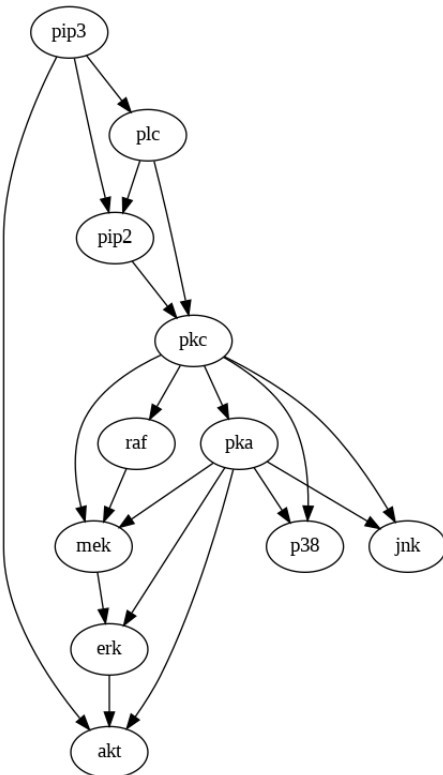

Figure 11: Causal graph of the ground truth adopted for Sachs protein data in this study.

In Figure 12, the results of the basic causal structure analysis via the PC, Exact Search, and DirectLiNGAM algorithms without prior knowledge are shown.

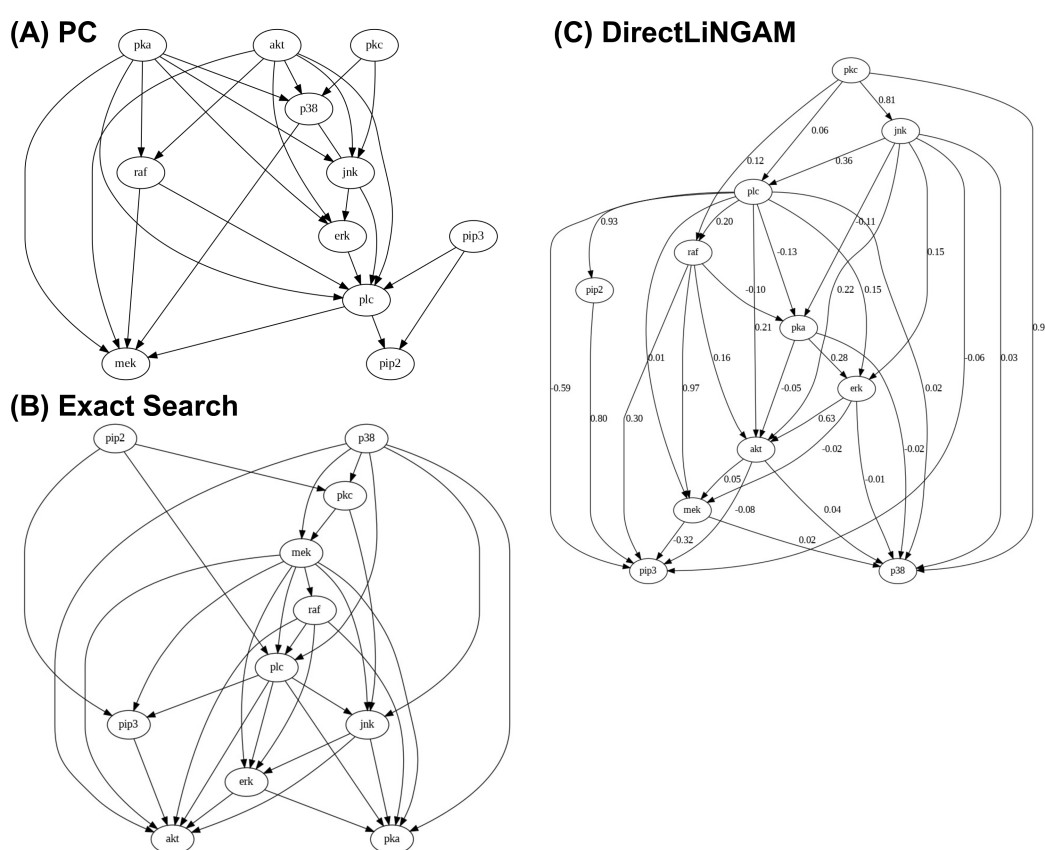

Figure 12: Results of SCD on Sachs data with (A) PC, (B) Exact Search, and (C) DirectLiNGAM.

### C.4 health-screening data (closed data and not included in GPT-4's pretraining materials)

To confirm that GPT-4 can adequately judge the existence of causal relationships with SCP, even if the dataset used in SCD is not included in the pretraining dataset of GPT-4, we have additionally prepared a health-screening dataset of workers in engineering and construction contractors, which is not disclosed because of its sensitive nature regarding personal handling and other private concerns. It contains seven continuous variables: body mass index ("BMI"), waist circumference ("Waist"), Systolic blood pressure ("SBP"), Diastolic blood pressure ("DBP"), hemoglobin A1c ("HbA1c"), low density lipoprotein cholesterol ("LDL"), and age ("Age"). The number of total points of this dataset is 123 151.

Although the causal relationships between all pairs of variables are not completely determined, we set two types of ground truths.

**(1) Directed edges interpreted as ground truths**   We empirically set four directed edges below as ground truths.

- "Age"→"BMI"(Clarke et al., 2008; Alley et al., 2008; Gordon-Larsen et al., 2010; Yang et al., 2021)

- at least one of "Age"→"SBP" and "Age"→"DBP"(Gurven et al., 2012)

- "Age"→"HbA1c"(Pani et al., 2008; RaviKumar et al., 2011; Dubowitz et al., 2014)

**(2) Variable interpreted as a parent for all other variables**   "Age" is an unmodifiable background factor. Furthermore, numerous medical studies have clearly demonstrated that aging affects "BMI," "SBP," "DBP," and "HBA1c." On the basis of this specialized knowledge, we interpret "Age" as a parent for all other variables.

The ground truths introduced above also appear in the result of DirectLiNGAM without prior knowledge, as shown in Figure 13. Although "Age"→"HbA1c" is confirmed in this result, the causal coefficient of this edge is relatively small. Thus, depending on the number of data points or the bias of the dataset, it is possible that this edge does not appear in all SCD methods without prior knowledge. For the experiment, to confirm that GPT-4 can supply SCD with adequate prior knowledge, even if a direct edge of the ground truth is not apparent, we repeated the sampling of 1000 points from the entire dataset, until we obtained a subset on which PC, Exact Search, and DirectLiNGAM cannot discern the causal relationship "Age"→"HbA1c" without prior knowledge.

The results of the SCD on the subset are shown in Figure 14, and this subset is adopted to confirm the effectiveness of the proposed method. All the SCD results confirm that "Age" → "HbA1c" does not appear, and "Age" is directly influenced by other variables, which we interpret as an unnatural behavior from the domain knowledge.

**(3) Limitations in the strict proof of the absence of any overlap with the pretraining data of GPT-4**   As explained in Section 4.2, we conclude that our unpublished health-screening dataset was not included in the pretraining data of GPT-4. This conclusion is based on circumstantial evidence such as the following:

- the original dataset consists of personal health-screening data from Japanese health insurance records

- the legal restrictions in Japan prohibiting the public use of the raw data

- our past usage, which does not involve direct exposure to the Internet or sharing with third parties, and has complied with the scope of data use agreed upon by the data providers and approved by the institute's ethics review board

For more rigorous validation, it is ideally recommended to technically confirm whether there is any overlap between our dataset and the pretraining data of GPT-4. Indeed, various methods have been proposed to detect potential data leakage into pretraining datasets of LLMs, or to estimate whether specific data are

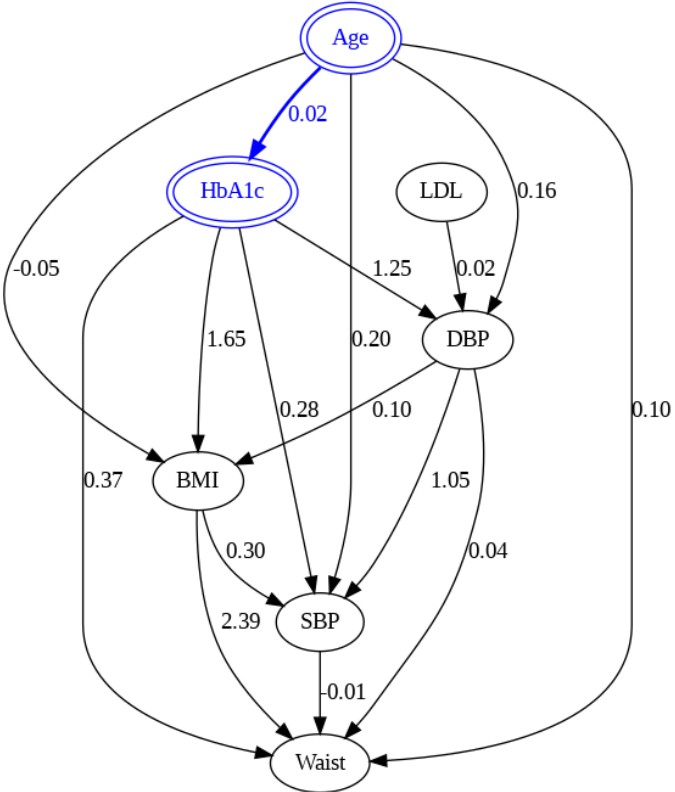

Figure 13: Causal graph suggested by DirectLiNGAM using full points of the health-screening data.

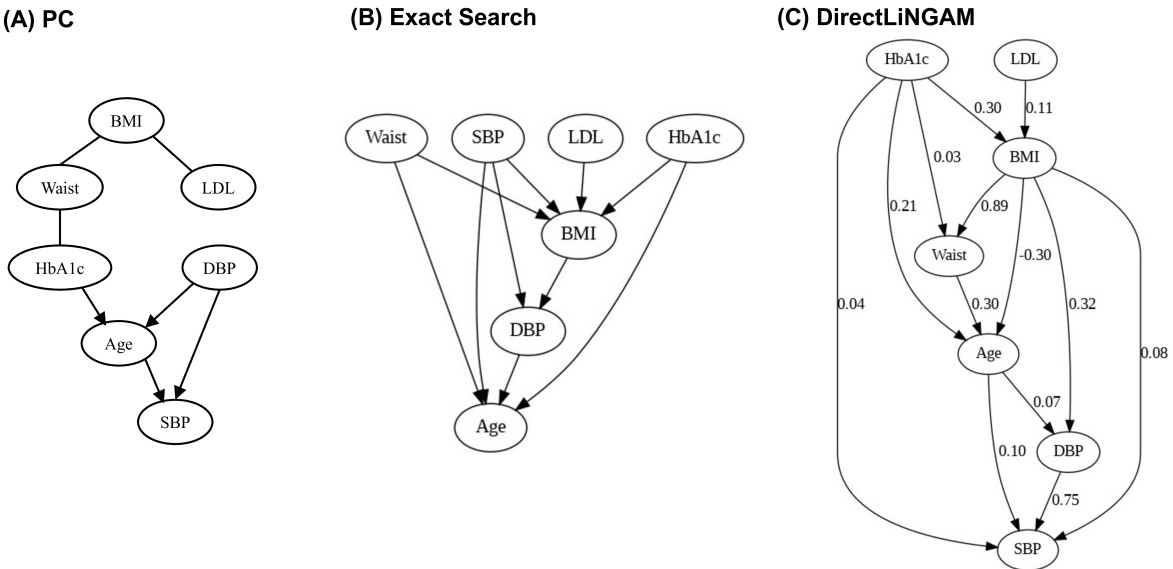

Figure 14: Results of SCD on the randomly selected subsample in the health-screening dataset with (A) a PC, (B) Exact Search, and (C) DirectLiNGAM.

included in them, such as $n$-gram matching (Singh et al., 2024), membership inference attacks (MIAs) (Duan et al., 2024), and the dataset inference method (Maini et al., 2024). However, there are still both technical and

ethical limitations in the technical confirmation of the absence of any overlap between our health-screening dataset and the pretraining data of GPT-4.

First, since neither the model nor the pretraining dataset of GPT-4 has been disclosed, it is impossible to verify any overlap with our health-screening dataset via techniques such as $n$-gram matching. Moreover, it must be carefully assessed whether applying other verification methods to our actual dataset, could itself pose a risk of data leakage, which would raise social and ethical concerns. Ensuring a sufficiently robust and secure environment for such validation would also require an additional layer of safeguards. In addition, even if these technical concerns were addressed, further issues remain regarding the scope of data use as agreed upon by the data providers. It would also be necessary to confirm compliance with legal restrictions, particularly concerning the use of personal records contained in our dataset.

Although the above challenges are highly relevant to the practical analysis of datasets containing confidential information via LLMs, addressing them requires thorough discussion of legal restrictions as well as technical demonstrations under carefully prepared conditions –both of which are far beyond the scope and the aim of our work. Consequently, from a practical standpoint, it is reasonable to conclude that our health-screening dataset was not included in the pretraining data of GPT-4, based on circumstantial evidence.

# D  Token Generation Probabilities as Confidence of LLMs and Sensitivity Analysis

In this section, we provide a supplemental explanation of the properties of the mean probability $\overline{p_{ij}}$, that the LLM generates the token "yes" as the response to the prompt asking for confidence in the causal relationship from $x_j$ to $x_i$, highlighting the properties of the token generation process in Transformer-based LLMs. We also discuss the mean probability $\overline{r_{ij}}$ that the LLM generates the token "no," and explain the symmetric behavior of $\overline{p_{ij}}$ and $\overline{r_{ij}}$ arising from the empirical relationship $\overline{p_{ij}} + \overline{r_{ij}} = 1$, under the assumption of the "faithfulness" of the LLM to the prompt. Moreover, we quantitatively evaluate the fluctuation of $\overline{p_{ij}}$ with the standard error $SE(p_{ij})$ as indices of the sensitivity or reliability of these probabilities measured in our experiments, through regression analysis with a phenomenological distribution function $SE(\overline{p_{ij}}) = a_p - b_p(\overline{p_{ij}} - 0.5)^2$. Finally, the precision of the classification of forbidden causal paths is evaluated through numerical simulations.

The discussion in this section not only contributes to the evaluation of the LLM's confidence in the causal relationship between pairs of variables, but also supports the reliability of the LLM outputs.

## D.1  Details of Token Generation Probability Calculations

As explained in Section 3.1, we quantitatively evaluate the generation probability of the token "yes" from GPT-4, as the confidence of GPT-4 regarding the existence of the causal relationship. Although the details of GPT-4's architecture have not been disclosed, it is naturally assumed that GPT-4 employs a temperature-adjusted softmax function for token generation, considering the following facts:

- The previous GPT models developed by OpenAI are based on the Transformer architecture (Radford et al., 2019; Brown et al., 2020)

- The temperature hyperparameter can be optionally set

- The log-probabilities of the most likely tokens at each token position can be obtained

In the process of text generation in a Transformer, the model first calculates the logit $z_k$ for token $c_k$ ($0 \leq k \leq K$, where $K$ is the number of candidate tokens) as a likelihood score for all candidate tokens on the basis of the prompted text $X$. Then, the conditional probability distribution of the emergence of token $c_a$ with the logit $z_a$, $P(c_a|X)$, is calculated through the temperature-adjusted softmax function as follows (Tunstall et al., 2022):

$$P(c_a|X,T) = \frac{\exp(z_a/T)}{\sum_{k=0}^{K} \exp(z_k/T)} \tag{10}$$

Here, $T$ is a temperature hyperparameter that adjusts the conditional probability distribution. Finally, the next token is sampled based on the conditional probability distribution above.

In this work, $T$ was fixed to 0.7. In the third step, the log-probability $L_{ij}^{(m)} = L^{(m)}(\epsilon_{\text{GPT4}}(q_{ij}^{(2)}) = \text{"yes"})$ shown in Algorithm 1 is directly sampled $M$ times via the optional function of GPT-4. Considering that the token generation probability is represented by Eq. (10), we have calculated the mean probability $\overline{p_{ij}}$ as follows:

$$\overline{p_{ij}} = \frac{\sum_{m=1}^{M} \exp(L_{ij}^{(m)})}{M} = \frac{\sum_{m=1}^{M} P^{(m)}\left(\text{"yes"}|q_{ij}^{(2)}, T\right)}{M} \tag{11}$$

We have interpreted $\overline{p_{ij}}$ calculated above as the confidence[26] of GPT-4 in the existence of the causal effect from $x_j$ on $x_i$.

---

[26] Although it is suggested that this interpretation is valid in common-sense reasoning from our experiments, notably, this probability simply represents the likelihood that the LLM selected the expected token as the answer to the prompt. In this sense, this "confidence" can be biased under some conditions since it depends on the architecture and the pretraining datasets of the LLM. Therefore, as discussed in Section 5, it is necessary to further improve our method to avoid hallucinations more reliably, especially for applications in highly specialized academic domains.

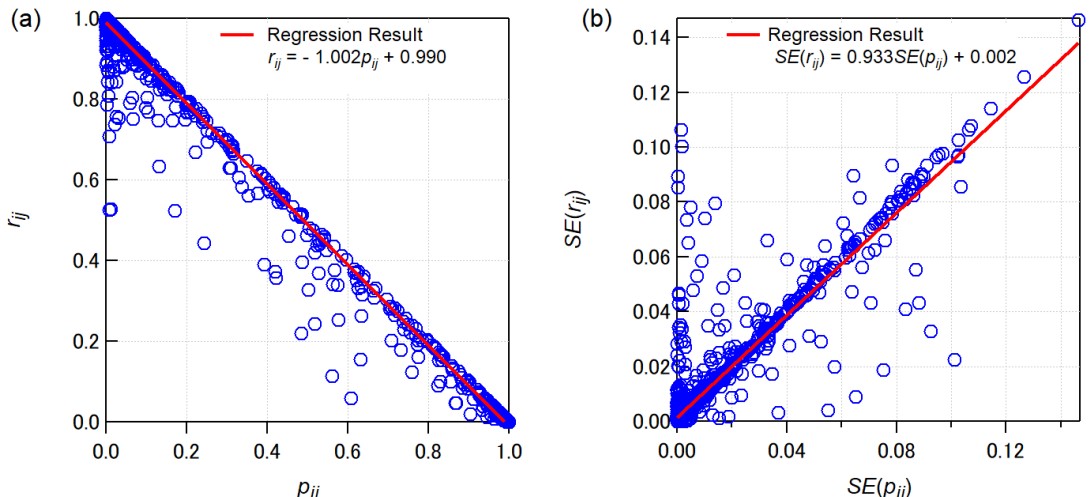

Figure 15: (a) Relationship between $\overline{p_{ij}}$ and $\overline{r_{ij}}$. The red line is the result of least squares regression with $\overline{r_{ij}} = \alpha \overline{p_{ij}} + \beta$. The estimated values ($\pm$ the standard errors) of the coefficients are $\alpha = -1.002 \pm 0.002 (\simeq -1)$ and $\beta = 0.990 \pm 0.001 (\simeq 1)$. (b) Relationship between $SE(\overline{p_{ij}})$ and $SE(\overline{r_{ij}})$. The red line is the result of least squares regression with $SE(\overline{r_{ij}}) = \gamma SE(\overline{p_{ij}}) + \delta$. The estimated values ($\pm$ the standard errors) of the coefficients are $\gamma = 0.933 \pm 0.011 (\simeq 1)$ and $\delta = 0.0015 \pm 0.0002 (\simeq 0)$.

### D.2 Confidence in the Absence of the Causal Relationship

Although the generation probability of the token "no" is not evaluated in the main text, it is valuable to discuss the ideal behavior of this probability for the subsequent discussion of the stability of $\overline{p_{ij}}$. In the same context as $\overline{p_{ij}}$, the mean probability of generating the token "no", $\overline{r_{ij}}$, is calculated as follows:

$$\overline{r_{ij}} = \frac{\sum_{m=1}^{M} P^{(m)}\left(\text{"no"}|q_{ij}^{(2)}, T\right)}{M} \tag{12}$$

Since GPT-4 is required to answer only with "yes" or "no," under the assumption that the response from GPT-4 is faithful to the prompt explained in Table 2, it is expected that $P^{(m)}(\text{"yes"}|q_{ij}^{(2)}, T) + P^{(m)}(\text{"no"}|q_{ij}^{(2)}, T) = 1$. In this section, we define this property as "faithfulness." Therefore, from Eqs. (11) and (12), the ideal relationship between $\overline{p_{ij}}$ and $\overline{r_{ij}}$ is expressed as follows:

$$\overline{r_{ij}} = -\overline{p_{ij}} + 1 \tag{13}$$

Since Eq. (13) is also expressed in a symmetrical form such as $\overline{r_{ij}} - 0.5 = -(\overline{p_{ij}} - 0.5)$, this property is expected to contribute to a clearer quantitative interpretation of the stability of $\overline{p_{ij}}$. This is particularly important for validating the latter discussion on the fluctuation of $\overline{p_{ij}}$.

### D.3 Stability of Token Generation Probabilities

For the quantitative evaluation of the stability of $\overline{p_{ij}}$, we compare $\overline{p_{ij}}$ and $\overline{r_{ij}}$, along with their standard errors ($SE(p_{ij})$ and $SE(r_{ij})$) estimated through trials conducted $M = 5$ times. We temporarily concatenate the results of LLM-KBCI on the datasets of AutoMPG, DWD, and Sachs (for all the prompting patterns and for all the SCD algorithms), and on the health-screening data (for all the prompting patterns with

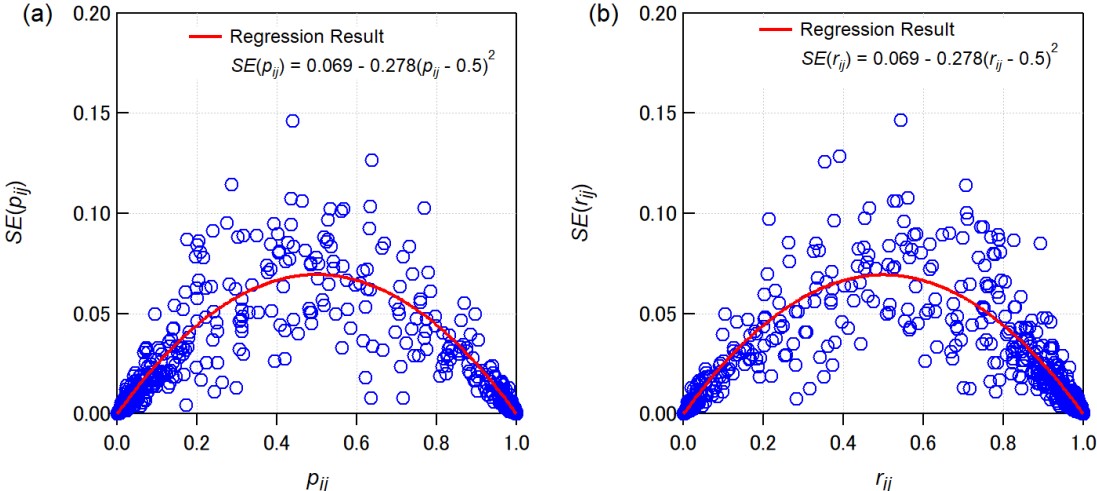

Figure 16: (a) Relationship between the mean probability $\overline{p_{ij}}$ and the standard error $SE(p_{ij})$. The red line is the result of least squares regression with $SE(\overline{p_{ij}}) = a_p - b_p(\overline{p_{ij}} - 0.5)^2$. The estimated values ($\pm$ standard errors) of the coefficients are $a_p = 0.0694 \pm 0.0007$ and $b_p = 0.2783 \pm 0.0030$. (b) Relationship between the mean probability $\overline{r_{ij}}$ and the standard error $SE(r_{ij})$. The red line is the result of least squares regression with $SE(\overline{r_{ij}}) = a_r - b_r(\overline{r_{ij}} - 0.5)^2$. The estimated values ($\pm$ standard errors) of the coefficients are $a_r = 0.0692 \pm 0.0007$ and $b_r = 0.2780 \pm 0.0032$.

DirectLiNGAM), under the assumption that the relations among $\overline{p_{ij}}$, $\overline{r_{ij}}$, $SE(p_{ij})$ and $SE(q_{ij})$ are almost independent of the domains. This holds even if the SCP is conducted within the realm of common-sense reasoning. The number of points of the concatenated data is 1650.

Figure 15 (a) shows the relation between $\overline{p_{ij}}$ and $\overline{r_{ij}}$, and the red line indicates the result of least squares regression with $\overline{r_{ij}} = \alpha\overline{p_{ij}} + \beta$. Considering that the estimated values of the causal coefficients ($\alpha = -1.002$, $\beta = 0.990$) exhibit similar behavior to Eq. (13), it can be inferred that the assumption of the "faithfulness" of the LLM generation to the prompts is valid in our experiments. Therefore, it seems approximately valid to assume the symmetrical relation $\overline{r_{ij}} - 0.5 = -(\overline{p_{ij}} - 0.5)$.

Moreover, Figure 15 (b) shows the relationship between $SE(\overline{p_{ij}})$ and $SE(\overline{r_{ij}})$, where the red line indicates the result of least squares regression with $SE(\overline{r_{ij}}) = \gamma SE(\overline{p_{ij}}) + \delta$. Given the estimated values of the causal coefficients ($\gamma = 0.933$, $\delta = 0.002$), it can be inferred that the approximation $SE(\overline{p_{ij}}) \simeq SE(\overline{r_{ij}})$ holds true. This fact provides further evidence for the approximately symmetric behavior expressed as $\overline{r_{ij}} - 0.5 = -(\overline{p_{ij}} - 0.5)$.

Figure 16 shows the relationship between $\overline{p_{ij}}$ and $SE(p_{ij})$ in (a) and the relationship between $\overline{r_{ij}}$ and $SE(r_{ij})$ in (b). These two graphs exhibit mutually related characteristics as follows:

- In the limit of $\overline{p_{ij}} \to 1$ or $\overline{r_{ij}} \to 0$, both $SE(p_{ij})$ and $SE(r_{ij})$ approach 0, indicating that $\overline{p_{ij}}$ and $\overline{r_{ij}}$ become highly stable when confidence in the presence of the causal effect from $x_j$ to $x_i$ is very high.

- In the limit of $\overline{p_{ij}} \to 0$ or $\overline{r_{ij}} \to 1$, both $SE(p_{ij})$ and $SE(r_{ij})$ approach 0, indicating that $\overline{p_{ij}}$ and $\overline{r_{ij}}$ become highly stable when confidence in the absence of the causal effect from $x_j$ to $x_i$ is very high.

Furthermore, the entire form of the $\overline{p_{ij}}$ - $SE(p_{ij})$ distribution is quantitatively similar to the $\overline{r_{ij}}$ - $SE(r_{ij})$ distribution. From this fact, it is expected that the functional form of $SE(p_{ij})$ explained with $p_{ij}$ is

approximately the same as that of $SE(r_{ij})$ explained with $r_{ij}$. Therefore, to interpret the stability of the probability easily and clearly, we assume a common distribution function between the $\overline{p_{ij}}$ - $SE(p_{ij})$ and $\overline{r_{ij}}$ - $SE(r_{ij})$ distributions, as well as the symmetric behavior shown in Eq. (13), and regress the $\overline{p_{ij}}$ - $SE(p_{ij})$ distribution with a phenomenological function as follows:

$$SE(p_{ij}) = a_p - b_p(\overline{p_{ij}} - 0.5)^2 \tag{14}$$

In the same context, the $\overline{r_{ij}}$ - $SE(r_{ij})$ distribution is regressed with a phenomenological function as follows:

$$SE(r_{ij}) = a_r - b_r(\overline{r_{ij}} - 0.5)^2 \tag{15}$$

Ideally, if the symmetrical behavior holds true, the relationship $(a_p, b_p) = (a_r, b_r)$ is expected. From the analysis results of the least squares regression, the coefficients of Eqs. (14) and (15) are estimated as $(a_p, b_p) = (0.0694, 0.2783)$ and $(a_r, b_r) = (0.0692, 0.2780)$. Thus, it is confirmed that $(a_p, b_p) \simeq (a_r, b_r)$ holds true.

From the results above, the standard error of $p_{ij}$ becomes the maximum value (expected to be 0.062) when $\overline{p_{ij}} = 0.5$. In our experiments, the fluctuation becomes critical at the criteria $\alpha_1$ and $\alpha_2$ explained in Section 3.1. From the phenomenological analysis shown above, the expected standard errors of $\overline{p_{ij}}$ at the criteria ($\alpha_1 = 0.05$ and $\alpha_2 = 0.95$ in our experiments) are estimated through Eq. (14) to be approximately 0.013. Therefore, when $\overline{p_{ij}}$ is near the threshold of 0.05 or 0.95, the probability fluctuation around this standard error can lead to a different $\boldsymbol{PK}$ matrix.

### D.4 Effect on the Decision of the Prior Knowledge Matrix

For a more detailed analysis of the effect of the fluctuation on the determination of $\boldsymbol{PK}$, we focus on two indices of sensitivity: (A) the statistically estimated probability that the true value of $p_{ij}$ ($p_{ij}^{true}$) satisfies the inequality $p_{ij}^{true} < \alpha_1$, and (B) the area under the curve (AUC) calculated through a Monte Carlo simulation.

#### (A) Statistically Estimated Probability of True Confidence

For a simple simulation, we assume that $SE(p_{ij})$ is calculated with Eq. (14) and that $(a_p, b_p) = (0.0694, 0.2783)$, as estimated through the regression analysis in Appendix D.3. Moreover, we assume that the distribution of $p_{ij}^{true}$ is expressed with a Gaussian distribution, with mean value $\overline{p_{ij}}$ and standard deviation $SE(\overline{p_{ij}})$, as described in Eq. (16).

$$f_{p_{ij}^{true}}(\overline{p_{ij}}, SE(\overline{p_{ij}}), p) = \frac{1}{\sqrt{2\pi}SE(\overline{p_{ij}})} \exp\left(-\frac{(p - \overline{p_{ij}})^2}{2SE(\overline{p_{ij}})^2}\right) \tag{16}$$

Under the assumptions above, $P(p_{ij}^{true} < \alpha_1)$, the probability that the true confidence probability $p_{ij}^{true}$ is below the forbidden edge threshold $\alpha_1$, is calculated as follows:

$$P(p_{ij}^{true} < \alpha_1) = \int_0^{\alpha_1} dp f_{p_{ij}^{true}}(\overline{p_{ij}}, SE(\overline{p_{ij}}), p) \tag{17}$$

Then, $P(p_{ij}^{true} < \alpha_1)$ is numerically calculated via Eq. (17), and the result is shown in Fig. 17 (a). When $\overline{p_{ij}} \leq 0.04$, $p_{ij}^{true}$ is almost certainly below the threshold $\alpha_1 = 0.05$, and when $\overline{p_{ij}} \geq 0.08$, $p_{ij}^{true}$ is almost certainly above 0.05. However, between $0.04 \leq \overline{p_{ij}} \leq 0.08$ (approximately 0.05), $P(p_{ij}^{true} < \alpha_1)$ decreases with increasing $\overline{p_{ij}}$. Although it is confirmed that the width of the region of this decreasing $P(p_{ij}^{true} < \alpha_1)$ is similar in the order of $SE(\overline{p_{ij}}) \simeq 0.013$ around $\overline{p_{ij}} = 0.05$, the slope of the decrease is gentler in the region of $\overline{p_{ij}} \geq 0.05$ than in the region of $\overline{p_{ij}} \leq 0.05$. This asymmetric property originates from the phenomenological nonlinear function of $SE(\overline{p_{ij}})$ as described in Eq. (14). If the threshold of the forbidden edge $\alpha_1$ is decreased, it is possible to shrink the region of decreasing $P(p_{ij}^{true} < \alpha_1)$ and the asymmetric slope. However, although we tentatively set $\alpha_1$ to 0.05, the optimal threshold is expected to be discussed in future research from a practical point of view. Notably, a lower $\alpha_1$ leads to a $\boldsymbol{PK}$ matrix, which includes more uncertain elements from domain expert knowledge and can be less effective in the fourth step in Fig. 1 than expected.

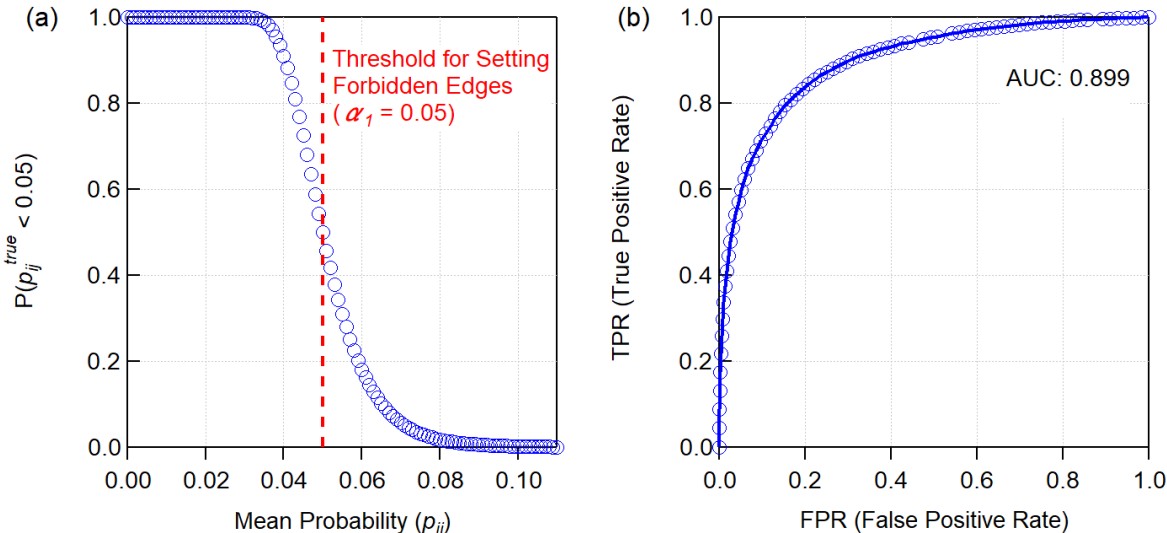

Figure 17: (a) Relationship between the calculated $P(p_{ij}^{true} < \alpha_1)$ and $\overline{p_{ij}}$ based on Eq. (17). (b) ROC curve estimated through a Monte Carlo simulation based on Eq. (16). The AUC calculated with this ROC curve is 0.899.

## (B) ROC and AUC Estimated through a Monte Carlo Simulation

Because $p_{ij}^{true}$ cannot be known in our experiments, it is impossible to construct the receiver operating characteristic (ROC) curve directly with the probability data obtained in our experiments. Instead, we have conducted a Monte Carlo simulation, through which $p_{ij}^{true}$ is randomly generated via the distribution described in Eq. (16). Furthermore, for the calculation of the ROC and AUC, we define the conditions for true positive (TP), false positive (FP), true negative (TN), and false negative (FN), in the context of judging whether the causal path from $x_j$ to $x_i$ is forbidden, as follows:

- TP: generated $p_{ij}^{true}$ satisfies $p_{ij}^{true} < \alpha_1$, and experimentally fixed $\overline{p_{ij}}$ satisfies $\overline{p_{ij}} < \alpha_1$.

- FP: generated $p_{ij}^{true}$ satisfies $p_{ij}^{true} \geq \alpha_1$, and experimentally fixed $\overline{p_{ij}}$ satisfies $\overline{p_{ij}} < \alpha_1$.

- TN: generated $p_{ij}^{true}$ satisfies $p_{ij}^{true} \geq \alpha_1$, and experimentally fixed $\overline{p_{ij}}$ satisfies $\overline{p_{ij}} \geq \alpha_1$.

- FN: generated $p_{ij}^{true}$ satisfies $p_{ij}^{true} < \alpha_1$, and experimentally fixed $\overline{p_{ij}}$ satisfies $\overline{p_{ij}} \geq \alpha_1$.

The practical Monte Carlo simulation was conducted 1000 times for each $\overline{p_{ij}}$ value, varying $\overline{p_{ij}}$ from 0.032 to $0.108^{27}$ in increments of 0.001, and the calculated ROC through this simulation is shown in Fig. 17. The AUC is estimated to be 0.899, indicating excellent discrimination of the forbidden direct causal path (Hosmer & Lemeshow, 2000). From this analysis, it is inferred that in the realm of common-sense reasoning, the decision process of **PK** from the confidence of GPT-4 at the third step in Fig. 1 according to the interpretation obtained in the second step, is statistically valid.

---

[27]In this region, $P(p_{ij}^{true} < \alpha_1)$ satisfies $0.001 < P(p_{ij}^{true} < \alpha_1) < 0.999$. Therefore, for the practical calculation of the ROC and AUC, we assume that when $\overline{p_{ij}}$ satisfies $\overline{p_{ij}} < 0.032$, $p_{ij}^{true}$ is under $\alpha_1$ and is always classified as TP, and that when $\overline{p_{ij}}$ satisfies $\overline{p_{ij}} > 0.108$, $p_{ij}^{true}$ is $\geq \alpha_1$ and is always classified as TN.

### D.5 Stability of SCD Results with $PK$

Although it is demonstrated that SCD results with $PK$ are generated through LLM processes in Section 4.1, considering the stability of $PK$ discussed in Appendix D.4, it is also important to evaluate the stability of the SCD results with $PK$ against fluctuations in the LLM's confidence probability matrix, $\overline{p}$.

For the supplemental experiment confirming the problem above, we first randomly generate the $\overline{p}$, assuming that each component is independently determined with the distribution function expressed in Eq. (16), repeated $N_{\text{sampling}}$ times for a sufficiently large $N_{\text{sampling}}$. Then, we conduct the transformation from $\overline{p}$ to $PK$ and perform SCD analysis with $PK$ for all the generated $\overline{p}$. Finally, for all the patterns of pairs of variables, we statistically analyze the $N_{\text{sampling}}$ results of the SCD outcomes and evaluate the stability of causal edge emergence or causal coefficients (in the case of DirectLiNGAM). Through this process, it is possible to assess the stability of the results of the SCD with $PK$ against the fluctuation of the LLM's confidence probability matrix.

We set $N_{\text{sampling}} = 10{,}000$, and for the simulation from real-world data, we adopt DWD climate data. To evaluate the stability of both the edge emergence and the causal coefficients for all the pairs of variables, we adopt DirectLiNGAM as the SCD algorithm. We also select the $\overline{p}$ generated through Pattern 2 (prompting with causal bootstrap probabilities), since it realizes the best improvement in the LiNGAM results, especially in precision, as shown in Table 3. The measured $\overline{p}$ and standard error matrix $SE$ obtained through $M = 5$ single-shot measurements are as follows:

$$\overline{p} = \begin{pmatrix} 0.000 & 0.000 & 0.000 & 0.000 & 0.000 & 0.000 \\ 0.997 & 0.000 & 0.007 & 0.000 & 0.999 & 0.989 \\ 0.739 & 0.999 & 0.000 & 0.000 & 0.000 & 0.384 \\ 0.000 & 0.000 & 0.000 & 0.000 & 0.000 & 0.000 \\ 0.874 & 0.010 & 0.976 & 0.000 & 0.000 & 0.981 \\ 0.000 & 0.000 & 0.000 & 0.000 & 0.000 & 0.000 \end{pmatrix} \tag{18}$$

$$SE = \begin{pmatrix} 0.000 & 0.000 & 0.000 & 0.000 & 0.000 & 0.000 \\ 0.001 & 0.000 & 0.006 & 0.000 & 0.000 & 0.004 \\ 0.140 & 0.000 & 0.000 & 0.000 & 0.000 & 0.144 \\ 0.000 & 0.000 & 0.000 & 0.000 & 0.000 & 0.000 \\ 0.066 & 0.002 & 0.011 & 0.000 & 0.000 & 0.026 \\ 0.000 & 0.000 & 0.000 & 0.000 & 0.000 & 0.000 \end{pmatrix} \tag{19}$$

The histograms of causal coefficients for all the randomly regenerated $\overline{p}$ across all pairs of variables are summarized in Table 18. The causal coefficients clearly exhibit minimal fluctuations, with each histogram resembling a delta function. This observation suggests that fluctuations in the probability matrix generated by the LLM do not degrade the stability of the SCD results.

There are two primary reasons for the stability of the SCD results with $PK$:

- **Robustness in the extreme values of LLM confidence probabilities:** As discussed in Section D.3, the standard error decreases when confidence probabilities approach 0 or 1. This behavior may contribute to the stability of $PK$.

- **Numerical datasets as the main factor in structure determination:** In DirectLiNGAM, causal coefficients are determined through regression analysis with LASSO on the dataset. This suggests that the numerical properties of the dataset play a significant role in structure determination.

### D.6 Limitations in the Sensitivity Analysis of Token Generation Probabilities

Although we have discussed the sensitivity of the token generation probabilities as the confidence of GPT-4 above, there are several limitations in elucidating the entire properties of the fluctuations induced by the

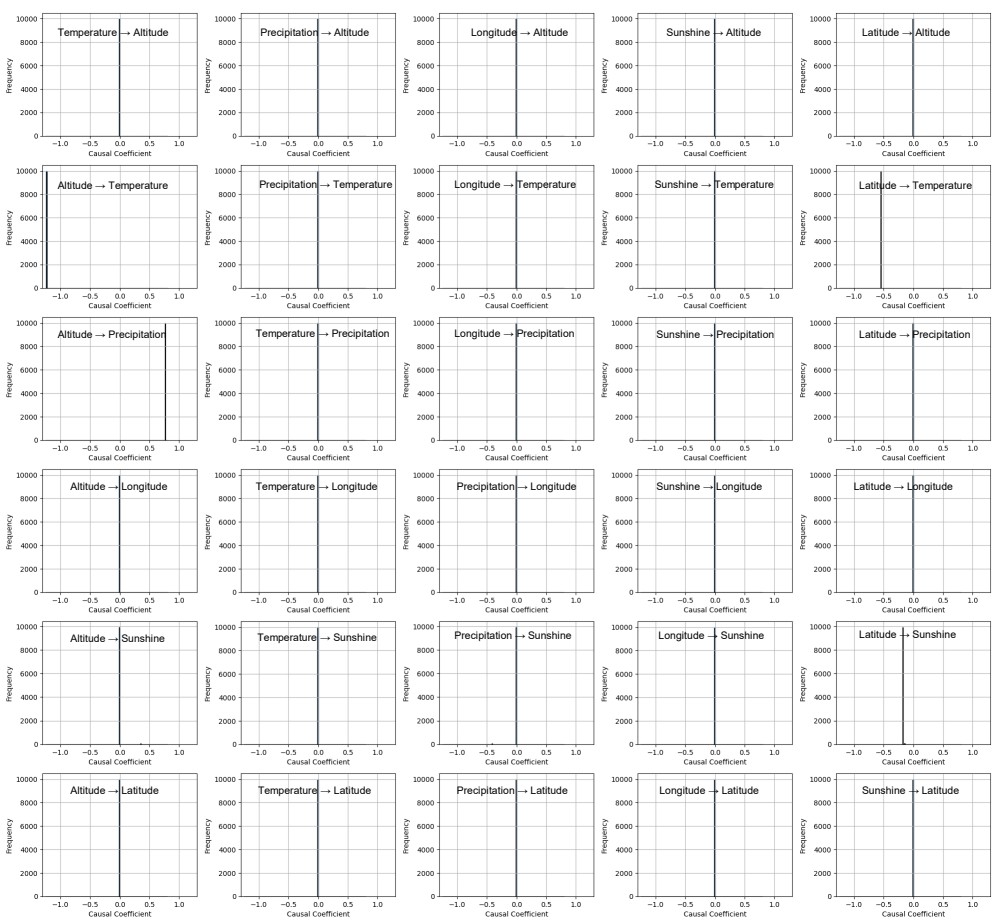

Figure 18: Histograms for all pairs of variables obtained from the DirectLiNGAM analysis of $N_{\text{sampling}} = 10,000$ randomly regenerated probability matrices for DWD climate data.

randomness of the LLMs. For further research to improve our methods, we introduce several additional points related to the measurement stability in the third step in Fig. 1.

### Sensitivity in the Explanation Obtained in Step 2

Although we have evaluated the probability fluctuation emergent in Step 3 of Fig. 1, $\overline{p_{ij}}$ and $SE(\overline{p_{ij}})$ can also depend on $a_{ij}$, the output in Step 2, on the basis of the relationship of $q_{ij}^{(2)} = Q_{ij}^{(2)}(q_{ij}^{(1)}, a_{ij})$. Because it is practically impossible to evaluate the fluctuation of $a_{ij}$ at Step 2 quantitatively, we have discussed the token generation probability and its sensitivity with a fixed $a_{ij}$.

Furthermore, we have not discussed in this work the validity of $a_{ij}$, as this is expected to be debated within the domain of research specific to LLMs, which is outside the scope of our study. However, although our method has proven to be effective in many cases through this research, it may be negatively impacted if an LLM makes biased judgments. Therefore, ongoing efforts to eliminate biases in LLMs and prevent hallucinations are required. Additionally, considering the future applications of this method, careful consideration and judgment will continue to be necessary.

### Imperfect Satisfaction of the Condition of "Faithfulness"

In our work, "faithfulness" is almost confirmed, as shown in Fig. (15). However, several points obviously deviate from the regression line, or from Eq. (13). Specifically, out of a total of 1650 points in the case of our analysis in this section, approximately 1400 points satisfy the relationship $0.99 < \overline{p_{ij}} + \overline{r_{ij}} < 1$. In contrast, for approximately 80 points, $\overline{p_{ij}} + \overline{r_{ij}} < 0.95$, indicating clear violations of "faithfulness."

The break of this "faithfulness" means that there is a generation probability of the token that is neither "yes" nor "no," at a level that cannot be ignored. One possible explanation for this case is that GPT-4 cannot completely assert "yes" or "no" on the existence of the causal effect from $x_j$ to $x_i$ with the knowledge obtained in Step 2. In more anthropomorphic terms, this could suggest that GPT-4 sometimes struggles with its decision-making.

For further rigid handling in the application of our method, it is required to prevent the break of "faithfulness," or to extend our quantitative handling of the token generation probability to the one that permits the possibility of this behavior.

### Temperature Dependence of the Fluctuation and "Faithfulness"

As described in Eq. (10), the probability distribution depends on the temperature parameter. Because a higher temperature leads to a higher generation probability of the token, unexpected at lower temperatures, it is possible that this parameter can also affect the fluctuations of $p_{ij}$ and "faithfulness." Although we have fixed this parameter at 0.7 in all the experiments in this work, further discussion on the optimal temperature for this task is expected.

# E Composition of Adjacency Matrices Representing Causal Structure and Evaluation

## E.1 Composition of Prior Knowledge Matrices

As shown in Algorithm 2, the composition rule of $PK$ depends on the type of SCD method adopted, and the decision criteria of forced and forbidden edges are tentatively set at 0.95 and 0.05, respectively. Although PC and DirectLiNGAM can be augmented with constraints for both forced and forbidden directed edges or paths, Exact Search can only be augmented with constraints for forbidden directed edges. In this section, the composition rule for $PK$ is described in detail for all the SCD algorithms we have adopted in this work.

**For PC** For the matrix representation of $PK$, the values of the matrix elements are determined as follows:

- Case 1. If $x_i \rightarrow x_j$ is forced (i.e., $p_{ij} \geq 0.95$), then $PK_{ij}$ is set to 1.

- Case 2. If $x_i \rightarrow x_j$ is forbidden (i.e., $p_{ij} < 0.05$), then $PK_{ij}$ is set to 0.

- Case 3. If the existence of $x_i \rightarrow x_j$ cannot be determined immediately from the domain knowledge generated by GPT-4 (i.e., $0.05 \leq p_{ij} < 0.95$), then $PK_{ij}$ is set to $-1$.

This ternary matrix composition is based on the constraints of the prior knowledge matrix DirectLiNGAM, which will be explained later, to apply the generated $PK$ to DirectLiNGAM as quickly as possible. Although prior knowledge is represented as a matrix in the PC algorithm widely open in "causal-learn," both forced and forbidden edges can be set, and the possibility of other unknown edges can be explored. This property, which is similar to that of DirectLiNGAM, means that the prior knowledge for this PC algorithm can be represented in a ternary matrix, if we need to do so. Therefore, the composition rule of $PK$ for PC is set to be the same as that for DirectLiNGAM in this work, to treat it as consistently as possible.

**For DirectLiNGAM** Although the criteria for setting the values in $PK$ are the same as those for PC, the definitions of the values slightly differ. Although the prior knowledge for the PC algorithm in the "causal-learn" package corresponds to the existence of directed edges between pairs of variables, the prior knowledge for DirectLiNGAM is determined with the knowledge of directed paths. The values of the matrix elements are determined as follows:

- Case 1. If the directed path from $x_i$ to $x_j$ is forced (i.e., $p_{ij} \geq 0.95$), then $PK_{ij}$ is set to 1.

- Case 2. If the directed path from $x_i$ to $x_j$ is forbidden (i.e., $p_{ij} < 0.05$), then $PK_{ij}$ is set to 0.

- Case 3. If the existence of the directed path from $x_i$ to $x_j$ cannot be determined immediately from the domain knowledge generated by GPT-4 (i.e., $0.05 \leq p_{ij} < 0.95$), then $PK_{ij}$ is set to $-1$.

This ternary matrix composition, which uses 1, 0 and $-1$, is indeed implemented in the software package "LiNGAM."

**For Exact Search** While $PK$ in cases of PC and DirectLiNGAM is a ternary matrix, one must be careful that $PK$ in Exact Search is a binary matrix. The values of the matrix elements are determined as follows:

- Case 4. If $x_i \rightarrow x_j$ is forbidden (i.e., $p_{ij} < 0.05$), then $PK_{ij}$ is set to 0.

- Case 5. If $x_i \rightarrow x_j$ is forced, or if the existence of this causal relationship cannot be determined immediately from the domain knowledge generated by GPT-4 (i.e., $0.05 \leq p_{ij}$), then $PK_{ij}$ is set to 1.

Notably, although the definition of $PK_{ij} = 0$ in Case 4 for Exact Search is exactly the same as that in Case 2 for PC and DirectLiNGAM, the definition of $PK_{ij} = 1$ in Case 5 for Exact Search encompasses both Case 1 and Case 3 for PC and DirectLiNGAM. This difference must be considered when evaluating $PK$ in comparison with the ground truths, to interpret the results in a unified manner regardless of the SCD methods used.

### E.2 Composition of the Ground Truth Matrix

The representation of ground truths in matrix form can be simply realized via a binary matrix, provided that it is determined whether a directed edge exists for every possible pair of variables in the system. The composition rule for the ground truth matrix $GT$ is as follows:

- If $x_j \to x_j$ exists, then $GT_{ij}$ is set to 1.

- If $x_j \to x_j$ does not exist, then $GT_{ij}$ is set to 0.

The matrix representations of the ground truths of the benchmark datasets of Auto MPG, DWD, and Sachs shown in Appendix C are expressed as follows:

$$GT_{\text{AutoMPG}} = \begin{pmatrix} 0 & 0 & 0 & 1 & 0 \\ 0 & 0 & 1 & 1 & 0 \\ 1 & 0 & 0 & 0 & 0 \\ 0 & 0 & 0 & 0 & 0 \\ 0 & 0 & 1 & 0 & 0 \end{pmatrix} \tag{20}$$

$$GT_{\text{DWD}} = \begin{pmatrix} 0 & 0 & 0 & 0 & 0 & 0 \\ 1 & 0 & 0 & 1 & 0 & 1 \\ 1 & 0 & 0 & 1 & 0 & 0 \\ 0 & 0 & 0 & 0 & 0 & 0 \\ 1 & 0 & 0 & 0 & 0 & 0 \\ 0 & 0 & 0 & 0 & 0 & 0 \end{pmatrix} \tag{21}$$

$$GT_{\text{Sachs}} = \begin{pmatrix} 0 & 0 & 0 & 0 & 0 & 0 & 0 & 0 & 1 & 0 & 0 \\ 1 & 0 & 0 & 0 & 0 & 0 & 0 & 1 & 1 & 0 & 0 \\ 0 & 0 & 0 & 0 & 1 & 0 & 0 & 0 & 0 & 0 & 0 \\ 0 & 0 & 1 & 0 & 1 & 0 & 0 & 0 & 0 & 0 & 0 \\ 0 & 0 & 0 & 0 & 0 & 0 & 0 & 0 & 0 & 0 & 0 \\ 0 & 1 & 0 & 0 & 0 & 0 & 0 & 1 & 0 & 0 & 0 \\ 0 & 0 & 0 & 0 & 1 & 1 & 0 & 1 & 0 & 0 & 0 \\ 0 & 0 & 0 & 0 & 0 & 0 & 0 & 0 & 1 & 0 & 0 \\ 0 & 0 & 1 & 1 & 0 & 0 & 0 & 0 & 0 & 0 & 0 \\ 0 & 0 & 0 & 0 & 0 & 0 & 0 & 1 & 1 & 0 & 0 \\ 0 & 0 & 0 & 0 & 0 & 0 & 0 & 1 & 1 & 0 & 0 \end{pmatrix} \tag{22}$$

### E.3 Calculation of Metrics for the Evaluation of Structural Consistency with Ground Truths (SHD, FPR, FNR, Precision, F1 Score)

Structural metrics such as the SHD, FPR, FNR, precision, and the F1 score are commonly calculated for performance evaluation in various machine learning and classification contexts, and are compared with the ground truth data. In a similar context, the causal structures inferred by LLM-KBCI and SCD, especially for benchmark datasets with known ground truths, can also be evaluated via these metrics.

For the practical evaluation of the SCD results in this study, we use the ground truth matrices defined for the benchmark datasets in Eq.(20), (21) and (22) as references, and we measure these metrics via the adjacency matrices that are calculated directly in the SCD algorithms or easily transformed from the output causal graphs. Similarly, the calculation of these metrics for evaluating the LLM-KBCI outputs is carried out via $PK$.

However, it must be noted that there can be some arguments on the definition of metrics for $PK$ on the basis of $GT$, because the definition of the matrix elements of $PK$ shown in Appendix E.1 is partially different from that of $GT$ described in Appendix E.2. In particular, although $GT_{ij}$ is a binary variable that is completely determined by whether $x_j \to x_i$ exists or not, $PK_{ij}$ can be set to $-1$ for PC and DirectLiNGAM and to 1 for Exact Search, when it is not impossible to definitively assert the presence or absence of $x_j \to x_i$ from the domain knowledge.

Therefore, although a reasonable extension of the definitions of these metrics may be required for the above case, in this study, we evaluate these metrics from $|\boldsymbol{PK}|$, in which both $PK_{ij} = 1$ and $PK_{ij} = -1$ are interpreted as "tentative assertions of the presence of $x_j \rightarrow x_i$" and are treated identically. This processing of $\boldsymbol{PK}$ can also be interpreted as that of temporarily adopting the composition rule of $\boldsymbol{PK}$ for Exact Search as it is for the evaluation of the SHD, FPR, FNR, precision, and the F1 score, for all the SCD methods. With this processing, $\boldsymbol{PK}$ is handled in a unified manner regardless of the SCD methods used. We believe that this approach is the best way to maintain the original concept of composing $\boldsymbol{PK}$, while aiming for consistent discussion across all SCD methods.

**Calculation of SHD**   According to the original concept of the structural hamming distance (SHD), this metric is represented as the total number of edge additions, deletions, or reversals that are needed to convert the estimated graph $G'$ into its ground truth graph $G$ (Zheng et al., 2018; Cheng et al., 2022; Hasan et al., 2023). As in our study, if network graphs $G$ and $G'$ are represented by binary matrices $\boldsymbol{G}$ and $\boldsymbol{G'}$, respectively, where all the elements are either 0 or 1, then the total number of edge additions ($A$), deletions ($D$), and reversals ($R$) can be simply calculated as follows:

$$A(\boldsymbol{G'}, \boldsymbol{G}) = \sum_{i,j} \mathbf{1}(G_{ij})\mathbf{1}(G_{ji})\mathbf{1}(G'_{ij} - 1) \tag{23}$$

$$D(\boldsymbol{G'}, \boldsymbol{G}) = \sum_{i,j} \mathbf{1}(G'_{ij})\mathbf{1}(G'_{ji})\mathbf{1}(G_{ij} - 1) \tag{24}$$

$$R(\boldsymbol{G'}, \boldsymbol{G}) = \sum_{i,j} \mathbf{1}(G_{ij})\mathbf{1}(G_{ji} - 1)\mathbf{1}(G'_{ij} - 1)\mathbf{1}(G'_{ji}) \tag{25}$$

Here, we introduce the indicator function $\mathbf{1}(x)$, expressed as follows:

$$\mathbf{1}(x) = \begin{cases} 1 & \text{if } x = 0, \\ 0 & \text{otherwise } (x \neq 0). \end{cases} \tag{26}$$

As the SHD is defined as $SHD = A + D + R$, it is easily evaluated as follows:

$$SHD(\boldsymbol{G'}, \boldsymbol{G}) = \\ \sum_{i,j} \Big\{ \ \mathbf{1}(G_{ij})\mathbf{1}(G_{ji})\mathbf{1}(G'_{ij} - 1) \ + \ \mathbf{1}(G'_{ij})\mathbf{1}(G'_{ji})\mathbf{1}(G_{ij} - 1) \ + \ \mathbf{1}(G_{ij})\mathbf{1}(G_{ji} - 1)\mathbf{1}(G'_{ij} - 1)\mathbf{1}(G'_{ji}) \ \Big\} \tag{27}$$

For the evaluation of the SHD of LLM-KBCI outputs, $SHD(|\boldsymbol{PK}|, \boldsymbol{GT})$ is calculated with Eq. (27).

**Calculation of FPR, FNR, Precision, and F1 score**   In a similar context to the SHD, for the calculation of metrics such as the false positive rate (FPR) and the false negative rate (FNR), we prepare the equation for evaluating the number of true positive (TP), false positive (FP), true negative (TN), and false negative (FN) edges as follows:

$$TP(\boldsymbol{G'}, \boldsymbol{G}) = \sum_{i,j} \mathbf{1}(G_{ij} - 1)\mathbf{1}(G'_{ij} - 1) \tag{28}$$

$$FP(\boldsymbol{G'}, \boldsymbol{G}) = \sum_{i,j} \mathbf{1}(G_{ij})\mathbf{1}(G'_{ij} - 1) \tag{29}$$

$$TN(\boldsymbol{G'}, \boldsymbol{G}) = \sum_{i,j} \mathbf{1}(G_{ij})\mathbf{1}(G'_{ij}) \tag{30}$$

$$FN(\boldsymbol{G'}, \boldsymbol{G}) = \sum_{i,j} \mathbf{1}(G_{ij} - 1)\mathbf{1}(G'_{ij}) \tag{31}$$

Then, using Eq. (28)– (31), the definition of the FPR, FNR, precision, and the F1 score can be expressed as follows:

$$FPR(\boldsymbol{G'}, \boldsymbol{G}) = \frac{FP(\boldsymbol{G'}, \boldsymbol{G})}{TN(\boldsymbol{G'}, \boldsymbol{G}) + FP(\boldsymbol{G'}, \boldsymbol{G})} \tag{32}$$

$$FNR(\boldsymbol{G'}, \boldsymbol{G}) = \frac{FN(\boldsymbol{G'}, \boldsymbol{G})}{TP(\boldsymbol{G'}, \boldsymbol{G}) + FN(\boldsymbol{G'}, \boldsymbol{G})} \tag{33}$$

$$Precision(\boldsymbol{G'}, \boldsymbol{G}) = \frac{TP(\boldsymbol{G'}, \boldsymbol{G})}{TP(\boldsymbol{G'}, \boldsymbol{G}) + FP(\boldsymbol{G'}, \boldsymbol{G})} \tag{34}$$

$$F_1 score(\boldsymbol{G'}, \boldsymbol{G}) = \frac{2\,TP(\boldsymbol{G'}, \boldsymbol{G})}{2\,TP(\boldsymbol{G'}, \boldsymbol{G}) + FN(\boldsymbol{G'}, \boldsymbol{G}) + FP(\boldsymbol{G'}, \boldsymbol{G})} \tag{35}$$

For the evaluation of structural metrics such as the FPR of LLM-KBCI outputs, $FPR(|\boldsymbol{PK}|, \boldsymbol{GT})$, $FNR(|\boldsymbol{PK}|, \boldsymbol{GT})$, $Precision(|\boldsymbol{PK}|, \boldsymbol{GT})$ and $F_1 score(|\boldsymbol{PK}|, \boldsymbol{GT})$ are calculated with Eq. (32)–(35).

## F   Algorithm $A$ for Transformation of Cyclic $PK$ into Acyclic Adjacency Matrices and Selection of the Optimal Matrix

As briefly described in Section 3.1, for the case of DirectLiNGAM, an acyclicity of $\boldsymbol{PK}$ is also required. Thus, if the $\boldsymbol{PK}$ directly calculated from the probability matrix is cyclic, it must be transformed into an acyclic form. One possible method is to delete the minimum number of edges included in cycles to transform $\boldsymbol{PK}$ into an acyclic matrix. However, several solutions may be transformed from the same $\boldsymbol{PK}$ with the minimum manipulation of deleting edges. Therefore, we have decided to carry out causal discovery with DirectLiNGAM for every possible acyclic prior knowledge matrix to select the best acyclic prior knowledge matrix $\boldsymbol{PK}_A$ in terms of statistical modeling. The dataset is again fitted with a structural equation model under the constraint of the causal structure explored with DirectLiNGAM, assuming linear-Gaussian data, and the BIC is calculated. After this process is repeated, the acyclic prior knowledge matrix with which the BIC becomes the lowest is selected as $\boldsymbol{PK}_A$.

The overall transformation process is described in Algorithm 3. However, in the practical application of this method, completing the list of acyclic prior knowledge matrices $A$ incurs significant computational costs. Hence, as the number of variables increases, completing this calculation algorithm in a realistic time frame becomes more challenging.

For the future generalization and application of our inference method via DirectLiNGAM, the development of more efficient algorithms for transforming a cyclic matrix into an acyclic matrix is anticipated.

---

**Algorithm 3** Transformation of Cyclic $\boldsymbol{PK}$ into Acyclic Adjacency Matrices and Selection of the Optimal Matrix

---

**Input 1:** Cyclic prior knowledge matrix $\boldsymbol{PK_C}$
**Input 2:** Data $X$ with variables$\{x_1, ..., x_n\}$
**Input 3:** DirectLiNGAM Algorithm $L(X, \boldsymbol{PK})$
**Output:** Optimal acyclic matrix $\boldsymbol{PK_A}$
Initialize the temporal set for matrices $\boldsymbol{T} \leftarrow \{\boldsymbol{PK_C}\}$
Initialize the temporal set for the number of cycles $\boldsymbol{N} \leftarrow \{\}$
Initialize the temporal set for acyclic matrices $\boldsymbol{A} \leftarrow \{\}$
**repeat**
  **for** matrix $\boldsymbol{T_m} \in \boldsymbol{T}$ **do**
    Count the number of cycles in $\boldsymbol{T_m}$ as $N_m$
    Add $N_m$ to $\boldsymbol{N}$
  **end for**
  **if** $\exists \boldsymbol{N_m} \in \boldsymbol{N} \; N_m = 0$ **then**
    Detect all $\boldsymbol{T_m}$, which satisfies $N_m = 0$ and add them to $\boldsymbol{A}$
  **else**
    Initialize the temporal set for modified matrices $\boldsymbol{T'} \leftarrow \{\}$
    **for** $\boldsymbol{T_m} \in \boldsymbol{T}$ **do**
      Initialize the set for edges included in cycles $\boldsymbol{E_m} \leftarrow \{\}$
      Initialize the set for edges to be removed $\boldsymbol{F_m} \leftarrow \{\}$
      Detect all cycles in $\boldsymbol{T_m}$
      For each detected cycle, identify all the edges that form the cycle
      Add these edges to a set of edges to be removed $\boldsymbol{E_m}$
      Detect the most frequent edges "$x_i \leftarrow x_j$" in $\boldsymbol{E_m}$ as $(i, j)_f$
      $\forall (i_f, j_f)$ add $(i_f, j_f)$ to $\boldsymbol{F_m}$
      **for** $(i_f, j_f) \in \boldsymbol{F_m}$ **do**
        $\boldsymbol{T'_m} = \boldsymbol{T_m}$
        $\boldsymbol{T'_m}(i_f, j_f) \leftarrow 0$
        Add $\boldsymbol{T'_m}$ to $\boldsymbol{T'}$
      **end for**
    **end for**
    Replace $\boldsymbol{T}$ with $\boldsymbol{T'}$
  **end if**
**until** $\boldsymbol{A}$ is not empty
Initialize the optimal BIC value $B = $ None
Initialize the optimal acyclic matrix for prior knowledge in DirectLiNGAM $\boldsymbol{A}_{\text{optimal BIC}} = $ None
**for** $\boldsymbol{A_m} \in \boldsymbol{A}$ **do**
  Calculate the adjacency matrix (with 0 or 1 components) of the causal discovery result $\boldsymbol{Adj} = L(X, \boldsymbol{PK} = \boldsymbol{A_m})$
  Fit $X$ with the structural causal equation model represented in $\boldsymbol{Adj}$ assuming linear-Gaussian data
  Calculate the BIC with $\boldsymbol{Adj}$ and $X$ as $B_{\text{temp}}$
  **if** $B > B_{\text{temp}}$ or $B = $ None **then**
    $B \leftarrow B_{\text{temp}}$
    $\boldsymbol{A}_{\text{optimal BIC}} = \boldsymbol{A_m}$
  **end if**
**end for**
**return** $\boldsymbol{A}_{\text{optimal BIC}}$ as $\boldsymbol{PK_A}$

---

# G  Details of LLM-KBCI Results

It is also valuable to examine the details of the probability matrices generated by LLM-KBCI, both for the basic discussion of whether LLMs can generate a valid interpretation of causality from a domain expert's point of view, and for understanding the characteristics of SCP. In this section, the probability matrices generated by GPT-4 for Auto MPG data and DWD climate data, which are relatively easy to interpret within common daily knowledge, are shown and briefly interpreted. For comparison among various SCP patterns (Patterns 1–4) using the same SCD method as much as possible, the probability matrices generated by GPT-4 with SCP are shown only for DirectLiNGAM. We also briefly present the probability matrices of LLM-KBCI for the sampled sub-dataset of health-screening results.

## G.1  LLM-KBCI for Auto MPG data

In Table 15, the probabilities of the causal relationships of pairs of variables in the Auto MPG data are shown. The cells highlighted in green are those in which the directed edges are expected to appear from the ground truths shown in Figure 7.

For all the prompting patterns, although the probability of "Weight"→"Displacement", which is interpreted as one of the ground truth directed edges, is 0, the probability of reversed edge "Displacement"→"Weight" is non-zero and over 0.95 in Patterns 1–4. To understand this behavior and elucidate the true causal relationship between these two variables, further discussion is required, including the possibility of the hidden common causes that are excluded from the dataset we have used.

In addition, although we do not believe the existence of the directed edge of "Displacement"→"Acceleration," the probability of this causal relationship is greater than 0.85 for all the prompting patterns. This may be due to the property of the prompt for evaluating the probability. As shown in Table 2, GPT-4 is allowed to judge the existence of both direct and indirect causal relationships, to acquire a positive answer even if any intervening variables are not included in the dataset. However, for example, considering that the probabilities of both "Displacement"→"Horsepower" and "Horsepower"→"Acceleration," which are part of the ground truths, are relatively high, it is also possible that GPT-4 supports the hypothesis of some impact from "Displacement" on "Acceleration" partially due to confidence in the indirect causal relationship of "Displacement"→"Horsepower"→"Acceleration". If one wants to distinguish the direct and indirect causal relationships in the interpretation of the probability matrix, investigating the response from LLMs for the first prompt may lead to further understanding.

Some differences that can be related to the prompting patterns can also be observed. For example, the probability of "Horsepower"→"Mpg" in Pattern 1 is much smaller than that in the other patterns. Moreover, the probabilities of "Horsepower"→"Acceleration" in Patterns 1 and 3 are smaller than those of the other patterns, in which the probability of this edge is almost 1. A possible explanation for these behaviors is that the decision-making of GPT-4 is unsettled with the SCP, in which the causal structure inferred by DirectLiNGAM shown in Figure 8 (c) is included. As neither "Horsepower"→"Acceleration" nor "Horsepower"→"Mpg" appears in Figure 8 (c), despite the confidence in the existence of these edges only from the domain knowledge, the decision-making of GPT-4 may become more careful, considering the result of SCD. In future work, the types of decision-making of LLMs that are likely to be affected by the SCP should be investigated.

## G.2  LLM-KBCI for DWD climate data

In Table 16, the probabilities of the causal relationships of pairs of variables in the DWD climate data are shown. The cells highlighted in green are those in which the directed edges are expected to appear from the ground truths shown in Figure 9.

For all the prompting patterns, it is confirmed that all of the probabilities of the causal effects on "Altitude," "Longitude," and "Latitude" from other variables are 0. As these three variables are geographically given and fixed, the interpretation by GPT-4 that they act as parent variables that are not influenced by other factors is completely reasonable. Although "Altitude" and "Latitude" are somehow influenced according to the result of DirectLiNGAM without prior knowledge as shown in Figure 10 (c), the SCP, including these

Table 15: Probabilities of the causal relationships suggested by GPT-4 in Auto MPG data. The cells in which the directed edges are expected to appear from the ground truths, as shown in Figure 7, are highlighted in green.

**Pattern 0**

| EFFECTED\CAUSE | "Displacement" | "Mpg" | "Horsepower" | "Weight" | "Acceleration" |
|---|---|---|---|---|---|
| "Displacement" | - | 0.000 | 0.000 | 0.000 | 0.000 |
| "Mpg" | 0.999 | - | 0.997 | 1.000 | 1.000 |
| "Horsepower" | 0.999 | 0.000 | - | 0.000 | 0.000 |
| "Weight" | 0.635 | 0.000 | 0.000 | - | 0.000 |
| "Acceleration" | 0.996 | 0.023 | 0.998 | 0.998 | - |

**Pattern 1**

| EFFECTED\CAUSE | "Displacement" | "Mpg" | "Horsepower" | "Weight" | "Acceleration" |
|---|---|---|---|---|---|
| "Displacement" | - | 0.000 | 0.000 | 0.000 | 0.000 |
| "Mpg" | 1.000 | - | 0.128 | 0.484 | 0.058 |
| "Horsepower" | 1.000 | 0.056 | - | 0.001 | 0.000 |
| "Weight" | 0.994 | 0.000 | 0.000 | - | 0.000 |
| "Acceleration" | 0.859 | 0.000 | 0.828 | 0.998 | - |

**Pattern 2**

| EFFECTED\CAUSE | "Displacement" | "Mpg" | "Horsepower" | "Weight" | "Acceleration" |
|---|---|---|---|---|---|
| "Displacement" | - | 0.000 | 0.000 | 0.000 | 0.000 |
| "Mpg" | 1.000 | - | 0.999 | 1.000 | 0.984 |
| "Horsepower" | 1.000 | 0.000 | - | 0.000 | 0.000 |
| "Weight" | 0.997 | 0.000 | 0.000 | - | 0.000 |
| "Acceleration" | 0.995 | 0.002 | 0.996 | 0.999 | - |

**Pattern 3**

| EFFECTED\CAUSE | "Displacement" | "Mpg" | "Horsepower" | "Weight" | "Acceleration" |
|---|---|---|---|---|---|
| "Displacement" | - | 0.000 | 0.000 | 0.000 | 0.000 |
| "Mpg" | 0.977 | - | 0.969 | 0.754 | 0.547 |
| "Horsepower" | 1.000 | 0.051 | - | 0.696 | 0.010 |
| "Weight" | 0.954 | 0.000 | 0.000 | - | 0.000 |
| "Acceleration" | 0.981 | 0.000 | 0.435 | 0.809 | - |

**Pattern 4**

| EFFECTED\CAUSE | "Displacement" | "Mpg" | "Horsepower" | "Weight" | "Acceleration" |
|---|---|---|---|---|---|
| "Displacement" | - | 0.000 | 0.000 | 0.000 | 0.000 |
| "Mpg" | 0.995 | - | 0.994 | 0.997 | 0.940 |
| "Horsepower" | 0.999 | 0.314 | - | 0.006 | 0.000 |
| "Weight" | 0.999 | 0.000 | 0.012 | - | 0.000 |
| "Acceleration" | 0.964 | 0.000 | 0.989 | 0.814 | - |

unnatural results, has not affected the decision-making of GPT-4. From this behavior, it is inferred that the response regarding axiomatic and self-evident matters from GPT-4 is robust and unlikely to be affected by the SCP, even if the SCD result clearly exhibits unnatural behaviors.

In addition, although "Longitude"→"Temperature," which is assumed to be a ground truth, is unlikely to be asserted by GPT-4, "Temperature"→"Precipitation," which is not expected to be a ground truth, is likely to be asserted by GPT-4, across all the prompting patterns. For further interpretation of these unexpected behaviors from our ground truths, investigation of the response generated in the first prompting process is recommended. Interestingly, although the probabilities of "Longitude"→"Precipitation" are 0 in Patterns 0–2, they become non-zero finite values in Patterns 3 and 4, in which the causal coefficient of this directed

edge calculated with DirectLiNGAM is included in the SCP. This behavior may indicate that the SCP can assist the decision-making of GPT-4 even if it generates an incomplete response on causal relationships with its background knowledge.

Table 16: Probabilities of the causal relationships suggested by GPT-4 in DWD climate data. The cells in which the directed edges are expected to appear from the ground truths as shown in Figure 9 are highlighted in green.

**Pattern 0**

| EFFECTED\CAUSE | "Altitude" | "Temperature" | "Precipitation" | "Longitude" | "Sunshine" | "Latitude" |
|---|---|---|---|---|---|---|
| "Altitude" | - | 0.000 | 0.000 | 0.000 | 0.000 | 0.000 |
| "Temperature" | 1.000 | - | 0.891 | 0.000 | 1.000 | 1.000 |
| "Precipitation" | 1.000 | 0.999 | - | 0.000 | 0.001 | 0.995 |
| "Longitude" | 0.000 | 0.000 | 0.000 | - | 0.000 | 0.000 |
| "Sunshine" | 1.000 | 0.000 | 0.998 | 0.000 | - | 1.000 |
| "Latitude" | 0.000 | 0.000 | 0.000 | 0.000 | 0.000 | - |

**Pattern 1**

| EFFECTED\CAUSE | "Altitude" | "Temperature" | "Precipitation" | "Longitude" | "Sunshine" | "Latitude" |
|---|---|---|---|---|---|---|
| "Altitude" | - | 0.000 | 0.000 | 0.000 | 0.000 | 0.000 |
| "Temperature" | 0.384 | - | 0.034 | 0.000 | 0.856 | 0.011 |
| "Precipitation" | 0.025 | 0.999 | - | 0.000 | 0.036 | 0.026 |
| "Longitude" | 0.000 | 0.000 | 0.000 | - | 0.000 | 0.000 |
| "Sunshine" | 0.006 | 0.011 | 0.008 | 0.000 | - | 0.596 |
| "Latitude" | 0.000 | 0.000 | 0.000 | 0.000 | 0.000 | - |

**Pattern 2**

| EFFECTED\CAUSE | "Altitude" | "Temperature" | "Precipitation" | "Longitude" | "Sunshine" | "Latitude" |
|---|---|---|---|---|---|---|
| "Altitude" | - | 0.000 | 0.000 | 0.000 | 0.000 | 0.000 |
| "Temperature" | 0.997 | - | 0.007 | 0.000 | 0.999 | 0.989 |
| "Precipitation" | 0.739 | 0.999 | - | 0.000 | 0.000 | 0.384 |
| "Longitude" | 0.000 | 0.000 | 0.000 | - | 0.000 | 0.000 |
| "Sunshine" | 0.874 | 0.010 | 0.976 | 0.000 | - | 0.981 |
| "Latitude" | 0.000 | 0.000 | 0.000 | 0.000 | 0.000 | - |

**Pattern 3**

| EFFECTED\CAUSE | "Altitude" | "Temperature" | "Precipitation" | "Longitude" | "Sunshine" | "Latitude" |
|---|---|---|---|---|---|---|
| "Altitude" | - | 0.000 | 0.000 | 0.000 | 0.000 | 0.000 |
| "Temperature" | 0.919 | - | 0.016 | 0.003 | 0.615 | 0.973 |
| "Precipitation" | 0.585 | 0.996 | - | 0.175 | 0.002 | 0.008 |
| "Longitude" | 0.000 | 0.000 | 0.000 | - | 0.000 | 0.000 |
| "Sunshine" | 0.039 | 0.000 | 0.001 | 0.875 | - | 0.199 |
| "Latitude" | 0.000 | 0.000 | 0.000 | 0.000 | 0.000 | - |

**Pattern 4**

| EFFECTED\CAUSE | "Altitude" | "Temperature" | "Precipitation" | "Longitude" | "Sunshine" | "Latitude" |
|---|---|---|---|---|---|---|
| "Altitude" | - | 0.000 | 0.000 | 0.000 | 0.000 | 0.000 |
| "Temperature" | 0.982 | - | 0.023 | 0.029 | 0.990 | 0.958 |
| "Precipitation" | 0.826 | 0.987 | - | 0.927 | 0.010 | 0.797 |
| "Longitude" | 0.000 | 0.000 | 0.000 | - | 0.000 | 0.000 |
| "Sunshine" | 0.534 | 0.021 | 0.387 | 0.013 | - | 0.638 |
| "Latitude" | 0.000 | 0.000 | 0.000 | 0.000 | 0.000 | - |

### G.3 LLM-KBCI for Dataset of health-screening Results

In Table 17, the probabilities of the causal relationships of pairs of variables in our sampled sub-dataset of health-screening results are shown. The cells highlighted in red are those in which the directed edges are expected to appear, as described in Appendix C.4. In contrast, since "Age" is an unmodifiable background factor, it can be concluded that it is not a descendant of any other variables. Therefore, the probabilities in the cells highlighted in blue are expected to be 0.

Across all the prompting patterns, it is confirmed that all the probabilities of the causal effects on "Age" from other variables are indeed 0. From this fact, it is likely to be regarded by GPT-4 as axiomatic and self-evident that "Age" cannot be affected by other variables, and the judgment of causal relationships is not influenced by the SCP, even if the SCD result clearly exhibits unnatural behaviors as shown in Figure 14.

Table 17: Probabilities of the causal relationships suggested by GPT-4 in the sampled sub-dataset of health-screening results. The cells in which the directed edges are expected to appear from the ground truths are highlighted in red. In contrast, the probabilities in the cells highlighted in blue, are expected to be zero, since "Age" is expected to be a parent variable for all other variables.

**Pattern 0**

| EFFECTED\CAUSE | "BMI" | "Waist" | "SBP" | "DBP" | "HbA1c" | "LDL" | "Age" |
|---|---|---|---|---|---|---|---|
| "BMI" | - | 0.994 | 0.000 | 0.000 | 0.000 | 0.000 | 0.901 |
| "Waist" | 1.000 | - | 0.000 | 0.000 | 0.000 | 0.000 | 0.353 |
| "SBP" | 0.999 | 0.962 | - | 0.998 | 0.987 | 0.000 | 0.626 |
| "DBP" | 0.998 | 0.995 | 0.993 | - | 0.000 | 0.000 | 0.001 |
| "HbA1c" | 0.998 | 0.998 | 0.000 | 0.000 | - | 0.000 | 0.986 |
| "LDL" | 0.988 | 0.967 | 0.000 | 0.000 | 0.000 | - | 0.002 |
| "Age" | 0.000 | 0.000 | 0.000 | 0.000 | 0.000 | 0.000 | - |

**Pattern 1**

| EFFECTED\CAUSE | "BMI" | "Waist" | "SBP" | "DBP" | "HbA1c" | "LDL" | "Age" |
|---|---|---|---|---|---|---|---|
| "BMI" | - | 0.312 | 0.000 | 0.000 | 0.014 | 0.000 | 0.076 |
| "Waist" | 1.000 | - | 0.000 | 0.000 | 0.023 | 0.000 | 0.043 |
| "SBP" | 0.999 | 0.912 | - | 0.999 | 0.997 | 0.000 | 0.302 |
| "DBP" | 0.998 | 0.421 | 0.050 | - | 0.000 | 0.000 | 0.019 |
| "HbA1c" | 0.517 | 0.503 | 0.101 | 0.000 | - | 0.000 | 0.170 |
| "LDL" | 0.008 | 0.527 | 0.000 | 0.000 | 0.000 | - | 0.517 |
| "Age" | 0.000 | 0.000 | 0.000 | 0.000 | 0.000 | 0.000 | - |

**Pattern 2**

| EFFECTED\CAUSE | "BMI" | "Waist" | "SBP" | "DBP" | "HbA1c" | "LDL" | "Age" |
|---|---|---|---|---|---|---|---|
| "BMI" | - | 0.998 | 0.001 | 0.001 | 0.996 | 0.000 | 0.093 |
| "Waist" | 0.999 | - | 0.000 | 0.003 | 0.959 | 0.007 | 0.099 |
| "SBP" | 0.998 | 0.994 | - | 0.983 | 0.994 | 0.040 | 0.207 |
| "DBP" | 0.997 | 0.975 | 0.984 | - | 0.983 | 0.002 | 0.115 |
| "HbA1c" | 0.982 | 0.608 | 0.002 | 0.000 | - | 0.000 | 0.723 |
| "LDL" | 0.994 | 0.946 | 0.000 | 0.000 | 0.452 | - | 0.171 |
| "Age" | 0.000 | 0.000 | 0.000 | 0.000 | 0.000 | 0.000 | - |

**Pattern 3**

| EFFECTED\CAUSE | "BMI" | "Waist" | "SBP" | "DBP" | "HbA1c" | "LDL" | "Age" |
|---|---|---|---|---|---|---|---|
| "BMI" | - | 0.003 | 0.000 | 0.000 | 0.868 | 0.923 | 0.306 |
| "Waist" | 1.000 | - | 0.000 | 0.000 | 0.983 | 0.000 | 0.076 |
| "SBP" | 1.000 | 0.855 | - | 0.959 | 0.999 | 0.000 | 0.235 |
| "DBP" | 1.000 | 0.032 | 0.140 | - | 0.021 | 0.000 | 0.095 |
| "HbA1c" | 0.967 | 0.634 | 0.000 | 0.000 | - | 0.000 | 0.046 |
| "LDL" | 0.562 | 0.165 | 0.000 | 0.000 | 0.085 | - | 0.013 |
| "Age" | 0.000 | 0.000 | 0.000 | 0.000 | 0.000 | 0.000 | - |

**Pattern 4**

| EFFECTED\CAUSE | "BMI" | "Waist" | "SBP" | "DBP" | "HbA1c" | "LDL" | "Age" |
|---|---|---|---|---|---|---|---|
| "BMI" | - | 0.993 | 0.000 | 0.000 | 0.024 | 0.006 | 0.037 |
| "Waist" | 1.000 | - | 0.000 | 0.000 | 0.957 | 0.000 | 0.395 |
| "SBP" | 0.999 | 0.982 | - | 0.001 | 0.998 | 0.000 | 0.795 |
| "DBP" | 0.994 | 0.204 | 0.985 | - | 0.408 | 0.000 | 0.926 |
| "HbA1c" | 0.824 | 0.391 | 0.000 | 0.000 | - | 0.000 | 0.176 |
| "LDL" | 0.485 | 0.403 | 0.000 | 0.000 | 0.000 | - | 0.027 |
| "Age" | 0.000 | 0.000 | 0.000 | 0.000 | 0.000 | 0.000 | - |

# H    Applicability of the Proposed Method Across Various LLMs

In the main body of this paper, we fix the LLM to `GPT-4-1106-preview`, to maintain consistency and control across the trials with several kinds of datasets. In this section, however, to confirm the universal applicability of the proposed method, we conduct the same experiment of the proposed method with several LLMs, and confirm that adapting the results of LLM-KBCI as prior knowledge of SCD indeed further enhances the performance of SCD with various LLMs.

## H.1    Independently Assigning GPTs for Steps 2 and 3

For this supplemental experiment, in addition to `GPT-4-1106-preview`, we have adopted `GPT-3.5-turbo-1106`, `GPT-3.5-turbo-0125`, `GPT-4-0125-preview`, and `GPT-4o-2024-08-06`. Although these LLMs are mutually different in architecture and generation, the format of the input and output, and the API settings  (including the log-probability extraction) are standardized across the models. Therefore, the technical setup is standardized within this range of comparison, and it is easy to compare the performance of these LLMs in the proposed method.

As explained in Sections 1 and 3, LLMs are utilized both in Steps 2 and 3. Therefore, we choose the LLMs for these two steps independently, and confirmed every combination of LLMs in these steps. Furthermore, we adopt the prompting Pattern 0  (prompting without the results of SCD) for comparison, since the purpose of this experiment is to demonstrate that various LLMs can improve the SCD by providing prior knowledge. We also fix the dataset to DWD climate data, for the comparison of performance.

The experimental results are summarized in Table  18. In the same manner as Table  3 in Section  4.1, we evaluate the final SCD results obtained in Step 4, with the SHD, FPR, FNR, precision, and the F1 score, using the ground truth adjacency matrix $\textbf{\textit{GT}}$ as a reference. Furthermore, we also evaluate the CFI, RMSEA, and BIC of the causal structure obtained after Step 4, under the assumption of linear-Gaussian data; to evaluate the results with respect to the statistical validity of the calculated causal models. To evaluate the effect of $\textbf{\textit{PK}}$ augmentation by the LLMs in Pattern 0, the baseline result is that without $\textbf{\textit{PK}}$ (Baseline A).

It is clearly observed that in any combination of LLMs in Steps 2 and 3, the SHD, FPR, FNR, and BIC after Step 4 become equal to or under the values of Baseline A (without $\textbf{\textit{PK}}$), and that the precision and F1 score after Step 4 become larger than those of Baseline A. These facts clearly suggest that the proposed method in this paper indeed improves the SCD results regardless of the LLMs adopted in Steps 2 and 3.

However, it is not clear from this table what types of LLMs can further improve the SCD results.  For example, although the GPT-4 series has been reported to perform better than the GPT-3.5 series in many tasks (OpenAI, 2023), this tendency is not observed in Table  18. Moreover, although `GPT-3.5-turbo-0125` and `GPT-4-0125-preview` are updated models of `GPT-3.5-turbo-1106` and `GPT-4-0125-preview`, respectively, the improvement related to these updates is not observed in Table  18. Therefore, for the optimal selection of the LLMs for practical application of this method across various domains, further research is needed on the relationship between the LLM properties and the performance on causal reasoning tasks in each domain, including the possibility of adopting RAG or fine-tuning.

Table 18: Comparison of the SCD results after Step 4 in Pattern 0 (prompting without any SCD results) across various GPT models, on the DWD climate dataset. While the SHD, FPR, FNR, precision, and the F1 score are compared to evaluate how close the results are to the ground truths, CFI, RMSEA, and BIC are compared to evaluate the statistical validity of the calculated causal models. Lower values are superior for the SHD, FPR, FNR, RMSEA and BIC, and higher values for the precision, F1 score and CFI. The underlined values are the same as those in Table 3, which are obtained only with `GPT-4-1106-preview`. Baseline A is used for the comparison with the SCD results augmented with *PK* generated by the LLM, to evaluate the effect of ***PK*** augmentation by the LLM. It is suggested that the proposed method in this paper indeed improves the SCD results regardless of the combination of LLMs adopted in Steps 2 and 3.

| LLM for Step 2 | LLM for Step 3 | SHD↓ | FPR↓ | FNR↓ | Precision↑ | F1score↑ | CFI↑ | RMSEA↓ | BIC↓ |
|---|---|---|---|---|---|---|---|---|---|
| wo **PK**(Baseline A) | | 10 | 0.33 | 0.67 | 0.17 | 0.22 | 1.00 | 0.00 | 99.53 |
| gpt-3.5-turbo-1106 | gpt-3.5-turbo-1106 | 4 | 0.07 | 0.33 | 0.67 | 0.67 | 0.93 | 0.16 | 51.98 |
| | gpt-3.5-turbo-0125 | 4 | 0.07 | 0.33 | 0.67 | 0.67 | 0.93 | 0.16 | 51.98 |
| | gpt-4-1106-preview | 3 | 0.07 | 0.17 | 0.71 | 0.77 | 0.93 | 0.16 | 57.90 |
| | gpt-4-0125-preview | 4 | 0.07 | 0.33 | 0.67 | 0.67 | 0.93 | 0.16 | 51.98 |
| | gpt-4o-2024-08-06 | 5 | 0.07 | 0.50 | 0.60 | 0.55 | 1.00 | 0.00 | 46.81 |
| gpt-3.5-turbo-0125 | gpt-3.5-turbo-1106 | 4 | 0.03 | 0.50 | 0.75 | 0.60 | 0.98 | 0.09 | 40.80 |
| | gpt-3.5-turbo-0125 | 4 | 0.07 | 0.33 | 0.67 | 0.67 | 0.93 | 0.16 | 51.98 |
| | gpt-4-1106-preview | 5 | 0.07 | 0.50 | 0.60 | 0.55 | 1.00 | 0.00 | 46.81 |
| | gpt-4-0125-preview | 5 | 0.07 | 0.50 | 0.60 | 0.55 | 1.00 | 0.00 | 46.81 |
| | gpt-4o-2024-08-06 | 4 | 0.07 | 0.33 | 0.67 | 0.67 | 1.00 | 0.00 | 52.67 |
| gpt-4-1106-preview | gpt-3.5-turbo-1106 | 4 | 0.07 | 0.33 | 0.67 | 0.67 | 1.00 | 0.00 | 52.67 |
| | gpt-3.5-turbo-0125 | 4 | 0.07 | 0.33 | 0.67 | 0.67 | 1.00 | 0.00 | 52.67 |
| | gpt-4-1106-preview | 4 | 0.07 | 0.33 | 0.67 | 0.67 | 1.00 | 0.00 | 52.67 |
| | gpt-4-0125-preview | 4 | 0.07 | 0.33 | 0.67 | 0.67 | 1.00 | 0.00 | 52.67 |
| | gpt-4o-2024-08-06 | 4 | 0.07 | 0.33 | 0.67 | 0.67 | 1.00 | 0.00 | 52.67 |
| gpt-4-0125-preview | gpt-3.5-turbo-1106 | 5 | 0.07 | 0.50 | 0.60 | 0.55 | 1.00 | 0.00 | 46.81 |
| | gpt-3.5-turbo-0125 | 4 | 0.07 | 0.33 | 0.67 | 0.67 | 0.93 | 0.16 | 51.98 |
| | gpt-4-1106-preview | 5 | 0.07 | 0.50 | 0.60 | 0.55 | 1.00 | 0.00 | 46.81 |
| | gpt-4-0125-preview | 5 | 0.07 | 0.50 | 0.60 | 0.55 | 1.00 | 0.00 | 46.81 |
| | gpt-4o-2024-08-06 | 5 | 0.07 | 0.50 | 0.60 | 0.55 | 1.00 | 0.00 | 46.81 |
| gpt-4o-2024-08-06 | gpt-3.5-turbo-1106 | 4 | 0.03 | 0.50 | 0.75 | 0.60 | 0.98 | 0.09 | 40.80 |
| | gpt-3.5-turbo-0125 | 4 | 0.07 | 0.33 | 0.67 | 0.67 | 1.00 | 0.00 | 52.67 |
| | gpt-4-1106-preview | 5 | 0.07 | 0.50 | 0.60 | 0.55 | 1.00 | 0.00 | 46.81 |
| | gpt-4-0125-preview | 5 | 0.07 | 0.50 | 0.60 | 0.55 | 1.00 | 0.00 | 46.81 |
| | gpt-4o-2024-08-06 | 7 | 0.10 | 0.67 | 0.40 | 0.36 | 0.51 | 0.53 | 42.85 |

## H.2 Evaluation with Open-Source LLMs

To validate the universal effectiveness of the proposed method in this work across various LLMs, including open-source LLMs, we conducted the same experiments using several open-source LLMs. Specifically, we selected three representative open-source LLMs: `Llama-3.1-70B-Instruct` [28](team, 2024), `gemma-2-27b-it` [29](Team, 2024) , and `Qwen2.5-32B-Instruct`[30](Team, 2024; Yang et al., 2024). In these experiments, the same LLM was used in both Step 2 and Step 3. The parameter settings and prompts were consistent with those described in Section 3.1.

The results with three types of open-source LLMs are summarized in Table 19. It is clearly confirmed that the SHD, FPR, FNR, and BIC after Step 4 are lower than those of Baseline A (without $PK$) across all open-source LLMs adopted in this experiment. This result clearly supports the fact that the improvement with SCP does not merely comes from GPT-series models, but comes from the proposed method, which integrates LLMs into SCD.

---

[28]https://huggingface.co/meta-llama/Llama-3.1-70B-Instruct
[29]https://huggingface.co/google/gemma-2-27b-it
[30]https://huggingface.co/Qwen/Qwen2.5-32B-Instruct

Table 19: Comparison of the SCD results in Pattern 0 (prompting without any SCD results) with the DWD climate dataset we have conducted across several open-source LLMs. The underlined values are the same as those in Table 3, which are obtained only with `GPT-4-1106-preview`. Baseline A is used for the comparison with the SCD results augmented with *PK* generated by the LLM, to evaluate the effect of **PK** augmentation by the LLM. It is suggested that the proposed method in this paper indeed improves the SCD results with various open-source LLMs, even those outside the GPT series.

| LLM for Steps 2 and 3 | SHD↓ | FPR↓ | FNR↓ | Precision↑ | F1score↑ | CFI↑ | RMSEA↓ | BIC↓ |
|---|---|---|---|---|---|---|---|---|
| wo **PK**(Baseline A) | 10 | 0.33 | 0.67 | 0.17 | 0.22 | 1.00 | 0.00 | 99.53 |
| Llama-3.1-70B-Instruct | 4 | 0.07 | 0.33 | 0.67 | 0.67 | 1.00 | 0.00 | 52.67 |
| gemma-2-27b-it | 6 | 0.10 | 0.50 | 0.50 | 0.50 | 0.92 | 0.17 | 51.92 |
| Qwen2.5-7B-Instruct | 6 | 0.10 | 0.50 | 0.50 | 0.50 | 0.73 | 0.43 | 50.46 |
| gpt-4-1106-preview (used in the main body) | 4 | 0.07 | 0.33 | 0.67 | 0.67 | 1.00 | 0.00 | 52.67 |

# I Prompt Template Adjustment for Estimating LLM Confidence in Causal Relationships

To fix the prompt templates described in Tables 1, 2, and 12, we also considered the sensitivity of LLM responses to the prompts (Sclar et al., 2024; Cao et al., 2024). Although many functions and tools have been developed and released by the community for optimizing prompts, we carefully focused on how to ask LLMs about the causal relationship from $x_j$ to $x_i$. In this section, we provide a supplementary explanation of how we decided to adopt the following question: "If $x_j$ is modified, will it have a direct or indirect impact on $x_i$?" as the core sentence of the prompt template $q_{ij}^{(2)}$ in Table 2.

First we prepared 20 types of templates for asking LLMs about the causal relationship from $x_j$ to $x_i$ as summarized in Table|20. These templates were generated the `GPT-3.5-turbo`.

Table 20: List of prompt templates $Q_n(x_i, x_j)$ for asking LLMs about the causal relationship from $x_j$ to $x_i$. $Q_4(x_i, x_j)$ highlighted in bold is the original form of the adopted template.

| Number $n$ | Template $Q_n(x_i, x_j)$ |
|:---:|:---:|
| 1 | Does $x_j$ lead to $x_i$? |
| 2 | Can $x_j$ be attributed to the cause of $x_i$? |
| 3 | Is there a causal relationship between $x_j$ and $x_i$? |
| 4 | **If $x_j$ is modified, will it have an impact on $x_i$?** |
| 5 | We believe that $x_j$ is responsible for $x_i$. Is this assertion valid? |
| 6 | Is $x_i$ influenced by $x_j$? |
| 7 | Can changes in $x_j$ result in changes in $x_i$? |
| 8 | Is there evidence to suggest that $x_j$ is the cause of $x_i$? |
| 9 | Does $x_j$ have a direct effect on $x_i$? |
| 10 | If $x_j$ is manipulated, will $x_i$ be affected? |
| 11 | Is there a connection between $x_j$ and $x_i$, where $x_j$ is the cause and $x_i$ is the effect? |
| 12 | Can $x_j$ bring about $x_i$? |
| 13 | Will $x_i$ change if $x_j$ is altered? |
| 14 | Is there a correlation between $x_j$ and $x_i$, indicating a causal relationship? |
| 15 | Is $x_j$ a determining factor for $x_i$? |
| 16 | Can $x_j$ be considered a potential cause of $x_i$? |
| 17 | If $x_j$ is controlled, will it impact $x_i$? |
| 18 | Is there a link between $x_j$ and $x_i$, suggesting a causal association? |
| 19 | Does $x_j$ play a role in causing $x_i$? |
| 20 | Is there a causal-effect relationship between $x_j$ and $x_i$? |

Next, we prepared a brief benchmark test to evaluate the templates shown in Table 20. We adopted the ground truth relationships among the variables "hour of day," "temperature," and "electricity load," which are included in the Tübingen database for cause-effect pairs (Mooij et al., 2016). Figure 19 illustrates the benchmark ground truths used in this evaluation. On the basis of these ground truths, we created variable name pairs, as shown in Table 21, and filled them into the templates to construct six questions .

Under the conditions described above, we evaluated the templates via zero-shot prompting, and generated knowledge prompting (Liu et al., 2022). All the experiments were conducted using `GPT-3.5-turbo`, with the temperature fixed at 0.7. For all the questions, we repeated the same prompting 50 times, and calculated the accuracy for all the prompt templates in Table 20.

## I.1 Results of Zero-Shot Prompting

To obtain the answer in the form of yes or no, We simply used the prompting template shown in Table 22, with the question templates in Table 20 and variable the name lists in Table 21.

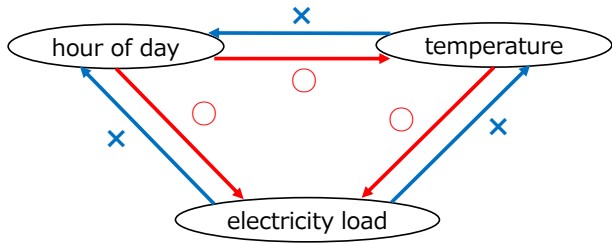

Figure 19: The benchmark ground truths based on the Tübingen database for cause-effect pairs (Mooij et al., 2016).

Table 21: Variable name pairs filled into the templates $Q_n(x_i, x_j)$ for constructing questions and the correct answers on the basis of Figure 19.

| Question Number | $x_i$ | $x_j$ | Correct Answer |
| --- | --- | --- | --- |
| 1 | the temperature | the hour of day | Yes |
| 2 | the electricity load | the hour of day | Yes |
| 3 | the electricity load | the temperature | Yes |
| 4 | the hour of day | the temperature | No |
| 5 | the hour of day | the electricity load | No |
| 6 | the temperature | the electricity load | No |

Table 22: Prompt template for zero-shot prompting experiments.

---

**Template for Zero-shot Prompting Experiments**

Let's discuss the problem of causation on energy consumption related with climates.
$Q_n(x_i, x_j)$
Please answer with <yes> or <no>.
No answers except these two responses are needed.

---

The top four and worst four templates, which are based on overall accuracy under zero-shot prompting, are summarized in Table 23. Although the template $Q_n(x_i, x_j)$ shows the best performance when $n = 4, 5, 12, 19$, the performance becomes poor when $n = 3, 14, 18.20$, under the prompt condition of zero-shot.

## I.2 Results of Generated Knowledge Prompting

To obtain the knowledge used in the prompting process described later, we utilized the ChatGPT plugin's functionality to connect to Wikipedia, using the prompt shown in Table 24 for the three variables. We then integrated all of the generated knowledge into a unified form, as shown in Table 25. Finally, the generated knowledge prompting (Liu et al., 2022) was performed using the template described in Table 26, in combination with the question templates from Table 20 and the variable name pairs from Table 21.

The top three and worst three templates, based on overall accuracy under generated knowledge prompt, are summarized in Table 23. Although the template $Q_n(x_i, x_j)$ shows the best performance when $n = 4, 12, 19$, the performance becomes poor when $n = 14, 8, 5$, under the prompting condition of generated knowledge prompting.

In terms of the universality of the proposed method, it is ideal to adopt a prompt template that performs well in both zero-shot prompting and generated knowledge prompting. From this perspective, and on the basis of the results shown in Table 23 and Table 27, the prompt templates with $n = 4$ and $n = 19$ appear to be the most effective. As shown in Table 20, both templates are simple and easy to modify if needed. However, whereas the template with $n = 19$ ("Does $x_j$ play a role in causing $x_i$?") treats $x_i$ as a phenomenon, the

Table 23: Summary of the experimental results of zero-shot prompting.

**Top 4 in Overall Accuracy**

| Template | Q.1 | Q.2 | Q.3 | Q.4 | Q.5 | Q.6 | Overall Accuracy |
|---|---|---|---|---|---|---|---|
| $Q_{19}(x_i, x_j)$ | 0.14 | 0.70 | 0.98 | 0.98 | 0.64 | 0.62 | 0.68 |
| $Q_4(x_i, x_j)$ | 0.14 | 0.68 | 0.96 | 0.98 | 0.72 | 0.54 | 0.67 |
| $Q_5(x_i, x_j)$ | 0.12 | 0.70 | 1.00 | 0.94 | 0.64 | 0.60 | 0.67 |
| $Q_{12}(x_i, x_j)$ | 0.10 | 0.72 | 0.98 | 0.96 | 0.64 | 0.58 | 0.66 |

**Worst 4 in Overall Accuracy**

| Template | Q.1 | Q.2 | Q.3 | Q.4 | Q.5 | Q.6 | Overall Accuracy |
|---|---|---|---|---|---|---|---|
| $Q_{20}(x_i, x_j)$ | 0.16 | 0.64 | 0.92 | 0.00 | 0.42 | 0.48 | 0.44 |
| $Q_3(x_i, x_j)$ | 0.16 | 0.64 | 0.94 | 0.08 | 0.32 | 0.46 | 0.43 |
| $Q_{14}(x_i, x_j)$ | 0.20 | 0.66 | 1.00 | 0.02 | 0.30 | 0.40 | 0.43 |
| $Q_{18}(x_i, x_j)$ | 0.08 | 0.56 | 0.96 | 0.08 | 0.42 | 0.42 | 0.42 |

Table 24: Prompt Used for Knowledge Acquisition. in the blank of $x_i$, "the hour of day, " "the temperature, " or "the electricity load" is filled.

**Template for Knowledge Acquisition**

In the context of the relation between the climate and the energy consumption, please provide the knowledge of $x_i$ in detail as much as possible.

template with $n = 4$ ("If $x_j$ is modified, will it have an impact on $x_j$?") treats $x_j$ as a variable. TTherefore, considering that our focus is on improving SCD using datasets composed of continuous variables, we judged that the template of $n = 4$ is the best fit for our purpose.

Table 25: Integrated knowledge for "the hour of day, " "the temperature, " and "the electricity load."

| Integrated Knowledge |
| --- |

1. about the hour of day:
Early Morning (12:00 AM - 6:00 AM)
Lower Energy Demand: Most people are asleep, leading to lower energy consumption for lighting, appliances, and electronics.
Heating/Cooling: Depending on the season and climate, heating or cooling systems may be in use, but generally at lower settings. Industrial Use: Some factories and industries operate 24/7, contributing to a baseline level of energy consumption.
Morning (6:00 AM - 9:00 AM)
Peak in Energy Demand: People wake up, turn on lights, and use appliances, leading to a surge in energy demand.
Heating/Cooling: In colder climates, heating systems may be turned up as people prepare for the day. In hotter climates, cooling systems may be activated.
Transport: Increased use of public and private transportation, contributing to fuel consumption.

Midday (9:00 AM - 3:00 PM)
Stable Demand: Energy demand stabilizes but remains high due to industrial and commercial activities.
Solar Energy: Peak hours for solar energy production, which can offset some fossil fuel consumption.
Air Conditioning: In hot climates, the demand for cooling can be very high, especially in commercial buildings.

Afternoon to Early Evening (3:00 PM - 7:00 PM)
Peak Demand: Another surge often occurs as people return home and use appliances, electronics, and lighting.
Heating/Cooling: Depending on the season, heating or cooling systems may be turned up again.
Solar Energy: Solar energy production starts to decline as the sun sets.

Late Evening (7:00 PM - 12:00 AM)
Declining Demand: Energy demand gradually decreases as people go to bed.
Heating/Cooling: Systems may still be in use but generally at lower settings.
Off-Peak: Some utilities use this time to generate energy at lower costs, sometimes using less efficient but cheaper power plants.

2. about the temperature:
Heating and Cooling
Heating in Cold Climates: In colder climates, a significant amount of energy is consumed for heating buildings. This is often done through natural gas or electric heating systems. The colder the climate, the more energy is required to maintain a comfortable indoor temperature.

Cooling in Hot Climates: Conversely, in hot climates, air conditioning is used extensively, which also consumes a large amount of electricity.

Seasonal Variations
Winter: Energy consumption often spikes during the winter months in colder climates due to heating needs.
Summer: In hot climates, the opposite is true; energy consumption can be highest in the summer due to air conditioning.

3. about the electricity load:
Seasonal Variations
Summer: Hot weather often leads to increased electricity consumption due to the use of air conditioning and cooling systems.
Winter: Cold weather can also lead to high electricity demand for heating, although in many places, heating is more commonly fueled by natural gas or oil.

Time of Day
Peak Hours: These are the times when electricity demand is highest, often in the late afternoon and early evening.
Off-Peak Hours: These are times when electricity demand is lower, typically late at night and early in the morning.

Climate Change
Changing Patterns: Climate change could lead to more extreme weather events, affecting electricity load patterns.
Adaptation: As the climate changes, new patterns of electricity consumption may emerge, requiring adaptive strategies.

Table 26: Prompt Used for Knowledge Acquisition. in the blank of $x_i$, "the hour of day, " "the temperature, " or "the electricity load" is filled.

---

**Template for Knowledge Acquisition**

---

Here is the basic knowledge of the hour of day, the temperature, and the electricity load, in the context of the relation between the climate and the energy consumption.

**(domain knowledge shown in Table 25)**

Using this knowledge, let's discuss the problem of causation on energy consumption related with climates.
$Q_n(x_i, x_j)$
Please answer with <yes> or <no>. No answers except these two responses are needed.

---

Table 27: Summary of the experimental results of generated knowledge prompting.

**Top 3 in Overall Accuracy**

| Template | Q.1 | Q.2 | Q.3 | Q.4 | Q.5 | Q.6 | Overall Accuracy |
|---|---|---|---|---|---|---|---|
| $Q_4(x_i, x_j)$ | 0.00 | 1.00 | 1.00 | 1.00 | 1.00 | 1.00 | 0.83 |
| $Q_{12}(x_i, x_j)$ | 0.00 | 1.00 | 1.00 | 1.00 | 1.00 | 1.00 | 0.83 |
| $Q_{19}(x_i, x_j)$ | 0.00 | 1.00 | 1.00 | 1.00 | 1.00 | 1.00 | 0.83 |

**Worst 3 in Overall Accuracy**

| Template | Q.1 | Q.2 | Q.3 | Q.4 | Q.5 | Q.6 | Overall Accuracy |
|---|---|---|---|---|---|---|---|
| $Q_{14}(x_i, x_j)$ | 0.00 | 0.92 | 1.00 | 0.00 | 1.00 | 1.00 | 0.65 |
| $Q_8(x_i, x_j)$ | 0.00 | 0.14 | 0.36 | 1.00 | 1.00 | 1.00 | 0.58 |
| $Q_5(x_i, x_j)$ | 0.00 | 0.00 | 0.12 | 1.00 | 1.00 | 1.00 | 0.52 |

# J  Other Potential Applications of the LLM-Generated Probability Matrix for Causal Inference

While the LLM-generated probability matrix is used as the basis for constructing the prior knowledge matrix to improve SCD performance, a careful understanding of the assumptions behind this matrix contributes to recognizing both its limitations and its potential for other applications. As previously explained in Appendix D.1, the $(i, j)$ component of the LLM-generated probability matrix is theoretically expressed as follows:

$$P(c = \text{"yes"}|q_{ij}^{(2)}, T) = \frac{\exp(z(c = \text{"yes"}, q_{ij}^{(2)})/T)}{\sum_{\text{for all } c} \exp(z(c, q_{ij}^{(2)})/T)} \tag{36}$$

Here, $q_{ij}^{(2)}$ is the prompt, the template of which is shown in Table 2, and $T$ is the temperature parameter. Moreover, $z(c, q_{ij}^{(2)})$ denotes the logit of the occurrence of token $c$ given the prompt $q_{ij}^{(2)}$. Notably, the conditional probability $P(c = \text{"yes"}|q_{ij}^{(2)}, T)$ is determined solely by $q_{ij}^{(2)}$, $T$, and the internal properties of the LLM that produce the logit $z(c, q_{ij}^{(2)})$. Therefore, we interpret this token generation probability as the LLM's confidence in the existence of a causal relationship from $x_j$ to $x_i$. Therefore, this probability is fundamentally different from the probability distributions used in conventional statistical causal inference, which are derived from observational data and vary with the values of the variables. Understanding this difference is key to recognizing both the limitations and the potential for applications of the probability matrix discussed in this work.

### Limitations in the Application to Conventional Causal Inference Frameworks

For example, within the potential outcomes framework, the probability-based average causal effect (ACE) from the intervened variable $x_j$ (fixed at 1 or 0) to the affected variable $x_i$ is expressed as follows:

$$\tau_{ACE}(x_i \leftarrow x_j) = P(x_i|do(x_j = 1)) - P(x_i|do(x_j = 0)) \tag{37}$$

However, since the LLM's confidence probability expressed in Eq. (36) is independent of the actual variables, and not a probability distribution that reflects real-world data, it cannot be directly applied to the statistical causal inference framework.

### Potential for Applying Edge-Probability-Based Causal Graph Discovery

Nevertheless, a study has also been reported on causal graph discovery and causal identification under the condition of probabilistic uncertainties at the edges of a causal graph(Akbari et al., 2023). In this research, the probability of an acyclic directed mixed graph (ADMG) $\mathcal{G}$ can be expressed as follows:

$$P(\mathcal{G}) = \prod_{e \in \mathcal{G}} p_e \prod_{e \notin \mathcal{G}} (1 - p_e) \tag{38}$$

Here, $p_e$ denotes the probability of the existence of edge $e$. If the probability in Eq. (36) is assumed to be equal to $p_e$, it becomes possible to identify the causal graph, which has the highest likelihood according to the domain expert knowledge in the LLM, by computing $\arg\max P(\mathcal{G})$. This idea may offer an alternative approach to reasonable causal inference -—by selecting the most likely causal graph based on domain expert knowledge as the structural basis for further analysis, which then leads to the step of conventional statistical causal inference.

