# OpenReview forum: "Integrating Large Language Models in Causal Discovery: A Statistical Causal Approach"
_TMLR — Accepted by TMLR_

### Review · Reviewer_zBLS · 2025-02-25

**Summary Of Contributions:**

As illustrated in the title, this paper integrates LLMs into statistic causality (SCD) by presenting a statistical causal prompting (SCP) approach, aiming to identify cause-and-effect relationships from data from different fields. The proposed SCP employs LLMs to generate prior knowledge based on initial SCD results, which are then used to refine the causal models. Experiments on benchmark datasets and a real-world health screening dataset, not included in the LLM's training data (not well-supported), show the improved performance of SCP under various metrics.

**Audience:**

Yes

**Broader Impact Concerns:**

As API providers may collect the input data, employing LLMs with sensitive data in practice, such as personal health data, can raise privacy and security concerns that need addressing.

**Claims And Evidence:**

No

**Requested Changes:**

1. The process of transforming LLM probabilities into prior knowledge matrices (Algorithm 2, Section 3.2) could be more clearly explained, especially for SCD algorithms with varying requirements. For instance, the paper notes constraints for Exact Search (binary, 0 forbidden, 1 otherwise, Appendix E) and DirectLiNGAM (acyclic transformation via BIC, Appendix F), but a unified framework would enhance clarity.
2. The paper mentioned the risk of false negatives in LLM judgments (Section 4.3, Simulation B), but more detailed strategies, such as ensemble methods or expert validation, could mitigate this. This is crucial for applications in healthcare, where missing true causal relationships could have significant consequences.
3. More LLMs should be employed to validate the effectiveness of the SCP, the improvement of current experiments may merely come from GPT-series models instead of the proposed pipeline.
4. Current metrics (SHD, FPR, FNR, precision, F1 score, CFI, RMSEA, BIC) are good but could be supplemented with metrics assessing the directionality and strength of causal relationships, such as causal effect sizes or sensitivity analyses beyond those in Appendix D.
5. Contributions 2 and 3 in the Introduction section share quite similar meanings, both are SCP benefiting SCD, no need to restate.

**Strengths And Weaknesses:**

### Strengths
1. The paper leverages LLMs to provide domain knowledge that enhances causal modeling, trying to address the challenge of systematically acquiring background knowledge for SCD, making this research interesting.
2. The proposed SCP is well-articulated, detailing a 4-step process, and it offers the possibility of automating labor-intensive tasks like variable selection or hypothesis generation as it could streamline causal analysis pipelines.
3. The experiments cover a range of datasets from distinct areas, including benchmark datasets with ground truths (e.g., Auto MPG ground truth at Cause-Effect Pairs, DWD climate data at DWD), demonstrating the effectiveness of the proposed SCP.

### Weaknesses
1. SCP’s performance depends largely on the LLMs, and the role of LLMs in causality discovery is not transparent, thus, incorporating the black-box LLMs can heart the high explainability of pure SCD algorithms.
2. The LLMs employed in this paper are all not open-source (all are GPT series, no other close- or open-source models), as knowledge generation, and integration are way more prompt engineering (shown in Tables 1,2,9, and 12) and LLMs are limited, the results of single-source LLMs cannot be that convincible and generalizable.
3. The authors mentioned that “this dataset has never been included in the LLM”, how can it be detected? Have the authors checked the similarity between the proposed data and LLM training data with n-gram or other metrics? There is no introduction about that.
4. Since using LLM-generated prior knowledge based on initial SCD results might lead to overfitting, especially if the intermediate results are not robust. This paper lacks experiments for classic SCD methods with intermediate results of knowledge generation and integration, which can illustrate the error propagation and help interpret the role of SCP components.
5. Introducing LLMs into SCD can take more computation than conventional approaches, this work didn’t discuss the time or computation costs.

---

> ### Author Response · Authors · 2025-03-24
> **Response to the Comments from Reviewer zBLS (part 1)**
>
> Dear Reviewer zBLS,
>
> We sincerely appreciate your fruitful comments.
> After carefully reviewing your feedback, we have revised our manuscript accordingly, with the changes highlighted in red font.
> Below, we provide detailed responses to each of your requested revisions.
>
>
> **Regarding Requested Changes**
> >1. The process of transforming LLM probabilities into prior knowledge matrices (Algorithm 2, Section 3.2) could be more clearly explained, especially for SCD algorithms with varying requirements. For instance, the paper notes constraints for Exact Search (binary, 0 forbidden, 1 otherwise, Appendix E) and DirectLiNGAM (acyclic transformation via BIC, Appendix F), but a unified framework would enhance clarity.
>
> As you have pointed out, constructing a unified framework for the composition of a prior knowledge matrix is ideal.
>
> However, differences in incorporating prior knowledge stem from variations in the optimization processes for discovering the optimal DAG in each SCD algorithm. Furthermore, we believe it is important to demonstrate the universal effectiveness of SCP across various SCD algorithms.
>
> Considering these factors, we have appended footnote 13 to explain the importance of a unified framework and provide practical tips for simpler interpretation.
>
>
> >2. The paper mentioned the risk of false negatives in LLM judgments (Section 4.3, Simulation B), but more detailed strategies, such as ensemble methods or expert validation, could mitigate this. This is crucial for applications in healthcare, where missing true causal relationships could have significant consequences.
>
> We recognize the importance of the point you have stated.
> Therefore, we have already discussed ways to mitigate this risk in the latter part of Section 4.3, specifically in the subsubsections "Toward Stabilization of LLMs' Confidence Probability Measurement with Improvement of Prompting Techniques" and "Toward Appropriate Application of the Proposed Method."
> In particular, expert validation has already been mentioned in this context.
>
> To clarify the structure of this discussion, we have added an introduction to the latter part of Section 4.3, where the mitigation of the risk mentioned in Simulation B is discussed.
> Furthermore, since it is essential to evaluate the performance at each step, we have added an explanation of this premise to the final part of Section 4.3.
> Additionally, we have included a reference to ensemble methods as the concluding sentence of Section 4.3.
>
>
> >3. More LLMs should be employed to validate the effectiveness of the SCP, the improvement of current experiments may merely come from GPT-series models instead of the proposed pipeline.
>
> In response to this comment, we have additionally conducted experiments with several open-source models (Llama, Gemma, and Qwen).
>
> Consequently, we have added Appendix H.2, which describes the result of these additional experiments.
> This result clearly supports the fact that the improvement with SCP does not merely come from GPT-series models but from the proposed method, which integrates LLMs into SCD.
> In the main body, we have also added a reference to these additional experiments in a footnote 7 in Section 3.
>
>
> >4. Current metrics (SHD, FPR, FNR, precision, F1 score, CFI, RMSEA, BIC) are good but could be supplemented with metrics assessing the directionality and strength of causal relationships, such as causal effect sizes or sensitivity analyses beyond those in Appendix D.
>
> We have appended Appendix D.5 for the supplemental analysis of the stability of the causal coefficients against the fluctuation of the LLM's confidence probability.
>
> >5. Contributions 2 and 3 in the Introduction section share quite similar meanings, both are SCP benefiting SCD, no need to restate.
>
> As you have pointed out, we have integrated Contributions (2) and (3) in Section 1.3 and enhanced the clarity of the meaning.

---

> > ### Comment · Reviewer_zBLS · 2025-04-03
> > **Most requested changes were addressed, one minor issue**
> >
> > Thank you for the time and efforts, most of my early requested revisions have been supplemented or explained.
> > Now I have a minor suggestion based on the authors' responses and the updated manuscript。
> >
> > >The current paper needs to be arranged more logically and clearly: some independent discussions and explanations of subsections or paragraphs scattered in sections/subsections can be summarized in the beginning paragraph, which could be convenient for reading and understanding the specific contents.

---

> ### Author Response · Authors · 2025-03-24
> **Response to the Comments from Reviewer zBLS (part 2)**
>
> **Regarding Weaknesses**
>
> >The authors mentioned that “this dataset has never been included in the LLM”, how can it be detected? Have the authors checked the similarity between the proposed data and LLM training data with n-gram or other metrics? There is no introduction about that.
>
> The statement is intended to insist that this health-screening dataset used in our work is:
> ・used in a restricted environment
> ・is not directly exposed to the internet for analysis
> ・is not shared with third parties
> and meaning that the exact same dataset is not included in the training data of the LLM.
> The biggest concern regarding data-leakage is, when an LLM meets the exact same data as what the model has already memorized,
> the LLM may merely reproduce previously learned insights.
> Then, we have confirmed the SCD on the unpublished health-screening dataset can be improved through our method, without reproduction of the memorized same data by LLM.
>
> On the other hand, as you pointed out, there is no definitive way to prove that the training data contains no similar data at all.
> Nevertheless, if an LLM can leverage its existing knowledge to address similar problems, without this should be seen as a desirable use case, as an expert system.
>
> To clarify this meaning, we have appended the explanation in Section 4.2.
>
> >Since using LLM-generated prior knowledge based on initial SCD results might lead to overfitting, especially if the intermediate results are not robust. This paper lacks experiments for classic SCD methods with intermediate results of knowledge generation and integration, which can illustrate the error propagation and help interpret the role of SCP components.
>
> Although it would be ideal for intermediate results generated by LLMs (such as an LLM’s confidence probability matrix) to be directly adopted as prior knowledge for SCD, the currently available SCD algorithms published as packages do not support this functionality.
> Implementing such a function would require extensive and complex discussions on how the components of LLM-generated intermediate results should be treated and evaluated as prior knowledge.
> Given this limitation, we introduced an approach for composing a prior knowledge matrix transformed from the probability matrix. This approach not only simplifies the interpretation of LLMs’ confidence in causal relationships but also enables the integration of LLM-derived prior knowledge into existing SCD algorithms. Additionally, we have evaluated the risk of error propagation in SCD results in Appendix D.5.
>
> Furthermore, while we understand concerns regarding potential overfitting when using LLM-generated prior knowledge based on initial SCD results, the experimental results of the proposed method clearly demonstrate performance improvements compared to the initial SCD results. In particular, Simulation A in Section 4.3 shows that obvious errors in the initial SCD results are consistently corrected after Step 3.
> We interpret this as an indication that while initial SCD results help guide the data-driven interpretation of LLMs, the domain knowledge embedded in the LLM adequately counterbalances this process, thereby preventing overfitting.
>
> >Introducing LLMs into SCD can take more computation than conventional approaches, this work didn’t discuss the time or computation costs.
>
> In response to theis comment, we have appended the discussion on computation costs and scalability in the end of Section 3.1.

---

> > ### Comment · Reviewer_zBLS · 2025-04-03
> > **Major concerns were addressed**
> >
> > These revisions and responses have diminished my doubts and concerns.
> >
> > However, I still suggest that the authors could employ some training data leakage detection methods to prove that there was no data leakage to make the statement well supported.

---

> > > ### Author Response · Authors · 2025-04-03
> > > **Additional explanation on this problem**
> > >
> > > >I still suggest that the authors could employ some training data leakage detection methods to prove that there was no data leakage to make the statement well supported.
> > >
> > > In the process of experiments and writing this manuscript, all the authors sincerely discussed and considered this issue.
> > > We acknowledge that various methods have been proposed to detect potential data leakage from LLMs or to estimate whether specific data was included in their pre-training datasets.
> > >
> > >
> > > However, in our case, since we used GPT-4,
> > > we do not have access to its closed pre-training data.
> > > Therefore, in a strict sense, it is impossible to verify the existence of the overlaps with our health-screening dataset using techniques such as n-gram matching in a controlled and appropriate environment.
> > >
> > >
> > > Moreover, even if we were to attempt other existing detection methods, conducting such verification with our actual dataset could itself pose a risk of causing data leakage, which would be a social concern.
> > > Ensuring a sufficiently robust and secure environment for such validation would also require an additional layer of safeguards, which may not be feasible at this stage.
> > >
> > >
> > > In addition, even if these technical concerns could be addressed, we are bound by agreements with data providers, the scope approved by our ethics review board, and, in some cases, legal restrictions, which prohibit the publication of the raw dataset containing personal records.
> > >
> > >
> > > For the above reasons, we have decided not to pursue further verification using such detection methods in the current study. Instead, we have included explanations regarding the nature of the dataset and the relevant legal constraints.
> > > Based on these facts, we believe it is reasonable to conclude that the health-screening dataset used in this study was not included in GPT-4’s pre-training dataset.
> > >
> > >
> > > However, if permitted, we will incorporate this additional discussion within 2–3 days as part of another suggested revision.

---

> ### Author Response · Authors · 2025-03-24
> **Response to the Comments from Reviewer zBLS (part 3)**
>
> **Regarding Broader Impact Concerns**
>
> >As API providers may collect the input data, employing LLMs with sensitive data in practice, such as personal health data, can raise privacy and security concerns that need addressing.
>
> We have appended the explanation in the subsection of 'Broader Impact Statement.' According to our proposed method, the raw dataset is not input into LLMs; only the results of SCD, such as causal coefficients and bootstrap probabilities, are provided. Therefore, this approach does not raise additional privacy and security concerns.

---

> > ### Comment · Reviewer_zBLS · 2025-04-03
> > **Privacy worries addressed**
> >
> > Inputting only SCD results could indeed avoid the privacy data leak.

---

> ### Author Response · Authors · 2025-04-03
> **Willing to reflect your suggestion (quick response)**
>
> Dear Reviewer zBLS,
>
> Thank you for your confirmation and additional comments.
>
> >The current paper needs to be arranged more logically and clearly: some independent discussions and explanations of subsections or paragraphs scattered in sections/subsections can be summarized in the beginning paragraph, which could be convenient for reading and understanding the specific contents.
>
> As a quick note, although we are not sure whether further revisions are allowed at this stage, we are willing to incorporate your comments within 2–3 days if permitted.
>
> Even if revisions are not allowed until the final decision, we promise to make the suggested changes sincerely, after the paper is accepted.

---

> ### Author Response · Authors · 2025-04-05
> **Now revised again!**
>
> Dear Reviewer zBLS,
>
> Now we have uploaded our revised manuscript, incorporating your additional comments in violet font.
>
> >The current paper needs to be arranged more logically and clearly: some independent discussions and explanations of subsections or paragraphs scattered in sections/subsections can be summarized in the beginning paragraph, which could be convenient for reading and understanding the specific contents.
>
> We have thoroughly reviewed the logical flow and readability of our manuscript,
> and found that it would be helpful to include an outline at the beginning of Sections 3 and 4, as both are relatively large.
>
> We also found that Section 3.1 was difficult to grasp at a glance, as it contains detailed technical explanations from multiple perspectives.
> Therefore, we have added an outline at the beginning of this subsection as well.
>
> >I still suggest that the authors could employ some training data leakage detection methods to prove that there was no data leakage to make the statement well supported.
>
> Although we have already addressed the technical and ethical limitations of detecting potential leakage of our health-screening dataset in another response,
> we have added this supplemental discussion at the end of Appendix C.4.
> Additionally, we have included a reference to this supplemental explanation at the beginning of Section 4.2.
>
>
> We hope that this version will address the issues you pointed out.
>
> Once again, we are truly grateful for your thoughtful and dedicated engagement with our work!

---

### Review · Reviewer_8HhM · 2025-03-08

**Summary Of Contributions:**

This paper proposes a novel method for knowledge-based causal inference that integrates LLMs with traditional statistical causal discovery (SCD) through a self-critique-type prompting strategy called Statistical Causal Prompting (SCP). The approach involves (1) zero-shot synthesis of a causal graph by an LLM, (2) prompting the LLM to critique its own response, (3) converting the dialogue consisting of critiques into probabilities of whether an edge exists in the causal graph, and (4) finally, incorporating these probabilities to update the graph. The authors validate their method on benchmark datasets and a closed health screening dataset, demonstrating that LLM-assisted SCD better aligns with ground truths than standard SCD. The paper also discusses the reliability, risks, and limitations of the approach.

**Audience:**

Yes

**Broader Impact Concerns:**

The paper discusses ethical implications, including risks related to biases in pretraining data of LLMs and concerns about transparency and interpretability of black-box models in high-stakes fields like healthcare.

While the paper discusses failure cases, more explicit strategies for mitigating these risks in practical applications would be valuable.

**Claims And Evidence:**

Yes

**Requested Changes:**

I would appreciate the following updates to secure my recommendation for acceptance:
- Clarifying the scalability of the approach would help understand how it would handle high-dimensional causal graphs.
- Highlighting key takeaways from large tables of results (e.g. table 3) would improve clarity and presentation.

I believe the following changes would greatly improve the impact and quality of the paper, although they are not critical for my recommendation:
- Exploring the impact of different prompt wording/formats and LLM hyperparameter settings on causal discovery results would be very informative.
- The LLM-log-probability-based prior knowledge used to update causal graphs imposes certain assumptions on the LLM's output distributions given an input dataset. These can be formalized under existing causal inference frameworks (e.g. potential outcomes) and doing so would make the assumptions and limitations of the method much clearer, especially to the causal inference community.

**Strengths And Weaknesses:**

Strengths:

- The integration of SCD with LLM-generated knowledge is an interesting approach that attempts to bridge statistical inference and domain expertise.
- The paper evaluates the approach on a dataset excluded from the LLM's pretraining, which strengthens claims of robustness.
- The authors provide a detailed discussion of failure cases, limitations, and areas for future improvements.

Weaknesses:

- The paper does not address scalability concerns; it remains unclear how well the method would perform on datasets with a larger number of variables.
- The sensitivity of SCP to different prompt wording and hyperparameters (such as temperature settings) is not extensively explored.
- The probabilistic transformation of LLM outputs into structured prior knowledge is heuristic, and its reliability across diverse domains remains uncertain.

---

> ### Author Response · Authors · 2025-03-24
> **Response to the Comments from Reviewer 8HhM**
>
> Dear Reviewer 8HhM,
>
> We sincerely appreciate your precious comments.
> After carefully reviewing your feedback, we have revised our manuscript accordingly, with the changes highlighted in red font.
> Below, we provide detailed responses to each of your requested revisions.
>
>
> **Regarding Requested Changes (Essential)**
>
> >Clarifying the scalability of the approach would help understand how it would handle high-dimensional causal graphs.
>
> Although the methodology does not change with the increase of the dimension of causal graphs, it has different impacts on the computation costs on SCD and LLM processes respectively. Focusing on this behavior, we have appended the explanation of the discussion on computation costs and scalability in the end of Section 3.1.
>
>
> >Highlighting key takeaways from large tables of results (e.g. table 3) would improve clarity and presentation.
>
> As you pointed out, summarizing the results for all patterns across the three datasets led to complexity and made it difficult to extract comparable results. Therefore, we have separated Table 3 into three tables, each corresponding to one of the datasets. We have also added key takeaways as sub-captions for each table.
>
>
> **Regarding Requested Changes (Others)**
>
> >Exploring the impact of different prompt wording/formats and LLM hyperparameter settings on causal discovery results would be very informative.
>
> We have added the preliminary experiments for selecting the prompt template used in this work as Appendix I.
>  With this supplementary material, we believe that readers can better understand the impact of different phrasings when asking LLMs about causality.
>  Regarding hyperparameter dependence, we have already provided a brief theoretical explanation of temperature dependence in Appendix D.
>
>
> >The LLM-log-probability-based prior knowledge used to update causal graphs imposes certain assumptions on the LLM's output distributions given an input dataset. These can be formalized under existing causal inference frameworks (e.g. potential outcomes) and doing so would make the assumptions and limitations of the method much clearer, especially to the causal inference community.
>
> Unfortunately, we believe it is difficult to directly apply the LLM-generated probability matrix to the potential outcomes framework or other major frameworks of causal inference, since the probability produced by the LLM simply represents its confidence in the existence of a causal edge, rather than a probability distribution that depends on the values of variables.
>
> However, methods for identifying causal effects and optimizing subgraphs based on probabilistic edges between variables have been discussed in prior work (https://arxiv.org/pdf/2208.04627).
> By combining the LLM-generated probability matrix with such methods, it may be possible to infer candidate causal graph structures—under the assumption that the LLM's confidence probabilities directly reflect the true likelihood of causal edge existence.
>
> We have included this discussion in Appendix J.
>
>
> **Regarding Broader Impact Concerns**
>
> >The paper discusses ethical implications, including risks related to biases in pretraining data of LLMs and concerns about transparency and interpretability of black-box models in high-stakes fields like healthcare.
> While the paper discusses failure cases, more explicit strategies for mitigating these risks in practical applications would be valuable.
>
> We strongly agree with you on the importance of having explicit strategies for mitigating the risks discussed in our paper.
> Accordingly, we have already addressed ways to mitigate these risks in the latter part of Section 4.3, specifically in the subsubsections titled "Toward Stabilization of LLMs’ Confidence Probability Measurement with Improvement of Prompting Techniques" and "Toward Appropriate Application of the Proposed Method."
> In particular, expert validation has already been mentioned in this context.
>
>
> To clarify the structure of this discussion on risk mitigation strategies, we have added an introductory paragraph at the beginning of the latter part of Section 4.3, where the mitigation of the risk highlighted in Simulation B is discussed.
>
> Additionally, since it is essential to evaluate the performance at each step, we have added an explanation of this premise to the final part of Section 4.3.

---

> > ### Comment · Reviewer_8HhM · 2025-04-05
> > **Response to Author Rebuttal**
> >
> > Thank you for the additions and clarifications! I have no further comments.

---

### Review · Reviewer_VFiT · 2025-03-13

**Summary Of Contributions:**

The paper proposed a new method for leveraging LLMs for causal discovery, with the main contribution being "statistical causal prompting (SCP)" to extract a measure of uncertainty about specify edges in a causal graph. The paper also experiments on an unpublished (and hence unseen by the LLM) dataset, which it claims hasn't been done before.

**Audience:**

Yes

**Broader Impact Concerns:**

The paper ends with reasonable statements about technical limitations and broader impact of the work.

**Claims And Evidence:**

No

**Requested Changes:**

__Essential__:
- The paper needs careful proofreading and editing for tense and active vs. passive sentence construction, especially in the abstract and contributions, to make it more what this paper contributes compared to what others have done.
For example, the abstract has sentences "It has been revealed..." then "Experiments have reaveled..." followed by "... we have demonstrated...". I can't tell if the first two passive sentences are supposed to be describing related work or specific results in this paper. I suspect the latter, but there's a similarly confusing active vs passive pattern in Section 1.3, so it's not as clear as it should be.

- There are several formal sounding but vague words and phrases that need to either be made more formal and explicit, or clarified somehow, or removed:
  - end of page 2: "more statistically valid causal models"
  - Section 1.3 (4): "clearly confirmed... robustly leads to more statistically valid causal models with natural ground truths"; this is an experiment on a single data set, so I don't find such a strong claim ("clearly confirmed" and "robustly") to be justified; also what is a "natural" ground truth?
  - second to last paragraph on page 4: "we detail herein how to achieve both statistical validity and natural interpretation with respect to domain knowledge at a maximally high level"; this needs proof and formal definitions for "statistical validity" and "maximally high" and more clarification what "natural interpretation" means.
  - third paragraph of page 6: "from an objective point of view"; what does this mean?
  - "naturalness" and "objectively" in Tables 1 and 2, respectively
  - "unnatural" in Figure 3 caption
  - "bias" is mentioned throughout the paper, but it's never clear to me exactly what this means: selection bias in causal modeling comes to mind, as does biased statistical estimation, but these are specific, formal ideas whereas the paper seems to be using "bias" in a more vague and undefined way

- Important details of the causal discovery methods used are missing or seem incorrect:
  - top of page 4: "non-parametric and score-based SCD method"; scores often make parametric assumptions---which specific score is used here?
  - bottom of page 5: which CI test is used for PC? If I'm not mistaken, the original A* algorithm paper uses a parametric score---which do you use?
  - Algorithm 1: how are the different SCD results used? (thinking of DAG vs CPDAG)
  - last paragraph of page 6: "if $PK_ij$ = Forbidden, the causal effect from $x_j$ to $x_i$ is forbidden from appearing in the SCD result"; more precisely, do you mean an ancestral graphical relation is forbidden, or a parental one?
  - footnote 10, page 9: you earlier claimed to use a non-parametric score-based method, so why not use that score instead of the Gaussian BIC?
  - need more clear distinction between casual discovery and causal inference: these seem to sometimes but not always be used interchangably in the paper, but that's confusing compared to the causality literature where these are often different tasks; I don't really see any causal inference in this paper

__Suggested__:
- First sentence in Section 1.3 (5) is missing a verb in the last phrase ", and the technical..."
- Top of page 4: should be "Clark" not "Clerk"
- add some discussion about or even experiment with a more naive SCD/LLM integration approach, like engineering a prompt to get the LLM to just directly tell you the DAG and associated edge confidences
- streamline the main text by moving many of the LLM prompting details to the appendix; just as many of the causal discovery details are moved to the appendix, since the paper is about trying to be general/agnostic to the particular algorithm used, so should much of the GPT-4 details and prompting patterns etc.
- emphasize the "common sense causal model based on broad/vast knowledge" aspect more in the abstract and intro

**Strengths And Weaknesses:**

__Strengths__:
- flexible approach (with respect to both the causal discovery method and the LLM used)
- the idea and implementation of SCP is an important contribution to the growing literature on integrating LLMs and causality

__Weaknesses__:
- writing could be improved (grammar, organization, main text vs appendix)
- formal details and rigor are missing, resulting in unjustified claims

---

> ### Author Response · Authors · 2025-03-24
> **Response to the Comments from Reviewer VFiT (part 1)**
>
> Dear Reviewer VFiT,
>
> We sincerely appreciate your insightful and constructive comments, which were highly focused on the details of our work.
> After carefully reviewing your feedback, we have revised our manuscript accordingly, with the changes highlighted in red font.
> Below, we provide detailed responses to each of your requested revisions.
>
> **Regarding Requested Changes (Essential)**
>
> >The paper needs careful proofreading and editing for tense and active vs. passive sentence construction, especially in the abstract and contributions, to make it more what this paper contributes compared to what others have done. For example, the abstract has sentences "It has been revealed..." then "Experiments have revealed..." followed by "... we have demonstrated...". I can't tell if the first two passive sentences are supposed to be describing related work or specific results in this paper. I suspect the latter, but there's a similarly confusing active vs passive pattern in Section 1.3, so it's not as clear as it should be.
>
>
> We have revised the abstract to clarify that the experiments mentioned are all conducted in our work. Additionally, we have updated Section 1.3 to emphasize that all the work discussed was carried out by the authors.
>
>
> >・There are several formal sounding but vague words and phrases that need to either be made more formal and explicit, or clarified somehow, or removed:
>
> >○end of page 2: "more statistically valid causal models"
>
>
> We realized that the phrase "whether it leads to more statistically valid causal models" was unnecessary and removed it from the last sentence of Section 1.2.
>
>
> >○Section 1.3 (4): "clearly confirmed... robustly leads to more statistically valid causal models with natural ground truths"; this is an experiment on a single data set, so I don't find such a strong claim ("clearly confirmed" and "robustly") to be justified; also what is a "natural" ground truth?
>
>
> We have tempered the language regarding experimental confirmation and revised the term "natural ground truth" to "established domain knowledge."
>
>
> >○second to last paragraph on page 4: "we detail herein how to achieve both statistical validity and natural interpretation with respect to domain knowledge at a maximally high level"; this needs proof and formal definitions for "statistical validity" and "maximally high" and more clarification what "natural interpretation" means.
>
>
> We have revised the term '"statistical validity" to "statistically well-fitting causal model" and removed the phrase "at a maximally high level." Additionally, we have replaced "natural interpretation" with "reasonable interpretation" to better convey consistency with domain expert knowledge.
>
>
> >○third paragraph of page 6: "from an objective point of view"; what does this mean?
>
>
> For the clarity, we have replace this expression with "considering the discussion of the second step regarding causal relationships from another perspective."
>
>
> >○"naturalness" and "objectively" in Tables 1 and 2, respectively
>
>
> In Table 1, "naturalness" refers to consistency with domain knowledge, while in Table 2, "objectively" indicates that the second step of our method should be interpreted from a third-party perspective to assess causal relationships between variable pairs. Although we acknowledge that the prompt expressions could be improved, we believe they are adequate for our experiments.
> Therefore, we have retained the original wording in Tables 1 and 2 to preserve the reproducibility of our work.
>
>
> >○"unnatural" in Figure 3 caption
>
>
> We have replaced "unnatural" with "unreasonable considering domain knowledge."

---

> ### Author Response · Authors · 2025-03-24
> **Response to the Comments from Reviewer VFiT (part 2)**
>
> >○"bias" is mentioned throughout the paper, but it's never clear to me exactly what this means: selection bias in causal modeling comes to mind, as does biased statistical estimation, but these are specific, formal ideas whereas the paper seems to be using "bias" in a more vague and undefined way
>
>
> In this paper, the "bias" is mainly used in two contexts:
>
>
> 1. context of bias in the dataset
>
> We do not specify the details of the bias here, as dataset biases can arise not only from selection bias but also from sampling bias, label bias, confirmation bias, and measurement bias. We believe our method can work effectively on the dataset which could contain biases in a general sense, rather than targeting a specific type.
>
>
> To clarify this point, we have added footnote 15.
> Furthermore, although it is possible that our subsample of the health screening dataset contains certain biases, it is difficult to assert the presence of any specific bias with certainty.
> Therefore, we have thoroughly reviewed all mentions of bias in relation to the health screening dataset, and have softened the tone accordingly.
>
>
> 2. context of judgement of LLMs
>
> In this context, "bias" refers to biases present in the datasets used for pre-training LLMs or biases in the prompts inputted into LLMs. In both cases, we have already used in the original manuscript, the terms "bias" or "biased"' whose meanings can be easily inferred from the context.
>
>
> >・Important details of the causal discovery methods used are missing or seem incorrect:
>
> >○top of page 4: "non-parametric and score-based SCD method"; scores often make parametric assumptions---which specific score is used here?
>
> >○bottom of page 5: which CI test is used for PC? If I'm not mistaken, the original A* algorithm paper uses a parametric score---which do you use?
>
>
> In this context, "non-parametric" indicates that the neither the functional form characterizing the relationship between a cause-effect pair of variables nor the distribution of variables is assumed during the discovery process. Therefore, we have revised the description of the A* exact search algorithm from "a non-parametric and score-based SCD method" to "a score-based SCD method."
>
>
> Furthermore, We have included this supplementary explanation in footnote 5. Additionally, since the A* exact search algorithm utilizes the Bayesian Information Criterion (BIC), we have detailed this aspect in footnote 6.
>
>
> Finally, we employed the Fisher's Z - test in conjunction with the PC algorithm and have provided further explanation in the beginning of Section 3.1.
>
>
> >○ Algorithm 1: how are the different SCD results used? (thinking of DAG vs CPDAG)
>
>
> Only one SCD method is selected from PC, Exact Search, and DirectLiNGAM, before the start of Algorithm 1.
> Moreover, the SCD results are in the form of DAG.
> To clarify the facts above, we have revised the expression of Algorithm 1.
>
>
> >○last paragraph of page 6: "if PKij = Forbidden, the causal effect from xj to xi is forbidden from appearing in the SCD result"; more precisely, do you mean an ancestral graphical relation is forbidden, or a parental one?
>
>
> While the prior knowledge for the PC and Exact Search algorithms in the "causal-learn" package corresponds to the existence of directed edges between pairs of variables, the prior knowledge for DirectLiNGAM is determined with the knowledge of directed paths.
> To clarify this, we have revised the explanation of the part you mentioned above.
>
>
> >○footnote 10, page 9: you earlier claimed to use a non-parametric score-based method, so why not use that score instead of the Gaussian BIC?
>
>
> The target of this method in this work is the functional causal analysis of the dataset with continuous variables.
> In this meaning, the analysis is expected to be connected to the SEM analysis, regardless of the SCD methods.
>
>
> We have appended the explanation on the problem above in the same footnote (now 16).

---

> ### Author Response · Authors · 2025-03-24
> **Response to the Comments from Reviewer VFiT (part 3)**
>
> >○need more clear distinction between casual discovery and causal inference: these seem to sometimes but not always be used interchangably in the paper, but that's confusing compared to the causality literature where these are often different tasks; I don't really see any causal inference in this paper
>
>
> Under the rigorous definition of causal inference as referring specifically to statistical causal inference methodology, as you mentioned above, it is true that this paper does not propose any new algorithms for statistical causal discovery or alternative frameworks for estimating causal effects, such as the potential outcomes framework.
>
>
> In this work, however, we interpret “causal inference” more broadly than the conventional statistical causal inference framework, which focuses on the quantitative estimation of causal effects. Specifically, we define the key term in our study as LLM-based Knowledge-Based Causal Inference (LLM-KBCI), as it serves as an important upstream task for generating candidate causal models through SCD, which are subsequently used in downstream causal effect analysis.
>
> In this sense, we consider both LLM-KBCI and SCD to be components of “causal inference” that precede the main causal effect analysis typically conducted within the statistical causal inference framework. We have added this interpretation as Footnote 3 on page 2.
>
>
> Although there are other instances where the term “causal inference” is used—particularly in the literature review section—we have intentionally retained them, as many related works that apply LLMs to causal problems share our broader interpretation of “causal inference.”
>
>
> Then, considering the definition of the range of “causal inference” above, we have thoroughly reviewed our manuscript and revised several parts that were originally written using the term “causal inference,” to clarify the meaning of this term in each context.
>
>
> **Regarding Requested Changes (Suggested)**
>
>
> >・First sentence in Section 1.3 (5) is missing a verb in the last phrase ", and the technical..."
>
>
> We have appended the verb "discuss" before the phrase "the technical and statistical limitations of the proposed method."
>
>
> >・Top of page 4: should be "Clark" not "Clerk"
>
>
> We have revised this typo.
>
>
> >・add some discussion about or even experiment with a more naive SCD/LLM integration approach, like engineering a prompt to get the LLM to just directly tell you the DAG and associated edge confidences
>
>
> Methods for directly inducing DAGs using LLMs have already been proposed and discussed, mainly without relying on real-world numerical datasets　(e.g. https://arxiv.org/abs/2305.00050 or https://openreview.net/forum?id=tv46tCzs83)
>
>
> However, based on your suggestion, we have recognized the importance of further discussing the use of the LLM-generated probability matrix for constructing DAGs, and we have appended this discussion in Appendix J.
>
> >・emphasize the "common sense causal model based on broad/vast knowledge" aspect more in the abstract and intro
>
>
> We have emphasized these ideal causal models in the abstract and Section 1.1 (at the beginning of p.2), to clarify the ultimate goal of the method proposed in this paper.

---

> ### Comment · Reviewer_VFiT · 2025-03-25
> **much improved, just one small follow-up**
>
> Thanks for the thorough responses! All my major concerns are addressed, but I have one follow-up question:
> > Only one SCD method is selected from PC, Exact Search, and DirectLiNGAM, before the start of Algorithm 1. Moreover, the SCD results are in the form of DAG. To clarify the facts above, we have revised the expression of Algorithm 1.
>
> PC returns a CPDAG. How is this converted into a DAG for Algorithm 1?

---

> > ### Author Response · Authors · 2025-03-25
> > **Revised once more regarding your follow-up question**
> >
> > Dear Reviewer VFiT,
> >
> > We really thank you for your quick and attentive confirmation!
> >
> > >PC returns a CPDAG. How is this converted into a DAG for Algorithm 1?
> >
> > We sincerely apologize for the incorrect response given earlier.
> > As you correctly pointed out, while the output of the A* Exact Search or the DirectLiNGAM algorithm in in the form of DAGs,
> > the PC algorithm returns a CPDAG, since it is possible that not all edge directions are determined by the orientation rules.
> >
> >
> > In light of this difference, we have re-uploaded the 2nd revised version of our manuscript.
> > Specifically, we have removed the phrase "in the form of DAG" from Algorithm 1 once again,
> > and added the above explanation in red font to the main text (at the bottom of page 5, Section 3.1).
> >
> >
> > Furthermore, as explained in Appendix B, our method can already be applied when the output of the first step is a CPDAG (including directed edges).
> > Accordingly, we have also revised the explanation in Footnote 14 to clarify this point.

---

> > > ### Comment · Reviewer_VFiT · 2025-03-25
> > >
> > > This makes sense, thanks for the quick revision and response!

---

### Comment · Action_Editor_mpuD · 2025-03-13
**The authors-reviewers discussion period started**

Dear reviewers,
thank you so much for your precious work.
The authors-reviewers period started, thus I kindly ask to develop an interaction which can bring in two weeks the reviewers to make their final reccomendation.
All the best

---

### Author Response · Authors · 2025-03-24
**Revision Completed with Requested Changes**

Dear reviewers,

We sincerely appreciate your valuable and constructive feedback once again.

The revised version of the paper has been uploaded, with all modified parts highlighted in red font.
We have also provided detailed individual responses to each reviewer's comments in their respective threads.

We look forward to further discussion.

Best regards, Authors of Paper 4060

---

### Author Response · Authors · 2025-05-06
**Camera-Ready Version Completed and Uploaded**

Dear AE and Reviewers,

We sincerely appreciate once again your fruitful discussions, which have greatly improved our paper to its final version.
The camera-ready version of the paper, along with the associated code, is now available online.

Best regards,
Authors of Paper 4060

---

### Decision · Action_Editor_mpuD · 2025-04-12

**Recommendation:** Accept as is

**Comment:**

The paper makes some relevant contributions; i) the proposed approach lokks flexible (with respect to both the causal discovery method and the LLM used), ii)  the idea and implementation of SCP is an interesting contribution to the growing literature on integrating LLMs and causality, iii) the paper evaluates the approach on a dataset excluded from the LLM's pretraining, which strengthens claims of robustness, iv) the authors provide a detailed discussion of failure cases, limitations, and areas for future improvements, v) the proposed SCP is well-articulated, detailing a 4-step process, and it offers the possibility of automating labor-intensive tasks like variable selection or hypothesis generation as it could streamline causal analysis pipelines, vI) numerical experimentes are extremely rich by including benchmark datasets with ground truths.

**Audience:**

All the reviewers agree that the topic presented, develloped and discussed by the manuscript are of interest for members of the community.

**Claims And Evidence:**

After an intense rebuttal and discusison period from all the three reviwers they all agree that the claims made in the paper are well-supported by evidence.